# 1 Climate engineering by mimicking the natural dust climate control:

- 2 the Iron Salt Aerosols method
- 3

## 4 Authors:

Franz Dietrich OESTE \*<sup>1</sup>, Renaud de\_RICHTER \*<sup>2</sup>, Tingzhen MING <sup>3</sup>, Sylvain CAILLOL <sup>2</sup>

- \* corresponding author

## 8 Affiliations & Addresses:

1 gM-Ingenieurbüro, Tannenweg 2, D-35274 Kirchhain, Germany. Email: <u>oeste@gm-</u> 10 <u>ingenieurbuero.com</u>

- 2 Institut Charles Gerhardt Montpellier UMR5253 CNRS-UM2 ENSCM-UM1 Ecole
- Nationale Supérieure de Chimie de Montpellier, 8 rue de l'Ecole Normale, 34296 Montpellier
- Cedex 5, France. Email: <u>renaud.derichter@gmail.com</u>
- 3 School of Civil Engineering and Architecture, Wuhan University of Technology, No. 122,
- Luoshi Road, Hongshan District, Wuhan, 430070 China.

## 17 Abstract

Power stations, ship, and air traffic are among the most potent greenhouse gas emitters and

are primarily responsible for global warming.

Iron salt aerosols (ISA), composed partly of iron and chloride, exert a cooling effect on climate in several ways. This article aims firstly to examine all direct and indirect natural climate cooling mechanisms driven by ISA tropospheric aerosol particles, showing their cooperation and interaction within the different environmental compartments. Secondly, it looks at a proposal to enhance the cooling effects by ISA in order to reach the optimistic target of the Paris climate agreement, to limit the global temperature increase between 1.5 and 2 °C.

- Mineral dust played an important role during the glacial periods: by using mineral dust as a natural analogue tool and by mimicking the same method used in nature, the proposed ISA method might be able to reduce and stop climate warming. The first estimations made in this article show that by doubling the current natural iron emissions by ISA into the troposphere,
- i.e. by about 0.3 Tg Fe per year, artificial ISA would enable the prevention or even reversal of
- global warming.
- The ISA method proposed integrates technical and economically feasible tools.
- Keywords

Iron salt aerosols, cooling the earth, reverse global warming, methane removal, CO<sub>2</sub> removal
 phytoplankton fertilization, tropospheric ozone reduction, cloud albedo, carbon capture and
 storage (CCS), climate engineering

#### 40 **1. Introduction**

The 5<sup>th</sup> assessment report of the Intergovernmental Panel on Climate Change (IPPC),
released in November 2014, states that Global Warming (GW) has already begun to
dramatically change continental and marine ecosystems.

A recently noticed effect is that the vertical mixing in oceans decreases and even reaches a
stagnation point [1], thus weakening the net oceanic cumulative intake of atmospheric CO<sub>2</sub>
[2, 3].

A consequence of decreasing vertical ocean mixing is a reduced or interrupted oxygen 48 supply to the depths of the ocean. Currently, the formation of low-oxygen areas in the oceans 49 is increasing [4, 5]. Furthermore, climate warming entails stratification of the water column 50 and blocks vertical flows. Stratification may develop by warming the upper water layer as well 51 as evaporation and precipitation. Generation of a fresh water layer on top of the water 52 column by precipitation, surface water runoff and melt water inflow induce stratification [6, 7]. 53 Even the opposite, brine generation by evaporation may induce stratification [8]. Stratification 54 blocks the oxygen transfer through the water column and triggers the formation of oxygen-55 depleted zones [9] that also emit nitrous oxide (N<sub>2</sub>O), a potent GHG and a powerful ozone 56 depleting agent.

As iron is part of many enzymes directing the bioenergetic transformation of nitrogen in the 58 ocean, it has an additional direct influence on the cycling of these elements through the 59 oceanic environment [10, 11].

The severest consequence to oceanic ecosystems of such stratification is the development of anoxic milieu within stratified ocean basins. An example of the development of halocline and chemocline stratification is the Black Sea [12]. This ocean basin has a stable halocline which coincides with a chemocline, dividing an oxic salt-poor surface water layer from a saline anoxic sulfidic deep layer with a black sapropel sediment rich in organic C at the basin bottom [12].

Geological past episodes with stratified ocean basins are regularly marked by black shale or black limestone as remnants of sapropel sediments. Stratified ocean basins during the Phanerozoic epoch occurred as a consequence of elevated CO<sub>2</sub> levels in the atmosphere. This caused high sea surface temperatures [13] and, as a global consequence a global increase of evaporation, precipitation and production of brines of higher concentrations.

It has been pointed out that the increasing melt water run-off from past polar and subpolar 72 ice layers may have induced the cover of denser ocean water by a melt water layer [6]. 73 According to Praetorius et al. [14] climate warming events during the last deglacial transition 74 induced subsurface oxygen minimum zones accompanied by sea floor anoxia in the 75 Northern Pacific. This melt water-induced stratification had been accompanied by melt water 76 iron-induced phytoplankton blooms. The generation of increasing precipitation and surface 77 water run-off accompanied by increasing brine production plus elevated surface water 78 temperatures during hot CO<sub>2</sub>-high climate episodes had similar consequences in the past 79 geological epochs [13].

Ocean basin stratifications may be induced by increasing precipitation with increased surface water run-off [7] or by increased brine production [8]. These ocean stratification event is characterized by regional to global ocean anoxia, black sediments with elevated organic C and hot greenhouse climate, as we learn from the whole Phanerozoic past [13] and was often accompanied by mass extinctions.

Even the largest mass extinction of ocean biota within the Phanerozoic epoch, during the Permian-Triassic transition, has been induced by high temperatures as a consequence of elevated  $CO_2$ -Levels, which induced the change of a well-mixed oxic to a stratified euxinicanoxic ocean [15].

What we have to face now is the extraordinary process developing from the recent situation:

the combination of the CO<sub>2</sub>-dependent temperature rise-generated precipitation increase,

plus melt water increase. Mankind has to find now the appropriate tool to stop this dangerous

stratification process.

Warming surface waters and decreasing input of cold, oxygenated surface water, trigger a temperature rise of sediments, transforming solid methane hydrate into gaseous methane (CH<sub>4</sub>) emissions in seawater [16]. CH<sub>4</sub> oxidation consumes additional oxygen, decreasing the oxygen content above those areas [17].

The same effects are expected with an anticipated increase in spring and summer coastal 98 upwelling intensity, associated with increases in the rate of offshore advection, decreasing 99 the nutrient supply while producing a spatial or temporal (phenological) mismatch between 100 production and consumption in the world's most productive marine ecosystems [18].

These events have the threatening consequence of a sprawling lack of oxygen in the 102 oceans. In such low-oxygen areas (sub-oxic to anoxic) only bacterial life is possible: higher 103 life forms can not exist there. Accordingly, an early result of the climate warming progression 104 could lead to a dramatic limitation of the oceanic food sources that will be needed for the 105 projected 9-10 billion people by 2050. The same deleterious consequences on seafood 106 supply can also result in ocean surface acidification through increased CO<sub>2</sub> dissolution in sea water and decreased flow of surface water currents to ocean basin bottoms, limiting reef fishand shelled mollusk survival [19].

Any decrease of the THC has severe consequences on all kinds of ecosystems as it further 110 triggers climate warming by different interactions. THC decrease induces a reduction or 111 eventual disappearance of the phytoplankton fertilizers Si, P, N and Fe extracted on the 112 ocean surface from their resources at the bottom of the ocean basins. Hydrothermal fluid 113 cycling by mid-ocean ridges, off-axis hydrothermal fluid fluxes, subduction-dependent 114 hydrothermal convection fluids, hydrothermal fluxes at hot spot sea mount and fluid 115 emissions from anaerobic sediments, contain said elements as dissolved or colloidal phase 116 [20-27]. The deeper water of all ocean basins is enriched by these fertilizers. A THC 117 decrease within the ocean basins will result in a decrease of the assimilative transformation 118 of  $CO_2$  into organic carbon.

Moreover, any THC decrease would further trigger the acidification of the ocean surface by 120 lowering or preventing the neutralization of dissolved  $CO_2$  and  $HCO_3^-$ , due to the alkalinity 121 decrease from hydrothermal sources [20, 28].

During the convective water flow through the huge alkaline ocean crust volume, estimated to about 20 - 540 x  $10^3$  km<sup>3</sup> yr<sup>-1</sup> [29], ocean water is depleted in O<sub>2</sub>, but enriched in its reductant content such as CH<sub>4</sub> [20, 30]. Further elements are enriched in this convective water flow through the Earth crust, essential for the existence of life. The re-oxygenation of this huge water volume is retarded or even impossible with a minimized THC.

According to model calculations [31] the THC might have significantly changed between the 128 last glacial and interglacial periods. During the Cenozoic epoch, ice covered pole caps 129 limited the incorporation of carbon in the form of carbonate into the oceanic crust compared 130 to the warm Late Mesozoic peroid [32]. The findings of Coogan & Gillis show that during ice-131 free periods, THCs were possible with much higher effectiveness than in modern times. Even 132 during those warm periods with low temperature gradients between polar and equatorial 133 oceans, an effective production of brines leading to buoyancy differences necessary for 134 development of effective THC may have ben generated [33]. However, increased inflow rates 135 of high density brines coming from shallow shelf regions with high evaporation rates, induced 136 several collapses or vertical reductions of the strong Cretaceous THC. From here and for 137 more than a million years, the lower parts of ocean basins have been filled with anoxic brines 138 [8]. Further aspects of ocean stratification are discussed in chapter 4.1.

Remnants of these anoxic events are black shale sediments [34]. During such THC 140 collapses, the uptake of  $CO_2$  into the oceanic crust stayed restricted to organic carbon 141 sediments. Additionally, the organic carbon productivity of the remaining oxic zone was 142 decreased, as well as eolic dust input, due to phytoplankton fertilizer production being limited 143 to continental weathering. These examples point out the sensitivity of the THC to disturbances. Without action, the weakness of our recent THC may worsen. Any THC collapse would not only result in severe damages to ecosystems, food chains, and food resources of the oceans, but would also lead to an acceleration of the increase of atmospheric CO<sub>2</sub> concentration, resulting in a faster climate warming than forecasted.

The best way to prevent such disturbing situations and consequences is to stop GW.

A realistic chance of averting this development is the controlled application of a climate cooling process, used several times by nature throughout the last ice ages with high efficiency and, based on loess dust. Loess is a wind-blown dust sediment formed by progressive accumulation and composed generally of clay, sand and silt (approximately a ratio of 20:40:40 respectively), loosely cemented by calcium carbonate.

The dust concentration in the troposphere increased during every cold period in ice ages and 156 reached a multiple of today's levels [35]. Dust deposition in the Southern Ocean during 157 glacial periods was 3 to 10 times greater than during interglacial periods, and its major 158 source region was probably Australia or New Zealand (Lamy et al., 2014). The windblown 159 dust and its iron content effect on marine productivity in the Southern Ocean is thought to be 160 a key determinant of atmospheric  $CO_2$  concentrations [36]. During high dust level periods, 161 the global average temperature fell down to 10°C [35, 37, 38], which is 4.5°C lower than 162 current global average temperature. Loess sediments in the northern and southern 163 hemisphere on continents and ocean floors originate from these cold dusty periods.

Former geoscientists had the predominant conception that the cold glacial temperatures had caused dustiness, and not the reverse [39]. Meanwhile more evidence accumulates that mineral dust was a main factor in the cause of the cold periods and that the iron (Fe) fraction of wind-blown dust aerosol fertilized the oceans' phytoplankton, activating the assimilative conversion of  $CO_2$  into organic carbon [37-42] and carbonate which composes the main dry body substance of phytoplankton, together with silica, another component of dust [43].

Evidence about the responsibility of iron-containing dust that triggered ice ages during the 171 late Paleozoic epoch are in discussion [44].

The biogeochemical cycles of carbon, nitrogen, oxygen, phosphorus, sulfur and water are well described in the literature, but the biogeochemical cycle of the Earth's iron is often overlooked. An overview of the progress made in the understanding of the iron cycle in the ocean is given by several authors [45, 46].

The current state of knowledge of iron in the oceans is lower than that of carbon, although numerous scientific publications deal with this topic [47-55], meanwhile the iron biogeochemical cycle in the atmosphere is described by fewer ones [56-58], on the contrary to the iron biogeochemical cycle in soil and land, as almost no recent publications details the current knowledge of iron in soils and over the landscape [59-61], a task we attempt to do inthis review.

The process of iron fertilization by injection of iron salt solution into the ocean surface had already been in discussion as an engineering scheme proposed to mitigate global warming 183 [62]. But iron fertilization experiments with FeSO<sub>4</sub> conducted over 300 km<sup>2</sup> into the Sub-184 185 Antarctic Atlantic Ocean, although doubling primary productivity of Chlorophyll a, did not 186 enhance downdraft particles' flux into the deep ocean [63]. The researchers attribute the lack 187 of fertilization-induced export into the deep ocean to the limitation of silicon needed for 188 diatoms. Thus, ocean fertilization using only iron can increase the uptake of CO<sub>2</sub> across the 189 sea surface, but most of this uptake is transient and will probably not conduct to long-term 190 sequestration [64]. In other experiments, the authors [65] find that iron-fertilized diatom 191 blooms may sequester carbon for centuries in ocean bottom water, and for longer in the 192 sediments, as up to half the diatom bloom biomass sank below 1 km depth and reached the 193 sea floor. Meanwhile dissolution of olivine, a magnesium-iron-silicate containing silica, with a 194 Mg:Fe ratio of nearly 9:1, resulted in 35% marine carbon uptake (with the hypothesis of 1% 195 of the iron dissolved and biologically available), with communities of diatoms being one of the 196 phytoplankton winners [66].

The idea of climate cooling by  $CO_2$  carbon conversion into organic sediment carbon by 198 addition and mixture of an iron salt solution into the ocean with the marine screw propeller 199 has been the object of controversial debates [67-69]. The eolic iron input per square meter of 200 ocean surface by natural ISA is in the single decadal order of mg Fe m<sup>-2</sup> yr<sup>-1</sup>. In comparison, 201 the artificial Fe input by ship screws is orders of magnitude above the natural fertilizing with 202 ISA.

The small content of water-soluble iron salts (IS) in the dust particles triggers this fertilization 204 effect [70], and the soluble iron deposition during glaciations had been up to 10 times the 205 modern deposition [71]. According to Spolaor et al. [72], most of the bioavailable water 206 soluble Fe(II) has been linked, during the last 55,000 years, to the fine dust fraction, as it was 207 demonstrated from ice cores from Antarctica. During late Paleozoic epochs, glacial stage 208 dust fluxes of ~400 to 4,000 times those of interglacial times had been found [73], which 209 gives an estimated carbon fixation ~2-20 times that of modern carbon fixation due to dust 210 fertilization. Photochemistry by sunshine is the main trigger of the transformation of the 211 primary insoluble iron fraction of dust aerosols into soluble iron salts [74], and the 212 understanding of how the different iron content and speciation in aerosols affect the climate 213 is growing [75]. Currently, increased sub-glacial melt water and icebergs may supply large 214 amounts of bioavailable iron to the Southern Ocean [76]. The flux of bioavailable iron 215 associated with glacial runoff is estimated at 0.40-2.54 Tg yr<sup>-1</sup> in Greenland and 0.06-216 0.17 Tg yr<sup>-1</sup> in Antarctica [77], which are comparable with aeolian dust fluxes to the oceans

surrounding Antarctica and Greenland, and will increase by enhanced melting in a warmingclimate.

However, CO<sub>2</sub> uptake by the oceans is not the only effect of iron dust. The full carbon cycle 220 is well described in the literature; meanwhile we know less about the iron biogeochemical 221 cycle. Recently the major role of soluble iron emissions from combustion sources became 222 more evident. Today the anthropogenic combustion emissions play a significant role in the 223 atmospheric input of soluble iron to the ocean surface [78]. Combustion processes currently 224 contribute from 20 to 100% of the soluble iron deposition over many ocean regions [79]. 225 Model results suggest that human activities contribute to about half of the soluble Fe supply 226 to a significant portion of the oceans in the Northern Hemisphere [80], and that deposition of 227 soluble iron from combustion sources contributes for more than 40% of the total soluble iron 228 deposition over significant portions of the open ocean in the Southern Hemisphere [81]. 229 Anthropogenic aerosol associated with coal burning are maybe the major bioavailable iron 230 source in the surface water of the oceanic regions [82]. The higher than previously estimated 231 Fe emission from coal combustion implies a larger atmospheric anthropogenic input of 232 soluble Fe to the northern Atlantic and northern Pacific Oceans, which is expected to 233 enhance the biological carbon pump in those regions [83].

The limited knowledge about dissolved or even dispersed iron distributions in the ocean confirms the work of Tagliabue et al. [55]: their calculation results about the residence time of iron in the ocean differs up to three orders of magnitude between the different published models.

The precipitation of any iron salt results from the pH and  $O_2$  content of the ocean water milieu. But the presence of organic Fe chelators such as humic or fulvic acids [54] as well as complexing agents produced by microbes [49] and phytoplankton [84], life forms prevents iron from precipitation. In principle, this allow the transport of iron, from its sources, to any place within the ocean across huge distances with the ocean currents [25]. But organic material as well as humic acids have limited lifetime in oxic environments due to their depletion at last to  $CO_2$ . But within stratified anoxic ocean basins their lifetime is unlimited.

The iron inputs into the ocean regions occur by atmospheric dust, coastal and shallow
sediments, sea ice, icebergs and hydrothermal fluids and deep ocean sediments [47, 49, 56,
57, 83, 85-87].

Microbial life within the gradient of chemoclines dividing anoxic from oxic conditions generate organic carbon from  $CO_2$  or  $HCO_3^-$  carbon [88-90]. The activity at these chemoclines are sources of dissolved Fe(II). Humic acid is a main product of the food chain within any life habitat. Coastal, shelf, and ocean bottom sediments, as well as hydrothermal vents and methane seeps are such habitats and known as iron sources (Boyd and Ellwood, 2010). Insoluble Fe oxides are part of the lithogenic particles suspended at the surface of the 254 Southern Ocean. In addition to organic phytoplankton substance, the suspended inorganics 255 accompany the gut passage through the krill bodies. During gut passage of these animals, 256 iron is reduced and leaves the gut in dissolved state [91]. There is no doubt that gut-microbial 257 attack on ingested organics and inorganics produce faeces containing humic acids. This 258 metabolic humic acid production is known from earth worm faeces [92] and human faeces 259 [93, 94]. The effect of iron mobilization from lithogenic particles by reduction during gut 260 passage has been found in termites too [95]. The parallel generation of Fe-chelating humic 261 acids during gut passage guarantees, that the Fe is kept in solution after leaving the gut into 262 the ocean. The examples demonstrate that every link of the ocean food chain may act as 263 source of dissolved iron.

The co-generation of Fe(II) and Fe-chelating agents at any Fe sources at the bottom, surface and shelves of the oceans is the precondition to the iron transport between source and phytoplankton at the ocean surface. But the transport between sources and the phytoplankton depends on the vertical and horizontal movement activity in the ocean basins [48, 54]. Any movement between iron sources and the phytoplankton-rich surface in stratified ocean basins keeps restricted to surface near Fe input from its sources (shelf sediments, melt water, icebergs, rivers, surface water runoff and dust input).

During the glacial maxima the vertical movement activity arrived to an optimum. According to 272 that, the Fe transport from basin bottom sources and dust sources to the phytoplankton were 273 at their maximum and produce maximum primary productivity at the ocean surface but the 274 carbon burial became the lowest during that time [96] although the greenhouse gases 275 (GHGs) were at their lowest levels during the glacial maximum. Causal for this seemingly 276 contradiction are the changing burial ratios of organic C / carbonate C at the basin bottom(s). 277 The burial ratio is high during episodes with stratified water column and it is very low during 278 episodes with vertical mixed water column as we demonstrate in chapter 4 in detail.

This review aims to describe the multi-stage chemistry of the iron cycle on the atmosphere, 280 oceans, lands, sediments and ocean crust. This article is a comprehensive review of the 281 evidence for connections between the carbon cycle and the iron cycle, and their direct and 282 indirect planetary cooling effects. Numerous factors influence the Fe-cycle and the iron 283 dissolution: iron speciation, photochemistry, biochemistry, red-ox chemistry, mineralogy, 284 geology. In order to perform an accurate prediction of the impact of Fe-containing dusts, sea 285 salt, and acidic components, the atmospheric chemistry models need to incorporate all 286 relevant interaction compartments of the Fe-cycle with sun radiation, chlorine, sulfur, nitrogen 287 and water. This review advocates a balanced approach to benefit from the Fe-cycle to fight 288 global warming by enhancing natural processes of GHG depletion, albedo increase, carbon 289 burial increase and of de-stratification of the ocean basins.

#### **Breakdown of sections:**

The next three sections describe nearly a dozen different climate cooling processes induced 293 by iron salt aerosols (ISA) and their interaction for modeling parameter development 294 (sections 2, 3, 4 and 5). Then estimation of the requirements in terms of ISA, to stop global 295 warming will be given in section 6, followed by the description of a suggested ISA enhanced 296 method to fight global warming and induce planetary cooling in section 7, and the possible 297 risks of reducing acids and iron emissions in the future in section 8, followed by a general 298 discussion and concluding remarks in sections 9 and 10. To our knowledge, this review 299 completes, with atmospheric and terrestrial compartments [97], the previous ocean global 300 iron cycle vision of Parekh [98], Archer and Johnson [50], Boyd and Ellwood [49] and of 301 many others. It advocates a balanced approach to make use of the iron cycle to fight global 302 warming by enhancing natural processes.

## 304 Components of the different natural cooling mechanism by ISA

The best known cooling process induced by ISA is the phytoplankton fertilizing stage 306 described in the introduction. But this process is only part of a cascade of at least 12 climate 307 cooling stages presented in this review. These stages are embedded within the coexisting 308 multi-component complex networks of different reciprocal iron induced interactions across 309 the borders of atmosphere, surface ocean, sediment and igneous bedrock as well as across 310 the borders of chemistry, biology, and physics and across and along the borders of 311 illuminated, dark, gaseous, liguid, solid, semi-solid, animated, unanimated, dead and different 312 mix phase systems. Some impressions according to the complexity of iron acting in the 313 atmospheric environment have been presented by Al-Abadleh [75].

The ISA-induced cooling effect begins in the atmosphere. Each of the negative forcing stages unfolds a climate-cooling potential for itself. Process stages 1-6 occur in the troposphere (chapter 2), stage 6 at sunlit solid surfaces, stages 7-8 in the ocean (chapter 3), and stages 9-12 in the oceanic sediment and ocean crust (chapter 4). Other possible cooling stages over terrestrial landscapes and wetlands are described in chapter 5. The more than 12 stages of this cooling process cascade operate as described below.

# 322

#### 2.1. ISA-induced cloud albedo increase

2. Tropospheric natural cooling effects of the iron cycle

ISA consists of iron-containing particles or droplets with a chloride content. Aerosols have 324 significant effects on the climate [99]. First, by direct scattering of radiation, and second, by 325 inducing a cloud albedo increase. The latter effect is induced by cloud whitening and cloud 326 life time elongation. Both effects induce a climate cooling effect by negative radiative forcing 327 of more than -1 W per square meter.

Aerosols have a climate impact through aerosol-cloud interactions and aerosol-radiation 329 interactions [100]. By reflecting sunlight radiation back to space, some types of aerosols 330 increase the local albedo (which is the fraction of solar energy that is reflected back to 331 space), producing a cooling effect [101]. If the top of clouds reflect back a part of the incident 332 solar radiation received, the base of clouds receive the longwave radiation emitted from the 333 Earth surface and reemit downward a part of it. Usually, the higher a cloud is, in the 334 atmosphere, the greater its effect on enhancing atmospheric greenhouse warming, and 335 therefore the overall effect of high altitude clouds, such as cirrus, is a positive forcing. 336 Meanwhile, the net effect of low altitude clouds (stratocumulus) is to cool the surface, as they 337 are thicker and prevent more sunlight from reaching the surface. The overall effect of other 338 types of clouds such as cumulonimbus is neutral: neither cooling nor warming.

More outgoing long-wave radiation is possible when the cirrus cover is reduced. Efficient ice nuclei (such as bismuth tri-iodide) seeding of cirrus cloud might artificially reduce their cover [102, 103].

In order to enhance the cooling effects of low altitude clouds, marine cloud brightening has
been proposed [104], for instance by injecting sea salt aerosols over the oceans. The effect
depends on both particle size and injection amount, but a warming effect is possible [105].

- Aerosol effects on climate are complex because aerosols both reflect solar radiation to space 346 and absorb solar radiation. In addition, atmospheric aerosols alter cloud properties and cloud 347 cover depending on cloud type and geographical region [106]. The overall effect of aerosols 348 on solar radiation and clouds is negative (a cooling effect), which masks some of the GHGsinduced warming. But some individual feedbacks and forcing agents (black carbon, organic 349 350 carbon, and dust) have positive forcing effects (a warming effect). For instance, brown clouds 351 are formed over large Asian urban areas [107] and have a warming effect. The forcing and 352 feedback effects of aerosols have been clarified [101] by separating direct, indirect, semi-353 direct and surface albedo effects due to aerosols.
- Differing to any natural dust iron-containing mineral aerosol, the ISA aerosol does not contain 355 any residual mineral components such as  $Fe_2O_3$  minerals known as strong radiation 356 absorbers. Previous studies have shown that iron oxides are strong absorbers at visible 357 wavelengths and that they can play a critical role in climate perturbation caused by dust 358 aerosols [108, 109]. As the primary ochre colored aerosol particles emitted by the ISA 359 (method I, see chapter 7) have small diameters of  $< 0.05 \,\mu\text{m}$  and are made of pure FeOOH, 360 they become easily and rapidly dissolved within the plume of acidic flue gas. The ISA FeOOH aerosol is emitted with the parallel generated flue gas plumes containing SO<sub>2</sub> and 361 362 NO<sub>x</sub> as sulfuric and nitric acid generators. ISA stays up for weeks within the troposphere 363 before precipitating on the ocean or land surfaces. Due to their small diameter and high

surface area, the aerosol particles will immediately react with HCl, generated as reaction product between sea-salt aerosol and the flue gas borne acids. The reaction product is an orange colored FeCl<sub>3</sub> aerosol: ISA. During day time the sunlight radiation bleaches ISA into FeCl<sub>2</sub> and °Cl; at the night time the re-oxidation of ISA plus HCl absorption generates ISA again. The FeCl<sub>2</sub> aerosol particles are colorless at low humidity; pale green during high humidity episodes. The day time bleaching effect reduces the radiation absorption of ISA to much lower levels comparing to oxides such as Fe<sub>2</sub>O<sub>3</sub>.

Hygroscopic salt aerosols act as cloud condensation nuclei (CCN) [110, 111]. ISA particles
are hygroscopic. High CCN particle concentrations have at least three different cooling
effects [112, 113]. Each effect triggers the atmospheric cooling effect by a separate increase
of earth reflectance (albedo) [114]:

- cloud formation (even at low super saturation);
- formation of very small cloud droplets, with an elevated number of droplets per
   volume, which causes elevated cloud whiteness;
- extending the lifetime of clouds, as the small cloud droplets cannot coagulate with
   each other to induce precipitation fall.
- **Figure 1** illustrates this albedo change due to ISA-CCN particles.

Figure 1. Process of tropospheric cooling by direct and indirect increasing of the quantity of
 different cloud condensation nuclei (CCN) inducing albedo increase by cloud formation at low
 supersaturation, cloud whitening and cloud life elongation

Additional to climate cooling effects, CCN-active aerosols might induce a weakening of tropical cyclones. The cooling potential of the ocean surface in regions of hurricane genesis and early development, by cloud whitening potential [115] shall be casual. Further effects such as delayed development, weakened intensity, early dissipation, and increased precipitation have been found [116, 117].

#### 394 **2.2.** Oxidation of methane and further GHGs

- Currently, methane (CH<sub>4</sub>) in the troposphere is destroyed mainly by the hydroxyl radical °OH. 396 From 3 to 4% CH<sub>4</sub> (25 Tg yr<sup>-1</sup>) [118, 119] are oxidized by °Cl in the troposphere, and larger 397 regional effects are predicted: up to 5.4 to 11.6% CH<sub>4</sub> (up to 75 Tg yr<sup>-1</sup>) in the Cape Verde 398 region [120] and ~10 to >20% of total boundary layer CH<sub>4</sub> oxidation in some locations [121].
- According to Blasing [99, 122, 123] the increase of the GHG  $CH_4$  since 1750 induced a radiative forcing of about +0.5 Watts per square meter. The research results of Wittmer et al. [124-127] demonstrated the possibility to reduce the  $CH_4$  lifetime by the ISA method significantly. According to Anenberg et al. [128] the health effects of the combination of increased  $CH_4$  and  $NO_x$  induced  $O_3$  levels in combination with an increase of black carbon are responsible for tens of thousands deaths worldwide.

Any increase in the °Cl level will significant elevate the depletion rate of  $CH_4$  and further 406 volatile organic compounds (VOCs) as well as ozone (O<sub>3</sub>) and dark carbon aerosol as 407 described in sections 2.3 and 2.4.

- Absorption of photons by semi-conductor metal oxides can provide the energy to produce an 409 electron-hole pair able to produce either a reduced or an oxidized compound. At suitable 410 conditions, UV and visible light can reduce a variety of metal ions in different environments 411 [129-131]. Photo-reduced metal compounds may further act as effective chemical reductants 412 [132, 133] and the oxidized compounds such as hydroxyl radicals or chlorine atoms, can 413 further act as effective oxidants. Zamaraev et al. [134] proposed the decomposition of 414 reducing atmospheric components such as CH<sub>4</sub> by photolytically induced oxidation power of 415 the oxides of iron, titanium and some further metal oxide containing mineral dust 416 components. In this sense Zamaraev designated the dust generating deserts of the globe as 417 "kidneys of the earth" [135] and the atmosphere as a "giant photocatalytic reactor" where 418 numerous physicochemical and photochemical processes occur [134]. Researches have 419 proposed giant photocatalytic reactors to clean the atmosphere of several GHGs, such as 420 N<sub>2</sub>O [136], CFCs and HCFCs [137] and even CO<sub>2</sub> after direct air capture [138], as almost all 421 GHGs can be transformed or destroyed by photocatalysis [139, 140].
- Oeste suggested [141] and Wittmer et al. confirmed [124-127] the emission of CH<sub>4</sub> depleting 423 chlorine atoms. This can be induced by 3 ways: sunlight photo reduction of Fe(III) to Fe(II) 424 from FeCl<sub>3</sub> or FeOOH containing salt pans, from FeCl<sub>3</sub> or FeOOH-containing sea spray 425 aerosols and from pure FeOOH aerosol in contact with air containing ppbv amounts of HCI. 426 Because the H abstraction from the GHG CH<sub>4</sub> as the first oxidation step by °Cl is at least 427 16 times faster compared to the oxidation by °OH, which is the main CH<sub>4</sub> oxidant acting in 428 the ISA-free atmosphere, concentration of CH<sub>4</sub> can be significantly reduced by ISA emission. 429 Figure 2 illustrates by a simplified chemical reaction scheme this climate cooling mechanism 430 by the ISA method: a direct cooling of the troposphere by CH<sub>4</sub> oxidation induced by ISA

particles.