# Peer review of "Climate engineering by mimicking the natural dust climate control"

_Earth System Dynamics, 2016_

## Referee Comment (RC1) · S. M. Elliott (Referee) · 25 Sep 2016

Author: Scott ELLIOTT, Climate Ocean Sea Ice Modeling (COSIM), Los Alamos National Laboratory, Los Alamos NM, 87545 USA, Email: sme@lanl.gov

The contribution under discussion from Oeste and colleagues reviews prospects for allowing, or even controlling, atmospheric chemical processes in order to enhance iron fertilization of the remote ocean. Emphasis is placed upon a routing of specific additives, through the exhaust or effluent from existing energy infrastructure and transportation modes. But the arguments extend high into the atmosphere, deep within the ocean and even penetrate the lithosphere.

[Figure]

The paper constitutes a veritable treasure trove of information. A huge variety of lower atmospheric and upper marine chemistry is treated in great detail, with complete contemporary referencing. Special attention is given to iron processing, as it may interact with the remainder of the periodic table across multiple geophysical fluid environments. General aerosol/cloud processing, gas phase photochemistry, surface ocean organosulfur controls and marine biogenic precursor emissions all enter the logic, along with much more. As a bonus, the text delves into long time-scale crustal geochemical cycling. Special attention is given to cross-talk among environmental phase states involving iron oxy-hydroxides, soot and finally aqueous-gas halogen chemistry. Continental shelf clathrates and their destabilization also appear in due course.

Clearly this work is exceptionally broad in its chemical (engineering) scope. It raises fascinating issues regarding the complexities which must be confronted as humankind considers manipulating global geochemical cycles which may be available and accessible. Inevitably, we as a species are likely to test many such concepts and implement some subset of them. Oeste et al. offer the most complete discussion of such topics that I have ever encountered. And I make this statement after a full career involving a substantial amount of geoengineering research. Breadth is in fact the primary virtue of the paper, and not one to be dismissed lightly.

Due in large part to its (breathtaking) scope, the Oeste paper is exceptionally difficult to ingest. Problems will be compounded for certain readers by a consistent use of highly nonstandard scientific English, with a strongly European tone. But the writing style is only a very minor problem. I had no trouble translating into more traditional and expected wording-sentence structure internally, as I was moving through the various sections. In fact I found myself COMPELLED to do so, as I was drawn by my own chemical curiosity into the guts of the material. My reward: An abundance of new and exciting concepts, all exceptionally well referenced.

About halfway through the paper, I was somewhat surprised to find that I had hit my own personal scientific limits as a reviewer. I like to boast of strong interdisciplinary

training, and consider myself relatively well-versed in most aspects of global scale environmental chemistry. During my career there has been serious involvement in the coupled and highly interactive processes of upper ocean and lower atmosphere, with further experience on either side of this regime –spanning downward into waters of the thermocline and Arctic Ocean shelf-Atlantic Layer plus up through the stratosphere. Still, the research described by Oeste and company pushed well beyond my expertise into several new and remote corners of the Earth System. The work is encyclopedic.

In fact it is truly outstanding in terms of its global physical-chemical scope. I found myself smiling and shaking my head in amazement as new themes kept entering. . . and entering and entering.

I was not able to identify ANY scientific errors anywhere in the paper. This is usually a very good sign. I noted only well-formulated, testable ideas as I moved through the text. My feeling is that the span of the authors' thinking will make it difficult for your journal to secure a single, all-encompassing review. Hence my recommendation is the following: Let us err on the side of free and open dissemination of scientific information. Publish the paper exactly as it stands. The work should prove to be a valuable source of information for not only the geoengineering community but for Earth System scientists and modelers in a general sense. My judgment is this –get the paper out there since it will be extremely useful to diverse research communities.

I have only one real criticism of Oeste et al. and though major, it is also very much worth exploring even as I suggest that publication is your best strategy. The concern is so critical that I will take an opportunity to devote several paragraphs to it here. Furthermore the points are mainly philosophical, and they are applicable to many generic geoengineering concepts now reverberating through the contemporary market place. Oeste and company offer a perfect springboard for the debate I would hope to engage.

It is of course fascinating to consider whether jet fuel combustion products, power plant plumes or a population of ship tracks might be augmented-spiked-manipulatedengineered to add to the tendency of iron cycling to counter the greenhouse effect. But the notion really boils down to combatting one form of large scale pollution with another. Is this perhaps where humanity should draw a geoengineering "line in the sand"?

There are many analogous cases citable from the literature, including papers on which I have been an author. At one point in the early nineties, I participated in evaluation of a scheme to mitigate the ozone hole by engineering the injection of light hydrocarbons -into the total volume of the stratosphere over Antarctica. The immediate inspiration was to convert chlorine radicals (the bare, reactive atoms) into the stored and thus safer form hydrochloric acid (vapor). But simple photochemical modeling performed relatively quickly by our group demonstrated that all was not as it seemed. First impressions had been very deceptive for us. We quickly learned that our elegant little idea would backfire badly. A combination of gas phase and heterogeneous (cloud-ice surface) chemical processes would in fact lead to EXTRA activation of chlorine into the atomic form –a shift TOWARD increased ozone depletion power.

This theme will likely be repeated many times during the era of global change, as increasingly well-educated and well-armed systems technologists explore a myriad of potential and imaginable mitigation concepts. But even a moderate amount of knowledge can be a dangerous thing. We must naturally expect the unexpected, from a singularly complex system which also happens to be our home. It is clear in the foreseeable future that our ability to conceive new and exciting geochemical cycles far outstrips our ability to understand them, whether in the mind's eye or computationally.

Earth System Models and the ever-so-human teams that build them are currently saturated, even by exercises as simple as inter-comparison of CO2 drawdown toward the deep sea or into soils. Ecodynamics lead in a well established manner to carbon storage disconnected from the atmosphere, and global warming may indeed alter the biological capability to provide such reservoirs. But carbon dioxide is only one especially long-lived greenhouse gas among many, whether anthropo- or bio-genic. And feedback loops abound which are highly nonstandard, extending far beyond atmospheric accumulation of infrared absorbers to. . .

–Marine and terrestrial biological aerosol sourcing, their relation to cloud condensation nucleus numbers, the attendant influence on short and long wave cloud forcing, ice nucleation with exquisite sensitivity in polar cloud reflection, biogeochemical influences within the sea ice system including pigment layers and exopolymer pore blockage, continental shelf response to riverine chemical change, coastal plumes and aerosol release. . . the list is longer in fact than the Oeste et al. paper itself.

The entire periodic table, Beilstein database and the Linnaean biological hierarchy should be enfolded into this book of unknowns. Multiple oxidation states and phases are intimately involved in making the requisite environmental chemical connections. Bacteria dominate the organic chemistry of the entire exterior of our planet, yet their processing of macromolecules is only crudely represented in current systems models. Surfactant chemistry and especially that of the biopolymers is especially understudied, while interfaces act as a set of critical filters at all major points of communication among bulk phases. In some cases, new geochemical cycles which must be considered as potential feedbacks lead to strong regional effects and the redirection of classic meteorological teleconnections –while cancelling themselves or each other to a large degree at the planetary scale!

These effects are invisible in all modern modeling approaches, reduced or detailed. The influences pass far beyond mere global warming. Critical human dimension issues of the global hydrological system are at stake, including precipitation rates and distributions, drought, flood, landslides, the recharging of aquifers and all manner of severe storms.

I will not even touch upon the implications for biodiversity and ecosystem (food web) structure.

Since the present-day Earth System Modeling culture is so resource limited that it can

barely push itself beyond traditional CO2, complex chemical and biological engineering concepts pose certain inevitable and inalienable risks. But nonetheless, these should (must) be vigorously and repeatedly revisited. Identifying the challenges and attendant problems forces us all to think in new and creative ways about the vast network of bio-geochemical controls now being uncovered, across the planetary climate state. Newer and more exotic feedback loops will continue coming to the fore –but history teaches us that this is entirely to be expected. The ozone war had its hole and then a wholly unfa-miliar heterogeneous chemistry, driven by multi-element crystalline phases. Southern Ocean iron may prove to be nitrate-appropriating at a vast and tele-connective scale. Biogenic aerosol particles pass information to marine cloud systems in ways which are potentially Gaian and self reinforcing for primitive organisms. Or there may be positive gain on a net basis. Arctic sea ice loss may yet prove to be accelerated by biological influences on coloration, absorption, organic restructuring and pack-internal radiation transfer.

Under these circumstances, my assessment is that the rich exposition of environmental iron chemistry offered to us by Oeste et al. should simply be published as is. It is dense and difficult to read but describes an even more difficult planetary milieu. The paper constitutes a literary metaphor for the only known biosphere. And this gives one appropriate pause. As you can see, the authors already have me thinking in ways I never have before -and I have been working on this sort of material continually over a very long career. But at the same time, I wish to add a note of caution. Earth System Models of the contemporary era are woefully inadequate to the task of truly evaluating such idea suites. And so of course my reader must now be smiling and remarking -like everyone else this guy Elliott has an agenda.

As a community, why don't we bounce off this sort of effort to improve the process treatments in our environmental system codes? This would occur precisely so that we can think more deeply about the biogeochemical details. The devil is in them, I have no doubt. Both our comprehension and management of the planet will benefit from

contemplation of Oeste et al. and the like. A cascade of investments in laboratory and field work will also be required, but this is a very natural outcome... and is anyone out there complaining?

In the interest of generating a timely response for this journal, I will submit my review at the present stage. But I hope to return soon with further comments, cast as a set of tables quantifying the main concepts invoked. I believe it can be shown rigorously that Earth System simulations of the present day can only be improved through disciplined, prioritized introduction of...

-More complete sets of the chemical elements, additional interfaces, minor phases and their transitions, biopolymers, the organisms which synthesize them, surfactants, modes of catalysis and much more. As indeed the authors in question here strongly imply. It's about the chemistry. I further believe that geoengineering propositions could be usefully categorized by their geocycling complexity, in terms much like the ones I have outlined. Plus a metric of "containment" could be incorporated... which would encapsulate the notion of pollutant versus pollutant.

I'll try hard to return to these issues in the near future, by submitting further and more persuasively annotated commentary. But for the moment, I recommend that this very speculative and complete review from the Oeste team be added to the ever-expanding geoengineering literary fray. I have with me a thorough listing of edits to the European English along with a few typo identifications, but I will only send such trivia if requested to do so.

---

## Referee Comment (RC2) · Anonymous Referee #2 · 12 Oct 2016

This is an interesting paper with some interesting ideas, but which fails to provide sufficient evidence that this is a good approach and has many errors.

The english is not adequate in the paper. The organization is poor (most of the conclusions talk about the cost-effectiveness of this solution, instead of pulling ideas together from the paper).

The paper is full of details, but it gives the impression that the authors don't really understand what they are saying, as there are not always the right papers cited, and the latest ideas included in the paper, which could be a consequence of the great breadth discussed in the paper. The paper does not convince that the feedbacks described will be large, just that they might exist. There are many other impacts NOT discussed that

could offset the impacts suggested in the paper.

I am only able to review limited portions of the paper: particularly the atmospheric aerosol and impact on land and ocean biogeochemistry, and I found these parts of the paper to be incomplete to the point of wrong.

In the introduction, the paper does a poor job describing the state of our knowledge of iron in the oceans. While I agree that we know less about iron than carbon this sentence is a problem: "meanwhile the iron biogeochemical cycle is only described in the ocean by few scientific publications (Boyd and Ellwood, 2010;Mahowald et al., 2005; Mahowald et al., 2009).:" Please correct these citations: Mahowald et la., 2005 and 2009 focus on the atmospheric iron cycle. For example, work by Ken Johnson, Moore et al., 2013 or Sigman et al., 2010 on the importance of iron in the oceans on ocean biogeochemistry in different time periods would be appropriate papers to cite here.

People have rejected the idea of iron fertilization of the oceans for many reasons, and this is not well described in the paper. Are you arguing we should go back and debate this? You are not really discussing the state of knowledge of this debate, or countering it, but rather just ignoring the debate here?

The authors do not seem to realize that if you add iron to aerosols, they will tend to absorb more incoming radiation, and thus warm the planet: so this is the opposite effect you want. Check Sokolik and Toon, 1999.

None of the section 2.1-2.4 convince me that these effects will be significant. I lost a bit of ability to understand after that, but it seems like many of these feedbacks are actually very long term, and not very helpful in the next 30-300 years (ie. Section 4.3: minimizing ch4 emissions from sediment and bedrock: is ch4 release from bedrock really a problem we have right now, or on geologic time scales?) The whole section 4.4 seems totally off base: it is not thought that the iron inputs from Amazon are important but rather the P inputs, and they only operate on geologic time scales.
I would recommend to the authors to focus one paper on each section, not just on describing possible mechanisms, but rather on calculating the impacts of ISA for each mechanism, and making sure the impacts are significant. Do a good job, get each idea published, and then you can pull them together later.

---

## Short Comment (SC1) · 17 Oct 2016

Regarding the phase partitioning described in the manuscript, I would like to make two remarks:

- On page 13, you write: "Iron is completely part of the liquid or solid phase, so the Henry's law constant is estimated to more than $10^6$ mol m$^{-3}$ Pa$^{-1}$ (Sander, 2015)", citing my compilation of Henry's law constants. However, iron is not included in my publication. I am now curious about the origin of the value $10^6$ mol m$^{-3}$ Pa$^{-1}$.

- Transfer between the gas phase and the aqueous phase is not determined by

short-lived radicals like ·Cl and ·OH, as you indicate in Fig. 3. Instead, longer-lived molecules like $H_2O_2$, $Cl_2$, and HOCl determine the distribution between the phases. Thus, it is not sufficient to consider only the Henry's law constants of OH and Cl.

---

## Short Comment (SC2) · 19 Oct 2016

In section 5, this manuscript discusses our estimate for the amount of dissolved Fe(III) in sea-salt aerosol required for a significant global increase of atomic Cl. Our experiments obtained the first quantitative determinations of the source strength of atomic Cl from iron doped salt pans (Wittmer et al., 2015a) and aerosol droplets (Wittmer et al., 2015b) as an experimental basis of the ISA method, and we wanted to give a conservative estimate on the basis of our salt-aerosol experiments and not to be involved into extrapolations to the real atmosphere (which would require detailed global model calculations).

In contrast to a salt pan, the aerosol technique has the advantage that we can calculate

the surface area of the aerosol from the measured size distribution and number density. This calculation required a minor correction for deposited aerosol on the chamber wall, where it was found to be less efficient in the production of atomic Cl. The correction procedure was described in Wittmer et al., 2015b, as follows:

"To quantify the particle deposition and its contribution to the active surface area (and thus to Cl activation), a test measurement was performed to determine the fraction of Cl release by the active wall surface compared with the active aerosol surface: the iron-doped artificial seawater sample (Cl /Fe(III) = 13) was injected and allowed to deposit totally for 17 h (<0.005 % of the surface area should have remained suspended) while keeping the RH at 80 %. Then the 'aerosol-free' chamber was irradiated, resulting in a Cl production that was 20 $\pm$ 4 % compared with the actual production measured for the same sample in an aerosol experiment (see section 3.3.2: Iron(III)-catalysed Cl atom production). This production was evaluated by taking the mean of the quotient of each total production (deposited and not deposited) normalised by the respective initial LWC directly after injection. In Eqn 3.3, the contribution of deposited, active aerosol surface area is accounted for by adding 20 % of the deposited surface area since the time of injection to the surface area when the lights were turned on (corrected for deposition)." Contour plots of the time-dependent size distributions are displayed in the supplementary material of our work.

The simulated salt pan has the disadvantage that the size and the surface of the crystalline grains are hard to characterize and that the depth of the quasi-liquid layer involved in the exchange with the gas phase is not well-defined because of its dependence on relative humidity. Control of relative humidity would require temperature control of the salt layer that is irradiated by a strong, bifocal solar simulator. The aerosol chamber has the advantage that the size distribution of the droplets can be characterized and that the required close contact between gas and liquid is established in a well-mixed and thus fairly homogeneously irradiated volume of air. The Köhler equation then describes the hygroscopic, deliquescent behavior of the FeCl3 doped salt

droplets.

We estimated the efficiency for Cl generation (p. 39, line 1226) from observed concentrations of atomic Cl (atoms per cm3) in simulation chamber experiments and calculated the corresponding source strengths (atoms per cm3 per hour) from the quasistationary state, established between the photochemical production of Cl and its consumption by reaction with the test mixture of hydrocarbons present in the gas phase (in the absence of any other reactants for the atomic Cl – the zero air contained less than 1 ppb of O3, less than 0.5 ppb of NOx and about 100 ppb of CH4). The observed concentrations are high because the aerosol surface in our chamber (10000-30000 $\mu$m2 per cm3) is about 150 times higher than in the marine boundary layer, where it is $\sim$60-200 $\mu$m2 per cm3 according to Warneck (1999). One may calculate the source strength per square cm of aerosol surface and try to base the calculation on this information, assuming a linear relationship. The inherent loss of aerosol surface by deposition was monitored in our experiments and considered in the calculation of the source strength that was corrected for a minor catalytic influence of the deposits on the chamber walls.

In our conservative estimate, we simply refer to the amount of dissolved Fe(III) in sea-salt aerosol corresponding to the performed experiments. According to Keene et al. (1990), 1785 Tg Cl- dissolved in sea salt are produced annually. To obtain a molar Cl-/Fe(III) ratio of 51, a mass of 56 Tg dissolved Fe(III) would be necessary according to the molar masses. Comparing the blank experiment (artificial sea salt without Fe(III), producing 0.3-0.4 $\times$10ˆ21 atoms cm-2 h-1) to the Cl-/Fe(III) = 51 experiment (producing 1.4-1.5 $\times$10ˆ21 atoms cm-2 h-1 with a fairly constant rate), one can estimate a fourfold increased production of Cl radicals in the gas phase. Based on the 150-167 times higher available aerosol surface area in our chamber, the fourfold increase was scaled down to 0.024-0.0267 (4/150 and 4/167) which corresponds to $\approx$2.5 %. The same, simple calculation was performed to estimate the 17-19 % increase when aiming for Cl-/Fe(III) = 13.

Going along the criticisms raised by Oeste et al., we justify our assumptions with the

following comments:

1. Instead of using a Cl-/Fe(III) molar ratio of 101, Wittmer used a Cl-/Fe(III) ratio of 51 to perform calculations at the suboptimal pH of 3.3-3.5: 1.9 x 10ˆ5 Cl/cm3 (Wittmer et al., 2015a);

Comment: We used a ratio of 51 since we hardly saw a significant effect with a ratio of 101 (see figure 3.4 in Wittmer et al., 20156). However, the low pH experiment (pH 2.1-2.3) is an error-prone basis for an estimate, since the pH is far away from the pH of the blank experiment (4.7-5.0). A blank experiment at such a low pH would be more suitable for a better estimation where also a higher Cl production can be expected. We admit that in our work we had a pH of 3.3-3.6 (see Table 3.2 in Wittmer et al, 2015b), and thus the fourfold increase is probably overestimated since a higher Cl production in the blank – due to a lower pH – would lead to a lower relative increase when adding Fe(III).

2. There is no other tropospheric Cl- source than sea-salt; Comment: This claim is correct for the conditions of our chamber, where other sources are considered by the blank experiment. We just assumed sea-salt as a Cl source since we considered this source in our experiments alone (induced by already dissolved Fe(III) in a sea-salt matrix).

3. The global production rate of sea-salt aerosol Cl- of 1785 Tg/year has to be doped with iron at a Cl-/Fe(III) molar ratio of 51. Comment: See comment on criticism 1.

4. ISA has the same particle size and corresponding surface range as sea-salt; 5. ISA has the same residence time as sea-salt aerosol in the troposphere

Comment to 4. and 5.: We do not claim that our estimation counts for the ISA particles. As described in our text, the estimation is focused on the Cl production by sea-salt aerosol (ignoring other sources). We only refer to the required amount of dissolved Fe(III) in the sea-salt aerosol compared to our experiments. Pyrogenic oxide particles

smaller than 0.1 $\mu$m, as proposed by the authors, are not covered by our estimation. The pH of the aerosol droplets (p. 39, lines 1233 and 1234) was not measured, but the pH of the stock solution (1 g NaCl per liter, doped with FeCl3), that is atomized to obtain the optimum size of the aerosol droplets. The pH of the droplets was not measured in our work. The low pH is a consequence of hydrolysis of FeCl3, and the gas phase is expected to contain the HCl gas volatilized from the atomized droplets of the stock solution. The lifetime reduction of CH4 in the troposphere (p. 36, line 1236) does not correspond to the lifetime reduction in the aerosol chamber but would depend on the vertical distribution of the ISA in the troposphere in comparison with the vertical distribution of CH4. We did not claim that the optimum efficiency of Cl production occurs at pH 2 and independent of the distribution of HCl between gas and particle: the lower the pH the less chloride will be available in the particle phase. For HCl adsorbed on humidified Fe2O3 (Wittmer and Zetzsch, 2016), the optimum pH will be much lower than 2 since this may dissolve some of the iron(III).

Finally, we would not like to further discuss our simplifying assumptions 2. to 5. (p. 40, line 1246-1275). These were meant to be restricted to Fe doped NaCl aerosol alone, implying an uncertainty of more than an order of magnitude of our estimate.

We agree with the argument 6. (line 1276-1282) that the larger specific surface of combustion-derived Fe2O3 is promising. This is the reason why we had started to look into the interaction of a pyrogenic Fe2O3 material (characterized by Mössbauer spectroscopy) with HCl gas (p 43, line 1383). The technical application of ISA for climate engineering (how to obtain and apply a finer size distribution of FeCl3, FeOOH or Fe2O3 or coatings of sublimed FeCl3 on solid carrier particles or to add iron-containing compounds to the fuel of combustion engines and power plants) is beyond the scope of our studies.

Bibliography:

Keene, W. C., Khalil, M., Aslam K., Erickson, D. J., McCulloch, A., Graedel, T. E.,

Lobert, J.M., Aucott, M. L., Gong, S. L., Harper, D. B., Kleiman, G., Midgley, P., Moore, R. M., Seuzaret, C., Sturges, W. T., Benkovitz, C. M., Koropalov, V., Barrie, L. A., Li, Y. F., 1999. Composite global emissions of reactive chlorine from anthropogenic and natural sources: Reactive chlorine emissions inventory. J. Geophys. Res. 104, 8429-8440.

Warneck, P., 1999. Chemistry of the Natural Atmosphere, Academic Press, San Diego.

Wittmer, J., Bleicher, S., Ofner, J., Zetzsch, C., 2015a. Iron (III)-induced activation of chloride from artificial sea-salt aerosol. Environmental Chemistry 12, 461-475.

Wittmer, J., Bleicher, S., Zetzsch, C., 2015b. Iron (III)-induced activation of chloride and bromide from modeled salt pans. J. Phys. Chem. A 119, 4373–4385.

Wittmer, J., Zetzsch, C., 2016. Photochemical activation of chlorine by iron-oxide aerosol. J. Atmos. Chem. DOI 10.1007/s10874-016-9336-6

---

## Author Comment (AC1) · 9 Nov 2016

The authors would like to thank Professor Scott M. Elliott for the careful reading of our manuscript and for his very constructive comments that helped us to improve our manuscript.

Our responses to the Reviewer 1 comments are included in the followings.

Comment 1: Starting end of page C3 till page C7, the referee formulates a basic critic point directed to most of the climate engineering methods. Even the methods aimed to remove greenhouse gases like CO2 from the atmosphere (CDR) including the ISA method proposed in this manuscript: "But the notion really boils down to combatting

one form of the large scale pollution with another."

Answer 1: Anderson (2016) reminded that of the 400 IPCC scenarios that keep warming below the Paris agreement target, "344 involve the deployment of negative emissions technologies", which he qualifies of "speculative" or requiring geoengineering.

A large part of the research devoted to climate engineering methods concerns SRM (sunlight reduction methods), like mimicking the effects of large volcanic emissions by adding sulphates aerosols into the stratosphere as suggested for instance by Crutzen (2006). Numerous other types of particles have been suggested for these aerosols for instance titania by Jones (2015). But SRM only buys time and has numerous drawbacks. On the one hand, SRM did not address the main cause of global warming (GHG emissions), nor prevents ocean acidification. On the other hand, several CDR technologies do, but their costs are much larger than SRM and the scale requested poses many technological challenges, for instance "scaling up carbon dioxide capture and storage from megatons to gigatons" (Herzog, 2011).

Very few CDR methods without emission of disadvantageous pollution are known. One of those is the Terra Preta method: it is characterized by the mixing of grinded bio-char into agricultural soils. The climate relevancies of this method are sustained fixation of former $CO_2$ carbon, minimizing fertilizer consumption and $N_2O$ emission reduction from the fertilized Terra Preta soils. Char has similar properties within the soil environment like humic substance, but in the environment char is resistant against oxidation. Comparing the Terra Preta method to other CDR methods like fertilizing the ocean by micro nutrients, results in lower specific material expenses by CDR methods per unit of $CO_2$ removed from the atmosphere (Betz et al. 2011). The ISA method we propose is a member of this CDR group, thus this result is also valid. In addition the further climate effects of the ISA method (like depletion of $CH_4$, tropospheric ozone, and soot, plus cloud whitening) reduce the specific material expense level. Also the ISA method mimics a natural phenomenon (mineral iron-dust transport and deposition) and only proposes to improve the efficiency of an already existing anthropogenic pollution.
Myriokefalitakis et al (2015) estimates that "The present level of atmospheric deposition of dissolved Fe over the global ocean is calculated to be about 3 times higher than for 1850 emissions, and about a 30% decrease is projected for 2100 emissions. These changes are expected to impact most on the high-nutrient–low-chlorophyll oceanic regions."Their model "results show a 5-fold decrease in Fe emissions from anthropogenic combustion sources in the year 2100 against in the present day, and about 45% reduction in mineral-Fe dissolution compared to the present day". Recently Boyd and Bressac (2016) suggested rapidly starting tests to determine efficiency and side effects of CDR ocean iron fertilizing methods.

Several experts, for instance Hansen et al. (2016), expressed recently the urgent warning that mankind has only short time left to address and control climate warming. As a consequence mankind ought to find out as soon as possible climate controlling matter which might generate the most effective and reversible climate cooling effects within the shortest period. Lifetime of ISA emissions in the troposphere are much shorter that of sulphates in the stratosphere. Of course, such tools and agents have to be rapidly evaluated against side-effects to ecosystems, human health, and last but not least their economic burdens.

Comment 2: (page C2) Due in large part to its (breathtaking) scope, the Oeste paper is exceptionally difficult to ingest. Problems will be compounded for certain readers by a consistent use of highly nonstandard scientific English, with a strongly European tone. (page C7) I have with me a thorough listing of edits to the European English along with a few typo identifications, but I will only send such trivia if requested to do so.

Answer 2: The authors would greatly appreciate if the Reviewer can send them his English, grammatical and other edits suggestions to improve their manuscript digestibility and readability.

Bibliography:

Anderson, K. (2015). Duality in climate science. Nature Geoscience, 8(12), 898-900.

Betz, G., Brachatzeck, N., Cacean, S., Güssow, K., Heintzenberg, J., Hiller, S., Hoose, C., Klepper, G., Leisner, T., Oschlies, A., Platt, U., Proelß, A., Renn, O., Rickels, W., Schäfer, S., Zürn, M.: Gezielte Eingriffe in das Klima? Eine Bestandsaufnahme der Debatte zu Climate Engineering. Sondierungsstudie für das Bundesministerium für Bildung und Forschung, ISBN: 3-89456-324-9, 2011, Kiel Earth Institute.

Boyd, P. W., & Bressac, M. (2016). Developing a test-bed for robust research governance of geoengineering: the contribution of ocean iron biogeochemistry. Phil. Trans. R. Soc. A, 374(2081), 20150299.

Crutzen, P. J. (2006). Albedo enhancement by stratospheric sulfur injections: a contribution to resolve a policy dilemma? Climatic change, 77(3), 211-220.

Hansen, J., Sato, M., Hearty, P., Ruedy, R., Kelley, M., Masson-Delmotte, V., Russell, G., Tselioudis, G., Cao, J., Rignot, E., Velicogna, I., Tormey, B., Donovan, B., Kandiano, E., von Schuckmann, K., Kharecha, P., Legrande, A.N., Bauer, M., Lo, K-L.: Ice melt, sea level rise and superstorms: evidence from paleoclimate data, climate modeling, and modern observations that 2 °C global warming could be dangerous. Atmos.Chem. Phys., 16, pp. 3761-3812, 2016

Herzog, H. J. (2011). Scaling up carbon dioxide capture and storage: From megatons to gigatons. Energy Economics, 33(4), 597-604.

Jones, A. C., Haywood, J. M., & Jones, A. (2015). Climatic impacts of stratospheric geoengineering with sulfate, black carbon and titania injection.

Myriokefalitakis, S., Daskalakis, N., Mihalopoulos, N., Baker, A. R., Nenes, A., & Kanakidou, M. (2015). Changes in dissolved iron deposition to the oceans driven by human activity: a 3-D global modelling study. Biogeosciences, 12(13), 3973-3992.

---

## Author Comment (AC2) · 9 Nov 2016

The authors would like to thank Prof. Rolf Sander for the careful reading of our manuscript and his valuable comments that will help us to clarify some points of our manuscript and correct a mistake.

Comment 1: On page 13, you write: "Iron is completely part of the liquid or solid phase, so the Henry's law constant is estimated to more than 106 mol m-3 Pa-1 (Sander, 2015)", citing my compilation of Henry's law constants. However, iron is not included in my publication. I am now curious about the origin of the value 106 mol m-3 Pa-1.

Answer 1: It was our mistake to insert the iron compounds into Table 1 (line 381).

Sorry for that. At tropospheric conditions and in its different possible states in ISA the iron content stays completely within the condensed phase (either as liquid or as solid). Thus any discussion about its Henry constant is obsolete: at tropospheric conditions the Henry constant of iron is of infinite height. Table 1 is changed accordingly (no more mention of iron compounds).

Comment 2: Transfer between the gas phase and the aqueous phase is not determined by short-lived radicals like $°Cl$ and $°OH$, as you indicate in Fig. 3. Instead, longer lived molecules like $H_2O_2$, $Cl_2$, and $HOCl$ determine the distribution between the phases. Thus, it is not sufficient to consider only the Henry's law constants of OH and Cl.

Answer 2: We agree that the short living radicals $°Cl$ and $°OH$ recombine very fast to $Cl_2$, $H_2O_2$, and $HOCl$. Iron exists at least in part as Fe(III) during nighttime and at least in part as Fe(II) during daytime. The methane oxidation by $°Cl$ and $°OH$ is restricted to the daytime because during night hours $°Cl$ and $°OH$ recombine fast to $Cl_2$, $HOCl$, and $H_2O_2$ in the dark (von Glasow, 2000). During daylight hours these recombination products photolyze again by regeneration of the radicals. But even during day time these radicals and their recombination products co-exist because the cycling between $°Cl$, $°OH$, $Cl_2$, $HOCl$, and $H_2O_2$. This cycling is activated by sunlight photolysis and radical recombination reactions (von Glasow, 2000, Luna et al. 2006). As we learn from Henry's law constants in Table 1 the oxygen species $°OH$ and $H_2O_2$ have a much higher tendency to stay in the liquid phase than the chlorine species $°Cl$ and $Cl_2$. $Cl_2$ has the tendency to react with water of neutral pH by producing HOCl. But the pH values of ISA, especially if ISA is emitted as acid flue gas plumes are lower than 3. Within this acidic region the tendency of HOCl generation from $Cl_2$ decreases to very low values and even at those humidity levels when the ISA particles become deliquescent the majority of the activated chlorine species will be localized in the $CH_4$ containing gaseous phase, not in the liquid phase. But even $°OH$ may leave the condensed phase into the gaseous phase at favorable circumstances into the gaseous phase (Nie et al., 2014) and may contribute there to the oxidation of $CH_4$ during clear dry conditions

without liquid phase at the Fe(III) surfaces.

These informations are inserted on a new column of Table 1 as follows:

Table 1: the Henry's law constants (Sander 2015) and daylight stability for different gaseous or vaporous components reacting with or produced by ISA in the troposphere

Substance Henry's law constant [mol m-3 Pa-1] Stability against tropospheric day light (+ stable; - unstable) CH4 1.4 x 10-5 + °Cl 2.3 x 10-2 + Cl2 9.2 x 10-4 - HCl 1.5 x 101 + HOCl 6.5 - °OH 3.8 x 10-1 + H2O2 8.3 x 102 -

Bibliography:

Luna, A. J., Nascimento, C. A. O., and Chiavone-Filho, O. (2006). Photodecomposition of hydrogen peroxide in highly saline aqueous medium. Brazilian Journal of Chemical Engineering, 23(3), 341-349.

von Glasow, R.: Modeling the gas and aqueous phase chemistry of the marine boundary layer. Ph.D. thesis, Univ. Mainz, Mainz, Germany, 2000 (Available at www.rolandvonglasow.de).

––––––––––––––––––––––––––––––

---

## Author Comment (AC4) · 15 Nov 2016

We would like thank Prof. Cornelius Zetzsch for reading our manuscript and for his valuable comments about section 5.

The entire comment of Prof. Zetzsch concentrates to the discussion in chapter 5.2 of the manuscript about the estimation of Wittmer et al., 2015a to the amount of ISA iron required for a significant $CH_4$ depletion increase by atomic Cl generation.

Overall comment: The comments from Prof. Zetzsch are concluded at the end of page C5 by: "The technical application of ISA for climate engineering (how to obtain and apply a finer size distribution of $FeCl_3$, $FeOOH$ or $Fe_2O_3$ or coatings of sublimed

FeCl3 on solid carrier particles or to add iron-containing compounds to the fuel of combustion engines and power plants) is beyond the scope of our studies. Âż The global comment (all the other pages) states that the experimental results described by Wittmer et al. (2015a, 2015b and 2016) are the first quantitative determination of the source strength of iron(III) photolysis-induced atomic Cl from aerosol and that the results are an experimental basis of the ISA method and that the conservative estimation of Wittmer et al., 2015a should not be involved into extrapolations to the real atmosphere and for climate engineering.

Reply: In the Wittmer et al., 2015a article the following sentences were found: "Furthermore, iron-doped sea-salt aerosols have been proposed as a method for climate engineering, aiming to enhance CH4 depletion with higher Cl levels in the marine boundary layer and to simultaneously fertilize the oceans95). Based on our results, one may try to estimate the feasibility of such a project95)". The concluding comment and statement of Prof. Zetzsch seems to contradict these sentences. These sentences and ref. 95 from the Wittmer et al., 2015a article refer to the ISA method variants 2 and 3 (chapter 6 of our manuscript) described by Meyer-Oeste, 2010 and this reference is the proof that the cited statement aims directly to the ISA method and "The technical application of ISA for climate engineering Âż. Furthermore, part of this work of Wittmer et al. has been presented in 2014 at the first "Climate Engineering Conference" in Berlin (Wittmer et al, 2014).

Comments of pages C4 and C5. These comments relate to page 40 lines 1246-1275 of the manuscript.

Reply: We took into account Prof. Zetzsch comments, we suppressed the word "wrong" and replaced the paragraphs concerned by the following sentences in the manuscript:

"Current CH4 depletion by °Cl is estimated from 3.3% (Platt et al., 2004) to 4.3% (Allan et al., 2007). According to the results of Wittmer (Wittmer et al., 2015b) at a Cl-/Fe(III) molar ratio of 101, this amount would rise fourfold: from 13 to 17%. 1. Wittmer et al.

[Figure]

used their results obtained at a Cl-/Fe(III) ratio of 51 at the pH of 3.3-3.5: 1.9 x 105 °Cl/cm3. We consider that this pH is suboptimal. Instead it should be used the results obtained at a Cl-/Fe(III) ratio of 101 at the pH of 2.1-2.3: 1.9 x 105 °Cl/cm3. Moreover, Wittmer et al. made two limitative estimations: 2. They only focused on the Cl delivery in the condensed state by coagulation as Cl- transfer option between ISA particles and the Cl source sea-salt aerosol ignoring other Cl sources, Cl aggregate states, and Cl transfer mechanisms. According to this model the ISA particles should lose in the daylight continuously their Cl- load by °Cl emission and as a consequence they could gain back Cl only by coagulation with sea-salt aerosol particles. As further consequences of this model the Cl-/Fe(III) ratio of ISA particles would decrease, their diameter increase and their residence time in the troposphere would decrease. But according to Graedel and Keene (1996) and Keene et al. (1999) the next prominent source of inorganic Cl in the troposphere beside sea-salt aerosol is vaporous HCl. This is the main source the ISA particles can refill the chloride lost by photolysis. The main Cl uptake mechanism from this Cl source is the sorption from the gaseous phase. Main HCl sources are the sea-salt reaction with acids, CH4 and further hydrocarbon reactions with °Cl (Keene et al., 1999), flue gases of coal, biomass and garbage combustion (McCulloch et al. 1999), as shown in the global Reactive Chlorine Emissions Inventory (Keene et al., 1999), HCl from chlorocarbons being a significant part (Sanhueza, 2001) in particular from CH3Cl which is the largest, natural contributor to organic chlorine in the atmosphere (Lobert et al, 1999). 3. They estimate that the global production rate of 1785 Tg/year of sea-salt aerosol Cl- has to be doped with iron at a Cl-/Fe(III) molar ratio of 51 meanwhile we consider it has to be estimated at a molar ratio of 101 (according to 1.). The calculations made with these limitative assumptions resulted in an iron demand of 56 Tg/year Fe(III) to obtain the desired CH4 depletion effect (Wittmer et al. 2015a). Whereas the calculations with a Cl-/Fe(III) ratio of 101 results in a Fe(III) demand of only 18 Tg/year – still with the limitative assumption that there is no further Cl- source than sea-salt. ISA is produced from pyrogenic iron oxides. Pyrogenic oxides have particle sizes lower than $0.1\mu$m. Diameters of the NaCl-diluted ISA particles of

the Wittmer tests (Wittmer et al. 2015a) are round about 0.5$\mu$m. This confirms the test results of Wittmer et al. as calculation basis without any cut. But Wittmer et al. made two other limitative assumptions: 4. ISA has the same particle size and corresponding surface range as sea-salt; 5. ISA has the same residence time as sea-salt aerosol in the troposphere."

Comment page C5 (beginning): "Pyrogenic oxide particles smaller than 0.1 $\mu$m, as proposed by the authors, are not covered by our estimation. The pH of the aerosol droplets (p. 39, lines 1233 and 1234) was not measured, but the pH of the stock solution (1 g NaCl per liter, doped with FeCl3), that is atomized to obtain the optimum size of the aerosol droplets. The pH of the droplets was not measured in our work. The low pH is a consequence of hydrolysis of FeCl3, and the gas phase is expected to contain the HCl gas volatilized from the atomized droplets of the stock solution. Âż

Reply: We agree that these particle sizes are not covered by Wittmer et al. estimations. As a matter of fact, known salt aerosol generation methods by vapor condensation or nebulization (Gupta et al., 2015; Biskos, et al. 2006) allow not only the flame descending ISA type 1 (Oeste, 2004), but also the condensation and nebulization descending ISA variants 2 and 3 to be produced with aerosol particle diameters between 0.1 and 0.01 $\mu$m. Diameters of salt aerosol particles according to these physical aerosol generation methods are up to, or more, than one order of magnitude smaller than of those used in the experiments by Wittmer et al., 2015a. According to Lim et al., 2006 and to Meyer-Oeste, 2010 the optimum $^\circ$Cl production by sunlight photolysis of FeCl3 solutions or ISA is generated in the acidic pH range. The efficient $^\circ$Cl generation is necessary for an efficient CH4 depletion ISA. Except if made by condensation and hydrolysis of FeCl3 vapor or by nebulization of pure FeCl3 solution, or produced by combustion to pyrogenic FeOOH and reaction and hydrolysis with HCl and H2O to FeCl3 solution: FeCl3 has an acidic pH from the beginning because it hydrolyses according to equation 1.

equation 1: FeCl3 + 2H2O → FeCl2OH + H3O+ + Cl-

**[ESDD]{.underline}**
Wittmer et al., 2015a referred to the ISA method (Meyer-Oeste, 2010) but in their extrapolation, they did not took into consideration the required pH of the ISA method.

Comments from page C5: Âń We did not claim that the optimum efficiency of Cl production occurs at pH 2 and independent of the distribution of HCl between gas and particle: the lower the pH the less chloride will be available in the particle phase. For HCl adsorbed on humidified $Fe_2O_3$ (Wittmer and Zetzsch, 2016), the optimum pH will be much lower than 2 since this may dissolve some of the iron(III). Finally, we would not like to further discuss our simplifying assumptions 2. to 5. (p. 40, line 1246-1275). These were meant to be restricted to Fe doped NaCl aerosol alone, implying an uncertainty of more than an order of magnitude of our estimate. We agree with the argument 6. (line 1276-1282) that the larger specific surface of combustion-derived $Fe_2O_3$ is promising. This is the reason why we had started to look into the interaction of a pyrogenic $Fe_2O_3$ material (characterized by Mössbauer spectroscopy) with HCl gas (p 43, line 1383). Âż

Reply: According to our chapter 2.2 the coagulation of small particles within aerosol plumes is retarded. Otherwise known bridging of intercontinental distances and particle abundance in polar glacial regions of these particles would be impossible. Wittmer et al., 2015a considered only sea-salt aerosol particles as transport vehicles for ISA and as only possible contact medium to gain chloride ions as °Cl source. It is well known that coal combustion is a major source of active chlorine (McCulloch et al., 1999; Keene et al., 1999; Sanhueza, 2001) as well as iron (Sedwick et al., 2007; Luo et al., 2008; Wang et al, 2015; Ito et al, 2016), thus both iron and chlorine are jointly issued by other mechanisms and sources. As stated in our chapter 5.2 below point 5, sea salt aerosol has residence times in the troposphere lower than one day according to its coarse particle diameters without any possible bridging of intercontinental distances. In reality the chloride transfer between sea-salt aerosol particles and ISA particles may take place without any touch or coagulation because the troposphere is an acidic environment. Troposphere is a source of organic and inorganic acids which are in permanent con-
tact with the sea-salt aerosol. The acid ingredients in contact with sea spray produce HCl. Further ISA is produced by combustion and even becomes elevated by flue gas plumes: acid precursors like SO2 or NOx are in higher concentrations within the flue gas plume comparing to the tropospheric environment. The acids generated by flue gas plume produce additional HCl by reaction with the sea-salt aerosol (von Glasow, 2000). As a result, ISA and ISA precursors may absorb any chloride requirement via HCl vapour from the sea-spray source by itself (Wittmer et al. 2016).

As a conclusion: Prof. Zetzsch comment did not present arguments against the results of our own extrapolations. We agree that extrapolations about the ISA requirement should be supported in the future by detailed model calculations, but the generation of such models and future experimental work will be the subject of separate articles.

Bibliography:

Biskos, G., Malinowski, A., Russell, L. M., Buseck, P. R., & Martin, S. T. (2006). Nano-size effect on the deliquescence and the efflorescence of sodium chloride particles. Aerosol Science and Technology, 40(2), 97-106.

Graedel, T. E., & Keene, W. C. (1996). The budget and cycle of Earth's natural chlorine. Pure and applied chemistry, 68(9), 1689-1697.

Gupta, D., Kim, H., Park, G., Li, X., Eom, H.-J., Ro, C.-U., 2015: Hygroscopic properties of NaCl and NaNO3 mixture particles as reacted inorganic sea-salt aerosol surrogates. Atmos. Chem. Phys., 15, pp. 379-3393.

Ito, A., & Shi, Z., 2016. Delivery of anthropogenic bioavailable iron from mineral dust and combustion aerosols to the ocean. Atmospheric Chemistry and Physics, 16(1), 85-99.

Keene, W.C., Khalil, A.M., Erickson III, D.J., McCulloch, A., Graedel, T.E., Lobert, J.M., Aucott, M.L., Gong, S.L., Harper, D.B., Kleiman, G., Midgeley, P., Moore, R.M., Seuzaret, C., Barri, L.A., Li, Y.F., 1999. Composite global emissions of reactive chlorine from anthropogenic and natural resources: Reactive Chlorine Emissions Inventory. Journal of Geophysical Research, 104(D7), 8429-8440.

Lim, M., Chiang, R., Amal, R. 2006: Photochemical synthesis of chlorine gas from iron(III) and chloride solution. Journal of Photochemistry and Photobiology A: Chemistry, 183, pp. 126-132.

Luo, C., Mahowald, N., Bond, T., Chuang, P. Y., Artaxo, P., Siefert, R., ... & Schauer, J. (2008). Combustion iron distribution and deposition. Global Biogeochemical Cycles, 22(1).

Lobert, J. M., Keene, W. C., Logan, J. A., & Yevich, R. 1999. Global chlorine emissions from biomass burning: Reactive chlorine emissions inventory. Journal of Geophysical Research: Atmospheres, 104(D7), 8373-8389.

McCulloch, A., Aucott, M. L., Benkovitz, C. M., Graedel, T. E., Kleiman, G., Midgley, P. M., & Li, Y. F. (1999). Global emissions of hydrogen chloride and chloromethane from coal combustion, incineration and industrial activities: Reactive Chlorine Emissions Inventory. Journal of Geophysical Research: Atmospheres, 104(D7), 8391-8403.

Meyer-Oeste, F.D., 2010: Method for cooling the troposphere. Int. Patent CA 2748680 A1.

Oeste, F.D., 2004: Climate cooling by interaction of artificial loess haze with seasalt haze induced by iron- or titanium-doped ship- and aircraft-fuel. GeoLeipzig, Gemeinschaftstagung DGG und GGW, Schriftenreihe der Deutschen Geologischen Gesellschaft, Heft 4.

Sanhueza, E. (2001). Hydrochloric acid from chlorocarbons: a significant global source of background rain acidity. Tellus B, 53(2).

Sedwick, P. N., Sholkovitz, E. R., & Church, T. M. (2007). Impact of anthropogenic combustion emissions on the fractional solubility of aerosol iron: Evidence from the Sargasso Sea. Geochemistry, Geophysics, Geosystems, 8(10).

von Glasow, R.: Modeling the gas and aqueous phase chemistry of the marine boundary layer. Ph.D. thesis, Univ. Mainz, Mainz, Germany, 2000 (Available at www.rolandvonglasow.de).

Wang, R., Balkanski, Y., Boucher, O., Bopp, L., Chappell, A., Ciais, P., ... & Tao, S. (2015). Sources, transport and deposition of iron in the global atmosphere. Atmospheric Chemistry and Physics, 15(11), 6247-6270.

Wittmer J., Bleicher S., Zetzsch C., Oeste FD., 2014. Exploring the activation of chloride by iron(III) salt for a reduction of methane as part of the ISA method for CE. Climate Engineering Conference Berlin. Berlin, DE, 2014.

Wittmer, J., Bleicher, S., Ofner, J., Zetzsch, C., 2015a: Iron(III)-induced activation of chloride from artificial sea-salt aerosol. Environmental Chemistry, 12(4), doi: 10.1071/EN14279.

Wittmer J., Bleicher S., Zetzsch C., 2015b. Iron(III)-induced activation of chloride and bromide from modeled salt pans. The Journal of Physical Chemistry A, 119, 4373-4358.

Wittmer, J. and Zetzsch, C., 2016: Photochemical activation of chlorine by iron-oxide aerosol. J. Atm. Chem., doi: 10.1007/s10874-016-9336-6.

---

## Author Comment (AC5) · 5 Dec 2016

We deeply thank the reviewer for his time and for his very helpful and important comments. We have modified the manuscript accordingly and we think that this has largely improved our manuscript in particular for the chapter concerning the effects of iron over the continents (chapter 5), and the concluding chapter 10. Please find below a detailed response to each comments point by point.

Comment 1: The English is not adequate in the paper. Reply 1: The article has been read again carefully and many corrections have been carried out and only one dictionary has been used the "American English". Also one of the reviewers kindly suggested sending a "listing of edits to the European English along with a few typo identifications".

[Figure]

We hope it would be possible to integrate them.

Comment 2: The organization is poor (most of the conclusions talk about the cost-effectiveness of this solution, instead of pulling ideas together from the paper). Reply 2 : Effectively the original section 8 was both a discussion and conclusion. The cost-effectiveness of the solution proposed is now on a separate discussion chapter 9 (pages 61-64) and the conclusion on chapter 10 (p64-67) now gathers the ideas of the article, summarizing the numerous cooling effects found and the pros and cons of the proposal made, in particular by introduction of table 3.

Comment 3: The paper is full of details, but it gives the impression that the authors don't really understand what they are saying, as there are not always the right papers cited, and the latest ideas included in the paper, which could be a consequence of the great breadth discussed in the paper. Reply 3: As will be seen below on the answer to comment 6, the right articles are now cited and corrections are made, in particular about the work by Mahowald et al, and Boyd and Ellwood, Moore et al. In order to support the latest ideas included in the paper, the recent work on anthropogenic iron emissions is completed (in the conclusion, p68), as well as the chapter 5 (p47-53) about iron over land, forests, wetlands and soil. Thanks to the reviewer comments on this chapter 5, the review has been completed and we believe has been greatly improved. Some criticism of the reviewer seems to point to incomplete or improper concentration of details presented in our manuscript. After the ISA effects on atmospheric CH4 and CO2 concentrations, one of the central aims of our review is the try to demonstrate that the most severe consequence of the ongoing climate warming will be the decline of vertical mixing in ocean basins, as a consequence of halocline-and/or thermocline-induced development of stratified water columns. We ought to give a deeper development to this dangerous consequence. According to this aspect we completed our review by the following details (page 3 in the introduction chapter 1): "The severest consequences of such stratification to oceanic ecosystems is the development of anoxic milieu within stratified ocean basins. A recent example of the

development of a halocline stratification is the Black Sea (Eckert et al., 2013). There exists a stable pycnocline which coincides with a chemocline, dividing an oxic surface water layer from an anoxic sulfidic deep layer and the basin bottom contains a black sapropel sediment rich in organic C (Eckert et al., 2013). On page 41 of sub-chapter 4.1: Typical marks of past episodes with stratified water columns in ocean basins are black shale or black limestones as sapropel remnants. Stratified ocean basins during the Phanerozoic epoch occurred as a consequence of elevated CO2 levels in the atmosphere. This caused high sea surface temperatures (Meyers, 2014), and as a global consequence: global increase of evaporation, precipitation and as well production of brines of higher concentrations. It has been pointed out, that the increasing melt water run-off from polar and subpolar ice layers may induce the cover of denser ocean water by a melt water layer (Hansen et al., 2016). The generation of increasing precipitation and surface water run-off accompanied with increasing brine production accompanied by elevated surface water temperatures during hot CO2-high climate episodes had just the same consequences in the past geological epochs as we learn from Meyers (Meyers, 2014). Even the largest mass extinction of biota within the Phanerozoic epoch during the Permian-Triassic transition has been induced by high temperatures as a consequence of elevated CO2-Levels which induced the change of a well-mixed oxic to a stratified euxinic-anoxic ocean (Kaiho et al., 2016). What we have to face now is the extraordinary process developing from the recent situation: the combination of the CO2-dependent temperature rise-generated precipitation increase plus melt water increase. Mankind has to find the appropriate tool to stop and lead back this dangerous stratification process."

Comment 4: The paper does not convince that the feedbacks described will be large, just that they might exist. There are many other impacts NOT discussed that could offset the impacts suggested in the paper. Reply 4: Concerning possible other impacts not previously discussed that could offset the impacts suggested in the paper, the answer is given on reply 8. It is correct that several of the cooling effects cited and explained in this literature review just mention that they might exist, because there is little or no bibliography available and more research is required to study and model their climate impacts and evaluate their feedbacks. For at list two of them (CO2 removal by ocean fertilization with anthropogenic atmospheric iron and CH4 depletion by °Cl and photocatalytic Fe(II)/Fe(II), or by iron imput in the oceans or in terrestrial landscapes and wetlands) the feedbacks described are expected to be large as there are proposed at the end of the manuscript as possible solutions to fight and stop global warming. Currently, methane (CH4) in the troposphere is destroyed mainly by the hydroxyl radical °OH. From 3 to 4% CH4 (25 Tg/yr) (Allan et al., 2007; Graedel and Keene, 1996) become oxidized by °Cl in the troposphere, and larger regional effects are predicted, up to 5.4 to 11.6% CH4 (up to 75 Tg/yr) in the Cape Verde region (Sommariva and von Glasow, 2012) and ~10 to >20% of total boundary layer CH4 oxidation in some locations (Hossaini et al., 2016). According to known data (Wittmer et al., 2015), we tried to estimate the methane depleting power of ISA in the troposphere (section 2.2). We found promising results. Even if ISA had only this methane depleting ability, without any others, we ought to go on testing it as a climate controlling tool, and these tests can be done first by computer simulations, climate models, outdoors measurements of existing phenomena and some small scale indoors and outdoors experiments. Recently presented research results of Sherwen et al. (Sherwen et al., 2016a; Sherwen et al., 2016b), about the impact of tropospheric halogens in the atmosphere have shown that some VOC hydrocarbons of the troposphere (ethane, propane and acetone) are depleted from 18 to 9% by °Cl oxidation, the rest by °OH oxidation. ISA sunlight photolysis of Fe(III) can multiply tropospheric °Cl content. This and the research results of Wittmer et al. 2015a demonstrate the CH4 depletion power by the °Cl generating ISA method.

Comment 5: I am only able to review limited portions of the paper: particularly the atmospheric aerosol and impact on land and ocean biogeochemistry, and I found these parts of the paper to be incomplete to the point of wrong. Reply 5: Concerning the impact on ocean biogeochemistry, the answer is formulated in the following reply 6. The answer about the atmospheric aerosol is given below on reply 8, and for the impacts

on land biogeochemistry, the answer is given on reply 10.

Comment 6: In the introduction, the paper does a poor job describing the state of our knowledge of iron in the oceans. While I agree that we know less about iron than carbon this sentence is a problem: "meanwhile the iron biogeochemical cycle is only described in the ocean by few scientific publications (Boyd and Ellwood, 2010; Mahowald et al., 2005; Mahowald et al., 2009).:" Please correct these citations: Mahowald et la., 2005 and 2009 focus on the atmospheric iron cycle. For example, work by Ken Johnson, Moore et al., 2013 or Sigman et al., 2010 on the importance of iron in the oceans on ocean biogeochemistry in different time periods would be appropriate papers to cite here. Reply 6: Many thanks for these suggestions. The introduction of the manuscript has been improved concerning the description of the state of the scientific community knowledge of iron in the oceans. Many thanks for the corrections suggested about the atmospheric (and not oceanic) iron cycle. The corrections have been introduced in the manuscript by a modification of the incriminated sentence by the following ones on page 6 of chapter 1 (Introduction): "The biogeochemical cycles of carbon, nitrogen, oxygen, phosphorus, sulfur and water are well described in the literature, but the biogeochemical cycle of the Earth's iron is often overlooked. An overview of the progress made in the understanding of the iron cycle in the ocean is given by several authors (Breitbarth et al., 2010; Raiswell and Canfield, 2012). The current state of knowledge of iron in the oceans is lower than that of carbon, and still numerous scientific publications deal with this topic (Archer and Johnson, 2000; Boyd and Ellwood, 2010; Johnson et al., 2002a, b; Misumi et al., 2014; Moore and Braucher, 2008; Moore et al., 2013; Tagliabue et al., 2015; Turner and Hunter, 2001), meanwhile the iron biogeochemical cycle in the atmosphere is described by fewer ones (Mahowald et al., 2005; Mahowald et al., 2009; Mahowald et al., 2010), on the contrary to the iron biogeochemical cycle in soil and land, as almost no recent publications details the current knowledge of iron in soils and over the landscape (Anderson, 1982; Lindsay and Schwab, 1982; Mengel and Geurtzen, 1986)."

Also in the introduction we have underlined the recent modeling of the importance of iron from anthropogenic sources on page 7: "Recently the major role of soluble iron emissions from combustion sources became more evident. Today the anthropogenic combustion emissions play a significant role in the atmospheric input of soluble iron to the surface ocean (Sedwick et al., 2007). Combustion processes currently contribute from 20 to 100% of the soluble iron deposition over many ocean regions (Luo et al., 2008). Model results suggest that human activities contribute to about half of the soluble Fe supply to a significant portion of the oceans in the Northern Hemisphere (Ito and Shi, 2015) and that deposition of soluble iron from combustion sources contributes more than 40% of the total soluble iron deposition over significant portions of the open ocean in the Southern Hemisphere (Ito, 2015). Anthropogenic aerosol associated with coal burning would be the major bioavailable iron source in the surface water of the oceanic regions (Lin et al., 2015). The higher than previously estimated Fe emission from coal combustion implies a larger atmospheric anthropogenic input of soluble Fe to the northern Atlantic and northern Pacific Oceans, which is expected to enhance the biological carbon pump in those regions (Wang et al., 2015b)."

The authors thank also the reviewer for the suggested citations. We have unsuccessfully searched for "...work by Ken Johnson, Moore et al., 2013 ż: related to "the importance of iron in the oceans on ocean biogeochemistry in different time periods" and we only found two references of these two authors together, dated from 2002. We did not find work from "Ken Johnson, 2013", but an article from "Ken Johnson, 2014" was not pertinent. We arrived to the conclusion that the reviewer meant another year and found that ń Kenneth S Johnson ż has done numerous interesting work on these topics (but not in 2013) and although not always the first author, we cited some articles which are related to the topics of our manuscript and that he co-authored (but without Moore as coauthor). We also found corresponding work from J. Keith Moore et al, 2013 (without Johnson as coauthor) and these works are now cited in several relevant parts of our manuscript. We thank the reviewer for his helpful suggestions that allowed completing our review. We decided not to refer to Sigman et al., (Sigman et al., 2010),

because they reduced their work about the vertical ocean cycle to the carbon burial in the ocean basin sediment phase. But this sediment is underlined by thick alkaline bed rock layers of a total volume of 2.3 x 1018 m3 containing olivine and pyroxene mineral phases and the basic igneous rock types as peridotite, serpentinite, diabas and basalt. This rock base below the sediment layer is known as intensive fractured aquifer. The convective water flow through this huge alkaline ocean crust volume is estimated to about 20 - 540 x 103 $km^3$/yr (Nielsen et al., 2006). The oceanic crust comprises the largest aquifer system of the Earth with an estimated rock volume of 2,3 x 109 $km^3$ and a fluid volume of 2 % of the total ocean or $\sim$107 $km^3$ (Orcutt et al., 2011). The total rock volume of the igneous oceanic crust is 6 to 10 times the total volume of the marine sediments (Ivarsson et al., 2016) The upper part of the igneous crust is characterized by extensive fracturing and microbial within the fractures (Bach and Edwards, 2003; Edwards et al., 2012; Orcutt et al., 2011). From insight into the ocean crust it is known, that it is full of veins and openings filled by carbonates (Coggon et al., 2012; Coggon et al., 2010).

Comment 7: People have rejected the idea of iron fertilization of the oceans for many reasons, and this is not well described in the paper. Are you arguing we should go back and debate this? You are not really discussing the state of knowledge of this debate, or countering it, but rather just ignoring the debate here? Reply 7: The limited knowledge about dissolved or even dispersed iron distributions in the ocean confirms the work of Tagliabue et al. (Tagliabue and Dutkiewicz, 2016): their calculation results about the residence time of iron in the ocean differs up to three orders of magnitude between the different published models. Recently arose the fact that iron from source regions can be transported over distances of several thousand kilometers across the ocean basins (Resing et al., 2015). The iron inputs into the ocean regions by atmospheric dust had been analyzed by several authors (Mahowald et al., 2005; Mahowald et al., 2009; Wang et al., 2015a). Boyd and Elwood (Boyd and Ellwood, 2010) report about the balance between iron supply and removal processes within the ocean. It is right that ocean iron fertilization is very controversial and has many opponents. The recent article from Boyd

and Bressac (Boyd and Bressac, 2016) opens again the debate and the discussion. Also in the latest IPCC report (IPCC 2013: Carbon and Other Biogeochemical Cycles) the ocean iron fertilization is mentioned as a carbon dioxide removal technology. This debate to decide if iron ocean fertilization is "credible and creditable" is out of the scope of this review article but as this method is compared to the ISA method, we completed the manuscript by the following paragraphs (p65). "The idea of ocean fertilization by iron to enhance the $CO_2$ conversion by phytoplankton assimilation came up within the last two decades. Proposed was the mixing of an iron salt solution by ships into the ocean surface. This idea was debated controversial. Example of this debate is the discussion between KS Johnson et al. and SW Chisholm et al. (Chisholm et al., 2002; Johnson and Karl, 2002). Deeper insight into this debate is given by Boyd and Bressac (Boyd and Bressac, 2016). The iron fertilization procedure tests done so far had been restricted to relatively small ocean regions (Boyd et al., 2007; Johnson et al., 2002a, b). These tests produced iron concentrations orders of magnitude above those produced by natural ISA processing which are in the single decadal order of milligrams of additional dissolved iron input per square meter per year. In this sense the ISA method is quite different from "iron fertilization". As known from satellite views, phytoplankton blooms induced by natural dust emission events from the Sahara, Gobi, and further dust sources there is no doubt about the fertilizing effect of iron. Meanwhile this kind of natural iron fertilization enhancing the transfer of $CO_2$-Carbon into organic sediment carbon via the oceanic food chain seems to be un-contradicted and accepted (Hansen et al., 2016)."

Comment 8: The authors do not seem to realize that if you add iron to aerosols, they will tend to absorb more incoming radiation, and thus warm the planet: so this is the opposite effect you want. Check Sokolik and Toon, 1999. Reply 8: The global aerosols effects are not trivail (Lohmann and Feichter, 2005; Mahowald et al., 2014) and the direct and indirect effects of absorbing aerosols in the troposphere depend of multiple variables and among them particle size distribution, optical depth, altitude, effects on cloud condensation nuclei, etc. (Conant et al., 2002; Nenes et al., 2002). For

instance the climate response of black carbon (BC) is highly dependent on the altitude of the aerosol: near the surface BC causes surface warming, whereas near the tropopause and in the stratosphere BC causes surface cooling, despite decreasing planetary albedo (Ban-Weiss et al., 2012; Crutzen, 2006). To take into account this reviewer's remark the following paragraphs are added to the manuscript on page 11: "Differing to any natural dust iron-containing mineral aerosol, the ISA aerosol does not contain any residual mineral components like $Fe_2O_3$ minerals known as strong radiation absorbers. Previous studies have shown that iron oxides are strong absorbers at visible wavelengths and that they can play a critical role in climate perturbation caused by dust aerosols (Sokolik and Toon, 1999; Zhang et al., 2015). Because the primary ochre colored aerosol particles emitted by the ISA (method I) have small diameters of <0.05 $\mu$m and are made of pure FeOOH they become easily and rapidly dissolved within the plume of acidic flue gas. The ISA FeOOH aerosol becomes emitted with the parallel generated flue gas plumes containing $SO_2$ and $NO_x$ as sulfuric and nitric acid generators. ISA stays up for weeks within the troposphere before becoming precipitated on the ocean surface. Because of their small diameter and high surface area, the aerosol particles will immediately react with HCl, generated as reaction product between sea-salt aerosol and the flue gas borne acids. The reaction product is an orange colored $FeCl_3$ aerosol: ISA. During day time the sunlight radiation bleaches ISA into $FeCl_2$ and $^{\circ}Cl$; at the night time the re-oxidation of ISA plus HCl absorption generates ISA again. The $FeCl_2$ aerosol particles are colorless at low humidity; pale green during high humidity episodes. The day time bleaching effect reduces the radiation absorption of ISA to much lower levels comparing to oxides like $Fe_2O_3$."

Comment 9: None of the section 2.1-2.4 convince me that these effects will be significant. I lost a bit of ability to understand after that, but it seems like many of these feedbacks are actually very long term, and not very helpful in the next 30-300 years (ie. Section 4.3: minimizing ch4 emissions from sediment and bedrock: is ch4 release from bedrock really a problem we have right now, or on geologic time scales?) Reply 9: We agree with the reviewer that CH4 release from methane hydrates sediment, permafrost and wetlands is a problem we have right now and in the very near future, and that CH4 emissions from bedrock is more a problem on geologic time scales. But to give the whole picture we need to depict the process. In order to be clear about these aspects we add the following paragraphs in the manuscript: "The oxygen-dependent life will have problems within a decreased vertical mixed ocean basin because of its decreasing oxygen content induced by climate warming. The additional input of methane bubbles increases the oxygen deficit death zones. Climate models predict declines in oceanic dissolved oxygen with global warming and the climate warming dependent decline of the oxygen content in many ocean regions has meanwhile become manifest (Stramma et al., 2010). Braking or even reversal of this trend by reducing the oxygen depleting methane emissions at least should help to prevent regions within the ocean basins from methane-induced oxygen deficit. Below the oceanic tropospheric regions with ISA sedimentation even the ocean water column becomes elevated in its iron content before sedimentation of the iron-containing particles at the bottom. Any methane molecule independent if it is existent in the sediment or just above in the water phase or excreted by bubbling into the water column further above, becomes oxidized far away before it arrives at the surface of the water column. By the help of iron containing enzymes even sulfate acts as an oxidant by the help of microbes to oxidize methane to CO2 and water. This prevents the water layers above from oxygen loss. Methane oxidation in wet environments by sulfate is done by sulfate or iron(III)-reducing microbes like archaea or bacteria (Basen et al., 2011). Because these microbes use iron-containing enzymes to do their anaerobic methane oxidation processes they act more efficient in iron-rich than in iron-poor environments (Sivan et al., 2011). The iron containing debris fall of ISA-fed dead phytoplankton and phytoplankton dependent food chain links even feeds the methane depleting sulfate reducer community within or near the sediment surface."

Comment 10: The whole section 4.4 seems totally off base: it is not thought that the iron inputs from Amazon are important but rather the P inputs, and they only operate on geologic time scales. Reply 10: We deeply thank the reviewer for pointing out to

us this insufficient description. Now the section 4.4 has been completely reviewed and is the full chapter 5 (pages 47-52). Effectively, for instance Okin (Okin et al., 2004), suggested that soil phosphorus turnover times range from 104 to 107 years. But ISA main components are iron and chloride, thus phosphorus is out of the scope of this review. To briefly summarize the paragraphs that have been added in the manuscript on chapter 5: • iron is involved in many important physiological processes in plants and numerous other living organisms; • although iron is only required in small amounts mainly as a micronutrient, and although iron is the fourth most abundant element of the earth's crust, because of solubility and bioavailability problems many living organisms including plants and humans are deficient in iron; • iron deficiency in human beings concerns 2 billion people and induces many heath problems among which anemia; • iron deficiency induced chlorosis in plants occurs mainly in calcareous and/or alkaline soils (30% of the Earth land surface). This problem can be solved by addition of soluble iron complexes to the soil, or by foliar sprays containing mineral iron (for instance FeSO4) or iron chelates (Fe-EDTA among others). FeSO4 and several organic iron complexes like iron-oxalate are known constituents of atmospheric dust, but unfortunately no published work was found about possible effects on plant chlorosis by foliar deposition of soluble iron from atmospheric dust or from anthropogenic combustion sources; • Fe(III) reduction by micro-organisms in peatlands, wetlands, rice paddies, tropical humid forests and other fresh water surfaces reduces the amount of CH4 release, either by inhibition of methanogens or by increasing CH4 oxidation. This is confirmed by experimental work consisting in artificial addition of several types of iron compounds. Unfortunately no published work was found about possible effects of natural dust or anthropogenic iron deposition on CH4 emissions by humid terrestrial landscapes, although some literature exists about the reduction of CH4 emissions by anthropogenic sulfate deposition.

Comment 11: I would recommend to the authors to focus one paper on each section, not just on describing possible mechanisms, but rather on calculating the impacts of ISA for each mechanism, and making sure the impacts are significant. Do a good job,

get each idea published, and then you can pull them together later. Reply 11: Finding a possible solution to the climate change challenge is rather complex and needs a multidisciplinary approach and the pieces of the puzzle are not independent but connected to each other by a web of interactions. Several experts (for instance (Hansen et al., 2016)) expressed the urgent warning that mankind has only short time left to address and control climate warming. As a consequence, mankind has to rapidly find possible ways of action, including feedback loops of influence, and cycles of causes. Of course, such tools and agents have to be rapidly evaluated against side-effects to ecosystems, human health, and last but not least their economic burdens. We believe that to improve climate models and integrated assessment models of global climate change it is needed to study inter-linkages between all natural and anthropogenic known or expected biogeochemical and physical climate impacts of the different elements studied and iron is an important one to integrate. The Gaia puzzle cannot be solved by using only a microscopic focus to find out and describe every fitting detail separately localized within the heap of puzzle pieces, all the more they are connected to each other by a web of interactions. Instead, a complete picture is necessary with an immediate presentation of all effects and interactions found out so far as a whole. Allowing an overall vision, with a description of the relevance of details and relevant influences as complete as possible is in our opinion the best way to act. This reviewer suggestion aims to the opposite: first, presentation of all puzzle pieces separately and then, after all the individual pieces have been described, at last publish the whole picture of the interactions of each of the separately described parts. In our eyes this recommendation cannot be the optimum solution to do an emergency job within the shortest time. Mankind has not enough time to act in such an inefficient way. To our knowledge it is the first time that as many iron-depending climate interactions are presented together and even if our review might be completed or needs corrections because some other impacts or effects are not discussed in the manuscript and could offset some of the impacts suggested, we are convinced that it is necessary to present all together as soon as possible the details found out. For optimum efficiency, we need to show to the
different geosciences' branches the most complete picture of the ISA method and of the iron effects on all the Earth compartments in order to prove the significance of its numerous impacts in comparison to possible alternatives.

Bibliography: Allan, W., Struthers, H., Lowe, D., 2007. Methane carbon isotope effects caused by atomic chlorine in the marine boundary layer: Global model results compared with Southern Hemisphere measurements. Journal of Geophysical Research: Atmospheres (1984–2012) 112. Anderson, W.B., 1982. Diagnosis and correction of iron deficiency in field crops‐an overview. Journal of Plant Nutrition 5, 785-795. Archer, D., Johnson, K., 2000. A model of the iron cycle in the ocean. Global Biogeochemical Cycles 14, 269-279. Bach, W., Edwards, K.J., 2003. Iron and sulfide oxidation within the basaltic ocean crust: implications for chemolithoautotrophic microbial biomass production. Geochimica et Cosmochimica Acta 67, 3871-3887. Ban-Weiss, G.A., Cao, L., Bala, G., Caldeira, K., 2012. Dependence of climate forcing and response on the altitude of black carbon aerosols. Climate dynamics 38, 897-911. Basen, M., Krüger, M., Milucka, J., Kuever, J., Kahnt, J., Grundmann, O., Meyerdierks, A., Widdel, F., Shima, S., 2011. Bacterial enzymes for dissimilatory sulfate reduction in a marine microbial mat (Black Sea) mediating anaerobic oxidation of methane. Environmental microbiology 13, 1370-1379. Boyd, P., Ellwood, M., 2010. The biogeochemical cycle of iron in the ocean. Nature Geoscience 3, 675-682. Boyd, P.W., Bressac, M., 2016. Developing a test-bed for robust research governance of geoengineering: the contribution of ocean iron biogeochemistry. Phil. Trans. R. Soc. A 374, 20150299. Boyd, P.W., Jickells, T., Law, C., Blain, S., Boyle, E., Buesseler, K., Coale, K., Cullen, J., De Baar, H., Follows, M., 2007. Mesoscale iron enrichment experiments 1993-2005: Synthesis and future directions. science 315, 612-617. Breitbarth, E., Achterberg, E.P., Ardelan, M., Baker, A.R., Bucciarelli, E., Chever, F., Croot, P., Duggen, S., Gledhill, M., Hassellöv, M., 2010. Iron biogeochemistry across marine systems–progress from the past decade. Biogeosciences 7, 1075-1097. Chisholm, S.W., Falkowski, P.G., Cullen, J.J., 2002. Response to the letter of Johnson, K.S. and Karl, D.M. Science 296, 467-468. Coggon, R.M., Teagle, D., Harris, M., John, C.,

Smith-Duque, C., Alt, J., 2012. Why Does Calcium Carbonate Precipitate in the Ocean Crust?, AGU Fall Meeting Abstracts, p. 0545. Coggon, R.M., Teagle, D.A., Smith-Duque, C.E., Alt, J.C., Cooper, M.J., 2010. Reconstructing past seawater Mg/Ca and Sr/Ca from mid-ocean ridge flank calcium carbonate veins. Science 327, 1114-1117. Conant, W.C., Nenes, A., Seinfeld, J.H., 2002. Black carbon radiative heating effects on cloud microphysics and implications for the aerosol indirect effect 1. Extended Köhler theory. Journal of Geophysical Research: Atmospheres 107. Crutzen, P.J., 2006. Albedo enhancement by stratospheric sulfur injections: a contribution to resolve a policy dilemma? Climatic change 77, 211-220. Eckert, S., Brumsack, H.-J., Severmann, S., Schnetger, B., März, C., Fröllje, H., 2013. Establishment of euxinic conditions in the Holocene Black Sea. Geology 41, 431-434. Edwards, K.J., Fisher, A.T., Wheat, C.G., 2012. The deep subsurface biosphere in igneous ocean crust: frontier habitats for microbiological exploration. Frontiers in Microbiology 3, 116-126. Graedel, T.E., Keene, W., 1996. The budget and cycle of Earth's natural chlorine. Pure and applied chemistry 68, 1689-1697. Hansen, J., Sato, M., Hearty, P., Ruedy, R., Kelley, M., Masson-Delmotte, V., Russell, G., Tselioudis, G., Cao, J., Rignot, E., 2016. Ice melt, sea level rise and superstorms: evidence from paleoclimate data, climate modeling, and modern observations that 2° C global warming is highly dangerous. Atmospheric Chemistry and Physics Discussions 15, 20059-20179. Hossaini, R., Chipperfield, M.P., Saiz‐Lopez, A., Fernandez, R., Monks, S., Brauer, P., Glasow, R., 2016. A global model of tropospheric chlorine chemistry: organic versus inorganic sources and impact on methane oxidation. Journal of Geophysical Research: Atmospheres. Ito, A., 2015. Atmospheric processing of combustion aerosols as a source of bioavailable iron. Environmental Science & Technology Letters 2, 70-75. Ito, A., Shi, Z., 2015. Delivery of anthropogenic bioavailable iron from mineral dust and combustion aerosols to the ocean. Atmospheric Chemistry and Physics Discussions 15, 23051-23088. Ivarsson, M., Bengtson, S., Neubeck, A., 2016. The igneous oceanic crust–Earth's largest fungal habitat? Fungal Ecology 20, 249-255. Johnson, K.S., Karl, D.M., 2002. Is ocean fertilization credible and creditable? Science 296, 467-468. Johnson, K.S., Moore, J.K.,

Smith, W.O., 2002a. A report on the US JGOFS workshop on iron dynamics in the carbon cycle. Citeseer. Johnson, K.S., Moore, J.K., Smith, W.O., 2002b. Workshop highlights iron dynamics in ocean carbon cycle. Eos, Transactions American Geophysical Union 83. Kaiho, K., Saito, R., Ito, K., Miyaji, T., Biswas, R., Tian, L., Sano, H., Shi, Z., Takahashi, S., Tong, J., 2016. Effects of soil erosion and anoxic–euxinic ocean in the Permian–Triassic marine crisis. Heliyon 2, e00137. Lin, Y.C., Chen, J.P., Ho, T.Y., Tsai, I., 2015. Atmospheric iron deposition in the northwestern Pacific Ocean and its adjacent marginal seas: the importance of coal burning. Global Biogeochemical Cycles 29, 138-159. Lindsay, W., Schwab, A., 1982. The chemistry of iron in soils and its availability to plants. Journal of Plant Nutrition 5, 821-840. Lohmann, U., Feichter, J., 2005. Global indirect aerosol effects: a review. Atmospheric Chemistry and Physics 5, 715-737. Luo, C., Mahowald, N., Bond, T., Chuang, P., Artaxo, P., Siefert, R., Chen, Y., Schauer, J., 2008. Combustion iron distribution and deposition. Global Biogeochemical Cycles 22. Mahowald, N., Albani, S., Kok, J.F., Engelstaeder, S., Scanza, R., Ward, D.S., Flanner, M.G., 2014. The size distribution of desert dust aerosols and its impact on the Earth system. Aeolian Research 15, 53-71. Mahowald, N.M., Baker, A.R., Bergametti, G., Brooks, N., Duce, R.A., Jickells, T.D., Kubilay, N., Prospero, J.M., Tegen, I., 2005. Atmospheric global dust cycle and iron inputs to the ocean. Global biogeochemical cycles 19. Mahowald, N.M., Engelstaedter, S., Luo, C., Sealy, A., Artaxo, P., Benitez-Nelson, C., Bonnet, S., Chen, Y., Chuang, P.Y., Cohen, D.D., 2009. Atmospheric Iron Deposition: Global Distribution, Variability, and Human Perturbations*. Annual Review of Marine Science 1, 245-278. Mahowald, N.M., Kloster, S., Engelstaedter, S., Moore, J.K., Mukhopadhyay, S., McConnell, J.R., Albani, S., Doney, S.C., Bhattacharya, A., Curran, M., 2010. Observed 20th century desert dust variability: impact on climate and biogeochemistry. Atmospheric Chemistry and Physics 10, 10875-10893. Mengel, K., Geurtzen, G., 1986. Iron chlorosis on calcareous soils. Alkaline nutritional condition as the cause for the chlorosis. Journal of Plant Nutrition 9, 161-173. Meyers, P.A., 2014. Why are the $\delta$13Corg values in Phanerozoic black shales more negative than in modern marine organic matter? Geochemistry, Geophysics, Geosystems 15, 3085-3106. Misumi, K., Lindsay, K., Moore, J.K., Doney, S.C., Bryan, F.O., Tsumune, D., Yoshida, Y., 2014. The iron budget in ocean surface waters in the 20th and 21st centuries: projections by the Community Earth System Model version 1. Biogeosciences 11. Moore, J., Braucher, O., 2008. Sedimentary and mineral dust sources of dissolved iron to the world ocean. Biogeosciences 5. Moore, J.K., Lindsay, K., Doney, S.C., Long, M.C., Misumi, K., 2013. Marine ecosystem dynamics and biogeochemical cycling in the Community Earth System Model [CESM1 (BGC)]: Comparison of the 1990s with the 2090s under the RCP4. 5 and RCP8. 5 scenarios. Journal of Climate 26, 9291-9312. Nenes, A., Conant, W.C., Seinfeld, J.H., 2002. Black carbon radiative heating effects on cloud microphysics and implications for the aerosol indirect effect 2. Cloud microphysics. Journal of Geophysical Research: Atmospheres 107. Nielsen, S.G., Rehkämper, M., Teagle, D.A., Butterfield, D.A., Alt, J.C., Halliday, A.N., 2006. Hydrothermal fluid fluxes calculated from the isotopic mass balance of thallium in the ocean crust. Earth and Planetary Science Letters 251, 120-133. Okin, G.S., Mahowald, N., Chadwick, O.A., Artaxo, P., 2004. Impact of desert dust on the biogeochemistry of phosphorus in terrestrial ecosystems. Global Biogeochemical Cycles 18. Orcutt, B.N., Sylvan, J.B., Knab, N.J., Edwards, K.J., 2011. Microbial ecology of the dark ocean above, at, and below the seafloor. Microbiology and Molecular Biology Reviews 75, 361-422. Raiswell, R., Canfield, D.E., 2012. The iron biogeochemical cycle past and present. Geochemical Perspectives 1, 1-2. Resing, J.A., Sedwick, P.N., German, C.R., Jenkins, W.J., Moffett, J.W., Sohst, B.M., Tagliabue, A., 2015. Basin-scale transport of hydrothermal dissolved metals across the South Pacific Ocean. Nature 523, 200-203. Sedwick, P.N., Sholkovitz, E.R., Church, T.M., 2007. Impact of anthropogenic combustion emissions on the fractional solubility of aerosol iron: Evidence from the Sargasso Sea. Geochemistry, Geophysics, Geosystems 8. Sherwen, T., Evans, M.J., Carpenter, L.J., Schmidt, J.A., Mickely, L., 2016a. Halogen chemistry reduces tropospheric O3 radiative forcing. Atmos. Chem. Phys. Discuss., doi 10. Sherwen, T., Schmidt, J., Evans, M., Carpenter, L., Großmann, K., Eastham, S., Jacob, D., Dix, B., Koenig, T., Sinreich, R., 2016b. Global impacts of

tropospheric halogens (Cl, Br, I) on oxidants and composition in GEOS-Chem. Atmospheric Chemistry and Physics Discussions, 1-52. Sigman, D.M., Hain, M.P., Haug, G.H., 2010. The polar ocean and glacial cycles in atmospheric CO2 concentration. Nature 466, 47-55. Sivan, O., Adler, M., Pearson, A., Gelman, F., Bar-Or, I., John, S.G., Eckert, W., 2011. Geochemical evidence for iron-mediated anaerobic oxidation of methane. Limnology and Oceanography 56, 1536-1544. Sokolik, I.N., Toon, O.B., 1999. Incorporation of mineralogical composition into models of the radiative properties of mineral aerosol from UV to IR wavelengths. Journal of Geophysical Research 104, 9423-9444. Sommariva, R., von Glasow, R., 2012. Multiphase halogen chemistry in the tropical Atlantic Ocean. Environmental science & technology 46, 10429-10437. Stramma, L., Schmidtko, S., Levin, L.A., Johnson, G.C., 2010. Ocean oxygen minima expansions and their biological impacts. Deep Sea Research Part I: Oceanographic Research Papers 57, 587-595. Tagliabue, A., Aumont, O., DeAth, R., Dunne, J.P., Dutkiewicz, S., Galbraith, E., Misumi, K., Moore, J.K., Ridgwell, A., Sherman, E., 2015. How well do global ocean biogeochemistry models simulate dissolved iron distributions? Global Biogeochemical Cycles. Tagliabue, A., Dutkiewicz, S., 2016. Iron Model Intercomparison Project (FeMIP). Working Group proposal submitted to SCOR April 2016. Available at: http://www.scor-int.org/Annual%20Meetings/2016GM/FeMIP.pdf. . Turner, D.R., Hunter, K.A., 2001. The biogeochemistry of iron in seawater. Wiley Chichester. Wang, R., Balkanski, Y., Bopp, L., Aumont, O., Boucher, O., Ciais, P., Gehlen, M., Peñuelas, J., Ethé, C., Hauglustaine, D., 2015a. Influence of anthropogenic aerosol deposition on the relationship between oceanic productivity and warming. Geophysical Research Letters 42. Wang, R., Balkanski, Y., Boucher, O., Bopp, L., Chappell, A., Ciais, P., Hauglustaine, D., Peñuelas, J., Tao, S., 2015b. Sources, transport and deposition of iron in the global atmosphere. Atmospheric Chemistry and Physics 15, 6247-6270. Wittmer, J., Bleicher, S., Ofner, J., Zetzsch, C., 2015. Iron (III)-induced activation of chloride from artificial sea-salt aerosol. Environmental Chemistry 12, 461-475. Zhang, X., Wu, G., Zhang, C., Xu, T., Zhou, Q., 2015. What is the real role of iron oxides in the optical properties of dust aerosols? Atmospheric Chemistry and Physics

15, 12159-12177.

---

## Author Response (AR1)

This PDF files contains on pages 1 and 2 the point-to-point response to the reviewers and the list of all relevant changes made in the manuscript, inclusive the revised figures, the expanded table 1 and the new tables 2 and 3. Then starting page 3 till the end on page 86, the revised manuscript with yellow marked paragraphs for every relevant changes made.

**Point-to-point response to the reviewers and list of all relevant changes made in the manuscript: "Climate engineering by mimicking the natural dust climate control: the Iron Salt Aerosols method".**

**Authors:**

Franz Dietrich OESTE *[1], Renaud de_RICHTER *[2], Tingzhen MING [3], Sylvain CAILLOL [2]

* corresponding authors

**Affiliations & Addresses:**

gM-Ingenieurbüro, Tannenweg 2, D-35274 Kirchhain, Germany. Email: oeste@gm-ingenieurbuero.com

Institut Charles Gerhardt Montpellier − UMR5253 CNRS-UM2 − ENSCM-UM1 – Ecole Nationale Supérieure de Chimie de Montpellier, 8 rue de l'Ecole Normale, 34296 Montpellier Cedex 5, France. Email: renaud.derichter@gmail.com

School of Civil Engineering and Architecture, Wuhan University of Technology, No. 122, Luoshi Road, Hongshan District, Wuhan, 430070 China.

| Comments from Dr. Elliott | | Completions of the paper as a response to the reviewers' comment | |
|---|---|---|---|
| Comment N° | Summary of reviewer's comment | Pages | Lines |
| 1 | General critic about geoengineering | 60-61 | 2024-2076 |
| 2 | European English used | English corrected in all the manuscript. Also Dr. Elliott proposed to send a list of edits and typos corrections | |

| Anonymous reviewers' comments | | Completions of the paper as a response to the reviewers' comment | |
|---|---|---|---|
| Comment N° | Summary of reviewer's comment | Pages | Lines |
| 1 | The English is not adequate | English corrected in all the manuscript | |
| 2 | Organization is poor; most conclusion talk is about cost effectiveness | 38, 39-41, 47, 62-63, 64-70 | 1219-1223, 1292-1332, 1572-1576, 2103-2385, 2222-2385, |
| 3 | Not always the right papers discussed and too great breadth discussed | 2-3, 6, 7-9 | 61-94, 181-193, 253-398 |

| 4 | Did not convince that the described feedbacks exist or may be large | Responses included in one or more of the responses 2-3 and 6-12 | |
| 5 | Atmospheric aerosol impacts on land and ocean biogeochemistry are incomplete or wrong | Responses included in one or more of the responses 2-3 and 6-12, especially 6, 8 and 10 | |
| 6 | Paper does a poor job describing the knowledge of iron in the oceans | 35, 43-45 | 1167-1171, 1407-1509 |
| 7 | People have rejected the idea of iron fertilization; this is not well described in the paper | 6, 60-61 | 210-215, 2025-2077 |
| 8 | Authors do not realize that if iron is added to aerosols, it may warm the planet by incoming radiation absorption | 49-50, 59 | 1660-1671, 2007-2018 |
| 9 | None of the effects (*albedo increase, methane oxidation, black carbon oxidation and ozone depletion*) convince that these effects will be significant | 10, 12-13, 18, 22 | 344-348, 421-433, 589-593, 721-725. |
| 10 | The dust input to the Amazon rainforest as contribution delivery to climate cooling seems totally off base | 50-53 | 1689-1799 |
| 11 | The reviewer recommends to focus for each mechanism in separate articles and then pull them together later | As a response the paper has been reorganized and an answer has been made to the reviewer insisting on the pluri-disciplinary needs to fight global warming and to complete biogeochemical cycles | |

| Comments from other readers | | Completions of the paper as a response to the comments received | |
|---|---|---|---|
| reader name | Summary of reviewer's comments | Pages | Lines |
| Sander | Data in table 1 is not from Sander, 2005 | 17 | 548-551 |
| Zetzsch 1 | The technical application of ISA for climate engineering is beyond the scope of our studies | Sentences in published work (ref. Wittmer et al., 2015a) state the contrary. | |
| Zetzsch 2 | Technical comments on pH and particle size | 54-55 | 1840-1855 |

[revised manuscript text omitted]

particles.

[Figure]

**Figure 2.** simplified chemical reaction scheme of the generation of chlorine radicals by iron salt aerosols under sunlight radiation and the reaction of the chlorine radicals with atmospheric methane.

At droplet or particle diameters below 1 µm, between 1 µm and 0,1 µm, contact or
coagulation actions between the particles within aerosol clouds are retarded [112, 142-144].
Otherwise the aerosol lifetime would be too short to bridge any intercontinental distance or
arrive in polar regions. That reduces the possible Cl⁻ exchange by particle contact. But
absorption of gaseous HCl by reactive iron oxide aerosols resulting in Fe(III) chloride
formation at the particle surfaces is possible [127]. Gaseous HCl and further gaseous chloro-
compounds are available in the troposphere: HCl (300 pptv above the oceans and 100 pptv
above the continents) [118], ClNO$_2$ (up to 1500 pptv near flue gas emitters) [145, 146] and
CH$_3$Cl (550 pptv remote from urban sources) [147, 148]. By or after sorption and reactions
such as photolysis, oxidation, and reduction, any kind of these chlorine species can induce
chloride condensation at the ISA particle surface. Acid tropospheric aerosols and gases such
as H$_2$SO$_4$, HNO$_3$, oxalic acid, and weaker organic acids further induce the formation of
gaseous HCl from sea-salt aerosol [149-151]. Since 2004, evidence and proposals for
possible catalyst-like sunshine-induced cooperative heterogeneous reaction between Fe(II),

Fe(III), $Cl^-$, $°Cl$, and HCl fixed on mineral dust particles and in the gaseous phase on the $CH_4$ oxidation are known [127, 141]. Further evidence of sunshine-induced catalytic cooperation of Fe and Cl came from the discovery of $°Cl$ production and $CH_4$ depletion in volcanic eruption plumes [152, 153]. Wittmer et al. presented sunshine-induced $°Cl$ production by iron oxide aerosols in contact with gaseous HCl [127]. Further evidence comes from $°Cl$ found in tropospheric air masses above the South China Sea [154]. It is known that the troposphere above the South China Sea is often in contact with Fe-containing mineral dust aerosols (~18 $g\ m^{-2}\ a^{-1}$) [155], which is further evidence that the Fe oxide-containing mineral dust aerosol might be a source for the $°Cl$ content within this area.

HCl, water content and pH within the surface layer of the aerosol particles depend on the relative humidity. Both liquid contents, $H_2O$ and HCl, grow with increasing humidity [156]. In spite of growing HCl quantity with increasing humidity, pH increases, due to decreasing HCl concentration within the surface layer. Hence, since the radiation induced $°Cl$ production decreases with decreasing pH, the $°Cl$ emission decreases in humid conditions [127]. Under dry conditions, even sulfate may be fixed as solid Na-sulfate hydrates. Solubilized sulfate slightly inhibits the iron induced $°Cl$ production [157].

Night or early morning humidity produces similarly the maximum chloride content on the liquid aerosol particles surface. During day time, the humidity decrease induces ISA photolysis and $Cl^-$ conversion to $°Cl$ production by decreasing water content and pH. The ISA particle surface layer comes to $Cl^-$ minima levels during after noon hours. In the continental troposphere with low sea salt aerosol level, these effects enable the pure ISA iron oxide aerosol particles to coat their surface with chloride solution at night and to produce chlorine atom emission at daytime.

Freezing has different effects on the primary wet ISA particles. Changing by CCN action to cloud droplets with solubilized chloride and iron content and when arriving to freezing conditions, the frozen ice is covered by a mother liquor layer with elevated concentration of both iron and chlorine. Some acids such as HCl do not decrease the mother liquor pH proportional to concentration and the behavior of the ice surfaces, grown from low salt content water, are different from high salt content water, thus the different kinds of ISA behave differently [158-160]. Direct measurements of molecular chlorine levels in the Arctic marine boundary layer in Barrow, Alaska, showed up to 400 pptv levels of molecular chlorine [161]. The Cl concentrations fell to near-zero levels at night but peaked in the early morning and late afternoon. The authors estimated that the Cl radicals oxidized on average more $CH_4$ than hydroxyl radicals, and enhanced the abundance of short-lived peroxy radicals.

Further investigations have to prove how the different types of ISA particles behave in clouds below the freezing point or in the snow layer at different temperatures: the primary salt-poor

Fe-oxide, the poor $FeCl_3$-hydroyzed and the $FeCl_3$-NaCl mixture, because the °Cl emission depends on pH, Fe and Cl concentration.

Additional to iron photolysis, in a different and day-time independent chemical reaction, iron catalyzes the formation of °Cl or $Cl_2$ from chloride by tropospheric ozone [162]. Triggering the

$CH_4$ decomposition, both kinds of iron and chlorine have a cooperative cooling effect on the troposphere: less GHG $CH_4$ in the atmosphere reduces the GH effect and allows more outgoing IR heat to the outer space [163].

These reactions had been active during the glacial period: Levine et al. [164] found elevated

$^{13}CH_4$ / $^{12}CH_4$ isotope ratios in those Antarctic ice core segments representing the coldest glacial periods. The much greater °Cl preference for $^{12}CH_4$ oxidation than $^{13}CH_4$ oxidation than by the °OH is an explanation for this unusual isotope ratio. Additional evidence gives the decreased $CH_4$ concentration during elevated loess dust emission epochs [165].

As shown in more detail in the next section 2.3, ISA produces °Cl and much more hydrophilic

°OH and ferryl as further possible $CH_4$ oxidants by the Fenton and photo-Fenton processes

[75]. To gain the optimal reaction conditions within the heterogeneous gaseous / liquid / solid phase ISA system in the troposphere the $CH_4$ reductant and the oxidant (Fenton and photo-

Fenton oxidant) have to be directed in a way, that oxidant and reductant can act within the identical medium.

As seen on table 1, according to the $CH_4$ Henry's law constant the preference of the 1.8 ppm tropospheric $CH_4$ is undoubtedly the gaseous phase. °Cl has also a preference for the gaseous phase.

**Table 1:** the Henry's law constants [166] and daylight stability for different gaseous or
vaporous components reacting with or produced by ISA in the troposphere

| Substance | Henry's law constant (mol $m^{-3}$ $Pa^{-1}$) | Stability against tropospheric day light (+ stable; - unstable) |
|---|---|---|
| $CH_4$ | $1.4 \times 10^{-5}$ | + |
| °Cl | $2.3 \times 10^{-2}$ | + |
| $Cl_2$ | $9.2 \times 10^{-4}$ | - |
| HCl | $1.5 \times 10^{1}$ | + |
| HOCl | 6.5 | - |
| °OH | $3.8 \times 10^{-1}$ | + |
| $H_2O_2$ | $8.3 \times 10^{2}$ | - |

Iron exists at least in part as Fe(III) during nighttime and at least in part as Fe(II) during daytime. The $CH_4$ oxidation by °Cl and °OH is restricted to the daytime as during night hours

°Cl and °OH recombine fast to $Cl_2$, HOCl, and $H_2O_2$ in the dark [167]. During daylight hours, these recombination products photolyze again by regeneration of the radicals. But even during day time these radicals and their recombination products co-exist due to the cycling
between $°Cl$, $°OH$, $Cl_2$, $HOCl$, and $H_2O_2$. This cycling is activated by sunlight photolysis and
radical recombination reactions [167, 168].

As we learn from Henry's law constants in Table 1 the oxygen species $°OH$ and $H_2O_2$ have a
much higher tendency to stay in the liquid phase than the chlorine species $°Cl$ and $Cl_2$. $Cl_2$
has the tendency to react with water of neutral pH by producing $HOCl$. But the pH values of
ISA, especially if ISA is emitted as acid flue gas plumes are lower than 3. Within this acidic
region the tendency of $HOCl$ generation from $Cl_2$ decreases to very low values and even at
those humidity levels when the ISA particles become deliquescent the majority of the
activated chlorine species will be localized in the gaseous phase containing $CH_4$, not in the
liquid phase.

But $°OH$ may leave the condensed phase into the gaseous phase at favorable circumstances
into the gaseous phase [169] and may contribute there to the oxidation of $CH_4$ during clear
dry conditions without liquid phase at the Fe(III) surfaces.

Comparably to the water-soluble Ammonia ($5.9 \times 10^{-1}$), $°OH$ has a similar Henry's law
constant. Therefore $°OH$ has the tendency to stay within hydrous phases during humid
conditions. This tendency is 16 times lower for $°Cl$. This property is combined with the
16 times higher reactivity in comparison to $°OH$. At an equal production of $°Cl$ and $°OH$, the
reaction of $°Cl$ with $CH_4$ has a probability of up to 250 times (16 x 16) that of the reaction of
$°OH$ with $CH_4$ when the ISA particles are wet and 16 times that of $°OH$ with $CH_4$ when the
ISA particles are dry. The probability of $CH_4$ oxidation by ISA derived $°Cl$ against ISA derived
$°OH$, may be restricted by the pH increase tendency within ISA during humid episodes
(decreased $°Cl$ generation on ISA with rising pH), to values fluctuating between the extremes
1 and 250. Independently of the kind of oxidants produced by ISA – during dry, clear sky, and
sunshine episodes - the ISA deriving oxidants produce maximum oxidant concentrations
within the $CH_4$-containing gaseous phase, producing optimum $CH_4$ depletion rates.

The $°Cl$ reactivity on most VOC other than $CH_4$ is at least one order of magnitude higher than
that of $°OH$ [170]. Halogen organics such as dichloromethane [171] as well as the
environmental persistent and bioaccumulating perfluoro organics such as perfluoro octane
sulfonate may be depleted by sunlit ISA [172].

### 2.3. Oxidation of organic aerosol particles containing black and brown carbon

Black carbon in soot is the dominant absorber of visible solar radiation in the atmosphere
[173]. Total global emission of black carbon is 7.5 Mt $yr^{-1}$ [174]. Direct atmospheric forcing of
atmospheric black carbon is +0.7 W $m^{-2}$ [174]. Above its climate relevance black carbon soot
induces severe health effects [128].

Andreae & Gelencsér [175] defined the differences between the carbons: black carbon contains insoluble elemental carbon, brown carbon contains at least partly soluble organic carbon. Black carbon contains as well additional extractable organics of more or less volatility and/or water-solubility [175, 176]..

Black and brown carbonaceous aerosols have a positive radiative forcing (warming effect) on clouds [177] as seen in sub-section 2.1, and also after deposition on snow, glaciers, sea ice or on the polar regions, as the albedo is reduced and the surface is darkened [178]. One of the most effective methods of slowing global warming rapidly on short-term is by reducing the emissions of fossil-fuel particulate black carbon, organic matter and reducing of tropospheric ozone [179].

Both aerosol types have adverse effects to health (human, animal, livestock, vegetal) and reducing its levels will save lives and provide many benefits [180].

Thus any tropospheric lifetime reduction of both dark carbons would gain cooling effects and further positive effects.

Both carbons are characterized by aromatic functions. The black carbons contain graphene structures; the brown ones have low-molecular weight humic-like aromatic substances (HULIS). HULIS derive from tarry combustion smoke residues and/or from aged secondary organic aerosol (SOA). The source of SOA are biogenic VOCs such as terpenes [181].

HULIS contain polyphenolic red-ox mediators such as catechol and nitro-catechols [182-

185].

The polyphenolic HULIS compounds are ligands with very strong binding to iron. Rainwater- dissolved HULIS prevent Fe(II) from oxidation and precipitation when mixing with seawater

[186]. Wood smoke derived HULIS nano-particles penetrate into living cell walls of respiratory epithelia cells. After arrival in the cells the HULIS particles extract the cell iron from the mitochondria by formation of HULIS iron complexes [187].

Beside iron, other metals such as manganese and copper have oxygen transport properties which improve the oxidation power of $H_2O_2$ by Fenton reactions generating °OH [188]. $H_2O_2$

is a troposphere-borne oxidant [189].

Polyphenolic and carboxylate ligands of HULIS enhance the dissolution of iron oxides. These ligands bind to un-dissolved iron oxides [75].

Iron and catechols are both reversible electron shuttles:

       $Fe^{2+} \longleftrightarrow Fe^{3+} + e$         (Eq. 1);

       catechol $\longleftrightarrow$ quinone + 2e   (Eq. 2).

The HULIS – iron connection enhances the oxidative degradation of organic compounds such as aromatic compounds [75].

Oxidant generation by reaction of oxidizable dissolved or un-dissolved metal cations such as

Fe(II), Cu(I) and Mn(II) with $H_2O_2$ had first been discovered for instance for Fe(II) in 1894

[190]. Since then these reactions are known as Fenton reactions. Mechanisms and
generated oxidants of the Fenton reactions are still under discussion.

According to the participating metal ligand oxidants such as $°OH$, $Fe(IV)O^{2+}$ (= Ferryl), $°Cl$,
$°SO_4^-$, organic peroxides and quinones may appear [191].

According to Barbusinsky et al. the primary reaction intermediate from $Fe^{2+}$ and $H_2O_2$ is the
adduct $\{Fe(II)H_2O_2\}^{2+}$ which is transformed into the ferryl complex $\{Fe(IV)(OH)_2\}^{2+}$. The latter
stabilizes as $\{Fe(IV)O\}^{2+} + H_2O$. Reductants may also react directly with $\{Fe(IV)O\}^{2+}$ or after
its decomposition to $Fe^{3+} + °OH + OH-$ by $°OH$. $Fe^{3+}$ reacts with $H_2O_2$ to $Fe^{2+}$ via $°O_2H$
development; the latter decays into $O_2 + H_2O$.

Light enhances the Fenton reaction effectiveness. It reduces $Fe^{3+}$ to $Fe^{2+}$ by photolysis
inducing $°OH$ or $°Cl$ generation, the latter in the case of available $Cl^-$, which reduces the $H_2O_2$
demand [192, 193].

This process is illustrated by figure 3.

[Figure]

**Figure 3**. Schematic representation of the cooling of the troposphere, by inducing the decrease of ozone and organic aerosol particles such as soot and smoke.

The Fenton reaction mechanism is dependent on pH and on the kinds of ligands bound to the Fenton metal. The reaction mechanism with oxidants of $SO_4^{2-}$, $NO_3^-$, $Cl^-$ and 1,2-dihydroxy benzene ligands had been studied [194].

In biological systems, 1,2-dihydroxy benzenes (catecholamines) regulate the Fenton reaction and orient it toward different reaction pathways [195].

Additionally, the fractal reaction environments like surface rich black and brown carbons and ISA are of considerable influence on the Fenton reaction. By expanding the aqueous interface, accelerations of the reaction velocity up to three orders of magnitude had been measured [196]. This may be one of the reasons why iron-containing solid surfaces made of fractal iron oxides, pyrite, activated carbon, graphite, carbon nanotubes, vermiculite, pillared clays, zeolites have been tested as efficient Fenton reagents [197-199].

Even the oxidation power of artificial Fenton and photo-Fenton systems is known to be high enough to hydroxylate aliphatic C-H bonds, inclusive $CH_4$ hydroxylation to methanol [200-202].

But the HULIS itself becomes depleted by the Fenton oxidation when it remains as the only reductant [195].

Like HULIS or humic substances, the different kinds of black carbons act as red-ox mediators due to their oxygen functionalities bound to the aromatic hexagon network such as hydroxyl, carbonyl, and ether [203, 204]. These functionalities act similarly as hydroquinone, quinone, aromatic ether, pyrylium and pyrone at the extended graphene planes as electron acceptor and donor moieties. Soot also possesses such red-ox mediator groups [205, 206]. Again these are ligands with well-known binding activity on iron compounds. Their difference to the HULIS ligands is that they are attached to stacks of aromatic graphene hexagon networks instead of mono- or oligo-cyclic aromatic hexagons of HULIS. As well as the HULIS red-ox mediator ligands these hydroxyl and ketone groups transfer electrons from oxidants to reductants and vice versa. Like the HULIS – iron couple, the black carbon - iron couple enhances the red-ox mediation above the levels of every individual electron shuttle [207-209]. Accordingly, any ISA doping of black carbons generates effective oxidation catalysts [210, 211].

Lit by sunlight the ISA doped soot represents an oxidation catalyst to adsorbed organics producing its own oxidants by the photo-Fenton reaction. In spite of the higher chemical stability of the graphene network of soot compared to HULIS soot, by wet oxidation further oxygen groups are fixed to the soot graphene stacks [212] increasing soot's hydrophilic property, which is necessary to arrange its rain-out. The hydroxyl radical attack resulting from the photo Fenton reaction at last breaks the graphene network into parts [213, 214]. Photo-

Fenton is much more efficient in °OH generation than Fenton, because Fe(III) reduction as
regeneration step occurs by Fe(III) photo reduction, rather than consuming an organic
reductant.

The oxidized hydrophilic carbon particles are more readily washed out of the atmosphere by
precipitation [215]. ISA accelerates this oxidation process as the iron-induced Fenton and
photo-Fenton reaction cycles produce hydroxyl and chlorine radical oxidants, speeding up
the soot oxidation.

Fe(III) forms colored complexes with hydroxyl and carboxylic hydroxyl groups too, particularly
if two of them are in 1,2 or 1,3 position, such as in oxalic acid. The latter belong to the group
of dicarboxylic acids known to be formed as oxidation products from all kind of volatile,
dissolved or particular organic carbons in the atmosphere [216]. Dicaboxylate complexes
with iron are of outstanding sensitivity to destruction by photolysation [217-220]: photolysis
reduces Fe(III) to Fe(II) by producing $H_2O_2$ and oxidation of the organic complex compounds.
Then Fe(II) is re-oxidized to Fe(III) by $H_2O_2$ in the Fenton reaction by generation of °OH [221].
According to their elevated polarity oxidation products containing hydroxyl and carboxyl
groups have increased wettability, are more water soluble and are thus rapidly washed out
from the atmosphere.

Due to their elevated reactivity compared to $CH_4$ the gas phase, oxidation of airborne organic
compounds by ISA-generated °OH or °Cl is enhanced. By eliminating black and brown
carbon aerosols, ISA contributes to global warming reduction and to decreasing polar ice
melting by surface albedo reduction caused by black-carbon snow contamination [173, 222].

The generation of ISA by combusting fuel oil with ferrocene or other oil soluble iron additives
in ship engines or heating oil burners has additional positive effects, because soot is
catalytically flame-oxidized in the presence of flame-borne ISA (detailed in chapter 6) as a
combustion product of the iron additive [223, 224].

**2.4.  Tropospheric Ozone depletion by ISA**

An additional GHG is the tropospheric ozone [179]. Carbon dioxide is the principal cause of
GW and represents $^2/_3$ of the global radiative forcing, but long lived methane and short lived
tropospheric ozone are both GHGs and respectively responsible of the 2nd and 3rd most
important positive radiative forcing.

According to Blasing [99, 122, 123] tropospheric $O_3$ has an atmospheric forcing of +0.4 Watt
per square meter. Any direct depleting action of tropospheric $O_3$ by the ISA-induced °Cl is
accompanied by an indirect emission decrease of $O_3$ as the reduction of $CH_4$ and further
VOC by the ISA method decreases the $O_3$ formation [225].

Reactive halogen species (mainly Cl, Br) cause stratospheric ozone layer destruction and
thus the "ozone layer hole". Tropospheric ozone destruction by reactive halogen species is also a reality [226]. Since long, °Cl and °Br are known as catalysts for $O_3$ destruction in the Stratosphere [227]. Investigations both in laboratory and nature have shown that °Br is a much more effective catalyst of ozone depletion within the troposphere than °Cl [161, 228, 229].

As discussed at the end of chapter 2.6 clear evidence exists, that the ozone depleting "bromine explosions" known as regular phenomenons developing from cost-near snow layers at sunrise in the polar spring [230, 231] are likely to be induced by the photolysed precipitation of iron containing dust. According to Pratt, bromide enriched brines covering acidified snow particles are oxidized by photolyzation to °Br.

In coastal areas of both the northern and southern Polar Regions during springtime, inert halide salt ions (mainly Br⁻) are converted by photochemistry into reactive halogen species (mainly Br atoms and BrO) that deplete ozone in the boundary layer to near zero levels [232]. During these episodes called "*tropospheric ozone depletion events*" or *"polar tropospheric ozone hole events*" $O_3$ is completely destroyed in the lowest kilometer of the atmosphere on areas of several million square kilometers and has a negative climate feedback or cooling effect [233].

In the tropics, halogen chemistry (mostly Br and I) is also responsible for a large fraction (~30%) of tropospheric ozone destruction [120, 234] and up to 7% of the global methane destruction is due to chlorine [121, 235]. It has been estimated that 25% of the global oxidation of $CH_4$ occurs in the tropical marine boundary layer [236]. A one-dimensional model has been used to simulate the chemical evolution of air masses in the tropical Atlantic Ocean [120] and to evaluate the impact of the measured halogens levels. In this model, halogens (mostly Br and I) accounted for 35−40% of total tropospheric $O_3$ destruction while the Cl atoms accounted for 5.4−11.6% of total $CH_4$ sinks. Sherwen et al. [226] estimate at -0.066 W m$^{-2}$ the radiative forcing reduction due to $O_3$ pre-industrial to present-day changes.

The ISA-induced increase of °Br concentration at sea-salt containing tropospheric conditions has been confirmed [125]. This establishes ISA as part of an ozone-depleting reaction cycle and additional cooling stage. This depletion effect of the GHG tropospheric ozone is worth noting.

**2.5. ISA induced phytoplankton fertilization albedo increase (by enhancing DMS-emissions) and $CH_4$ oxidation efficiency (by increasing MC- and DMS-emissions)**

One of the largest reservoirs of gas-phase chlorine is the about 5 Tg of methyl chloride (MC) in the Earth's atmosphere [147]. Methyl-chloride is released from phytoplankton [237] and from coastal forests, terrestrial plants and fungi [238].

Dimethylsulfide (DMS) is a volatile sulfur compound that plays an important role in the global sulfur cycle. Through the emission of atmospheric aerosols, DMS may control climate by influencing cloud albedo [239].

Currently, researchers [240] estimate that 28.1 (17.6–34.4) Tg of sulfur in the form of DMS

are transferred annually from the oceans into the atmosphere.

Ocean acidification has the potential to exacerbate anthropogenic warming through reduced

DMS emissions [241]. On the contrary, increased emissions of DMS and MC into the troposphere are a consequence of the ISA-induced phytoplankton growth and DMS + MC

release into the troposphere. DMS is oxidized in the troposphere to sulfuric and sulfonic acid aerosols, which are highly active CCN. This process enhances the direct ISA cooling effect according to cooling section 2.1 [239].

In contact with this acidic aerosol with sea spray aerosol, sulfate and sulfonate aerosols are formed and gaseous HCl is produced. Sulfate aerosols are known to have a negative radiative forcing (a cooling effect) [242].

A further HCl source is the oxidation of MC. Both effects induce the tropospheric HCl level to rise. According to cooling stage described in section 2.2, with the increased HCl level, additional chlorine atoms are produced by reaction with ISA. This effect further accelerates the $CH_4$ oxidation and its removal from the atmosphere, reducing its radiative forcing.

**2.6. Oxidation of $CH_4$ and further GHGs by sunlit solid surfaces**

Mineral aerosol particles adhere strongly to sunlit, dry and solid surfaces of rocks and stones.

A well-known remnant of the dust deposit in rock or stone deserts and rocky semi-arid regions is the orange, brown, red or black colored "Desert Varnish" coat covering stones and rocks. The hard desert varnish is the glued together and hardened residue of the primary dust deposit. Daily sun radiation and humidity change, as well as microbe and fungi influence grows up the varnish changing the primary aerosol deposit [243] by photolytic Fe(III) and

Mn(IV) reduction during daytime and night time oxidation of Fe(II) and Mn(II). The oxidation is triggered further by Mn and Fe oxidizing microbes adapted to this habitat [244, 245]. Desert varnish preserves the Fe and Mn photo reduction ability of the aerosol: lit by light the varnish can produce chlorine from chloride containing solutions [246]. The photo, humidity, and microbial induced permanent Fe and Mn valence change between night and day [247]

accompanied by adequate solubility changes seem to trigger the physico-chemical hardening of every new varnish layer.

The varnish is composed of microscopic laminations of Fe and Mn oxides. Fe plus Mn represent about $^1/_5$ of the varnish. Meanwhile $^4/_5$ of the laminations are composed of $SiO_2$, clay and former dust particles. Dominant mineral is $SiO_2$ and/or clay [248, 249]. There is little doubt that desert varnish can build up even from pure iron oxides or iron chloride aerosol deposits such as ISA. The optimum pH to photo-generate the methane oxidizing chlorine atoms from ISA is pH 2 [124]. Established by the gaseous HCl content of the troposphere

[118], a pH drop to pH 2 at the varnish surface is possible on neutral alkaline-free surfaces such as quartz, quartzite and sandstone. The humidity controlled mechanism acting between gaseous HCl and HCl dissolved in the liquid water layer absorbed on the solid iron oxide surface of ISA particles, as explained in the section 2.2, acts at the varnish surface analogue:

a $FeCl_3$ stock can pile up by Fe(II) oxidation and humidity-triggered HCl absorption during night time. The $FeCl_3$ stock at the varnish surface is consumed during daytime by photolytic

Fe(II) and chlorine atom generation.

ISA aerosol particles emit HCl during dry conditions. Like oxidic ISA, desert varnish absorbs

$H_2O$ and HCl from the atmosphere gathering it during night time as surface-bound $H_2O$, $OH^-$, and $Cl^-$ coat. During sunlit day time, chloride and water desorbs from Fe(III) as $°Cl$, $°OH$ and

$H_2O$, leaving Fe(II) in the varnish surface. The surface Fe(II) (and Mn(II)) is bound by oxygen bridges to the varnish bulk of Fe(III) (and Mn(IV)); may be like the combination of Fe(II) and

Fe(III) within magnetite. During night time the Fe(III) (and Mn(IV)) surface coat is regenerated by microbial and/or abiotic oxidation with $O_2$. It is worth mentioning, that desert varnish can exist only within dry regions.

Figure 4 illustrates the interactions of ISA at the phase borders of tropospheric aerosols, ocean surface, and dry solid surfaces.

[Figure]

**Figure 4**. Schematic representation of iron salt aerosols interactions with different solid surfaces:

Primary ISA precursor FeOOH particles (a) react with gaseous HCl by generation of

ISA as $FeCl_3$ coated on FeOOH particles (c).

Coagulation, condensation and chemical reaction with particles and vapors produce
different kinds of liquid and/or solid ISA variants and sediments:
(b) hydrolyzed $FeCl_3$ coated on soot and/or HULIS particles
(d) hydrolyzed $FeCl_3$ coated on ice crystals
(e) hydrolyzed $FeCl_3$ coated on salt crystals
(f) hydrolyzed $FeCl_3$ coated on ice crystals of snow layers (ISA sediment)
(g) hydrolyzed $FeCl_3$ dissolved in cloud droplets
(h) $FeCl_3$ hydrolysate residue on desert varnish (ISA sediment)
(j) hydrolyzed $FeCl_3$ as dissolved residue in ocean surface water fertilizes the
phytoplankton growth and at last triggers the generation of sulfuric, sulfonic and
dicarboxylic acids by emission of DMS, MC and further organics. This activates the
tropospheric generation of vaporous HCl by reaction of sea-salt aerosol (i) with the
acids. HCl again changes the ISA precursor FeOOH aerosol (a) to ISA (c).
Similar daytime dependent microbial activated abiotic photo-reduction and photo-oxidation
reaction cycles are known from aquifer environments [250]. Thus the $CH_4$ depletion of the
former ISA deposits will persist even after change into desert varnish. As explained chapter
2.2 continental HCl (300 pptv above the oceans and 100 pptv above the continents) [118],
$ClNO_2$ (up to 1500 pptv near flue gas emitters) [145, 146] and $CH_3Cl$ (550 pptv remote from
urban sources) [147, 148] and in deserts chloride salt containing dusts are direct and indirect
sources of chloride which could provide desert varnishes with $Cl^-$.

Furthermore, analogue to ISA deposited on solid desert surfaces, ISA depositions on dry
snow, snow cover and ice occurring in permanent snow-covered Mountain regions or within
polar and neighboring regions preserves its $CH_4$ destruction activity during sunlit day, spring,
and summer times [161].

The global area of the desert varnish surface does not change with changing dust
precipitation rates. It only depends on the precipitation frequency. It grows through
desertification and shrinks with increasing wet climate. Until now, quantitative measurements
about the specific amount of $CH_4$ depletion per square meter of desert varnish are not
known. Without this data, estimation about its influence on the $CH_4$ depletion and climate is
impossible.

The photochemical actions inducing $CH_4$ depletion of the desert varnish surfaces resulting
from dust precipitation are concurrent with the surfaces of deserts and semi deserts made of
sand or laterite soils. Their surface is colored by ochre to red iron oxide pigments. Their iron
components should act in principle by the same $CH_4$ depleting photochemistry such as ISA
and desert varnish.

As mentioned in chapter 2.4 the Cl and Br activation by iron photolysis changes after division
of the ingredients by freezing or drying of the former homogenous liquid between solid salt-
poor ice and liquid brine coat or solid salt and liquid brine coat. This inhomogeneous partition
phenomenon of the predominant transformation of aerosol droplets into solid, and vice versa, applies to snow or salt layers containing a proportion of ISA.

It has been shown that cooling precipitation of the buffering influence of salts such as carbonates, sulfates and chlorides of bromide and chloride rich mother liquors on arctic snow packs or ice particles can minimize their buffering capacity against pH change [160, 231,

251]. Similar mechanisms may act when liquid aerosol particles become solid by drying.

Then, the uptake and contact over time of solid iron-bearing particles and airborne organic and inorganic acids and acid precursors on, or with, ice crystal surfaces may drop the pH of the former alkaline particle surface, into the reaction conditions of the bromide oxidation by iron(III) photo reduction.

According to Kim et al. (Kim et al., 2010) the photogeneration of Fe(III) oxides, proceeding slow at pH 3.5 in bulk solution, becomes significantly accelerated in polycrystalline arctic ice.

This effect is accompanied by an acceleration of the physical dissolution of the Fe(III) oxides by freezing ice [252, 253].

The contact of arctic snow layers with iron oxides is confirmed by Kim [252]. Dorfman [254]

found recent loess dust sedimentation rates in the Alaskan Arctic Burial Lake of 0.15 mm/a.

According to the research results from artificial iron doped salt pans [125] iron salt doped sea-salt aerosols [124] or sea-salt doped iron oxide aerosols or pure iron oxide aerosols in contact with gaseous HCl [127] chloride and bromide in sun-lit surfaces are oxidized to °Cl and °Br by photo-reduced Fe(III) if the pH of the reaction media is 3.5 or lower.

As known from the bromine explosions, they appear on acidified first-year tundra and first- year sea ice snow lit by sunlight [230]. According to Kim et al. and Dorfman et al. the year-old snow layers contain iron(III). This confirms, that sufficient reaction conditions exist to produce bromine explosions by oxidation of iron(III) photoreduction.

Continents have considerable areas where the out flowing water is drained into "endorheic"

water bodies and not into the oceans. Endorheic lakes have no outlets other than evaporation and thus dissolved salts and nutrients concentrate over time. Large surfaces of these basins are covered by salt crusts, salt marshes, salty soils, or salt lakes. Most of these areas are situated within desert or semi-desert areas [255]. These salt environments gain iron from precipitating dust or from iron containing brines they have precipitated from. As far as these environments become acidic they oxidize $CH_4$ by iron photolysis induced °Cl [125].

To summarize the climate-relevant action of ISA within the troposphere according to chapters

2.1-2.6: $CH_4$, VOC, $O_3$ and dark carbon aerosol plus cloud albedo, in sum, have a similar effect on the climate warming as $CO_2$. The ISA method will have significant reductions in

$CH_4$, VOC, $O_3$ are anticipated by the test results from Wittmer et al. [124-127] and significant reductions in dark carbon aerosol and significant increase in cloud albedo are anticipated by the literature cited. We found no arguments against these statements. This allows the conclusion that only within the troposphere the ISA method should have significant climate cooling effects.

**3. Oceanic natural cooling effects of the iron cycle**

**3.1. Biotic $CO_2$ conversion into organic and carbonate carbon**

Vegetation uses the oxidative power of organic metal compounds induced by photon
absorption, oxidizing water to oxygen and reducing $CO_2$ by organic carbon generation
(photosynthesis by chlorophyll, a green Mg-Porphyrin complex). This assimilation process is
retarded by prevailing iron deficiency in the oceans which retards the phytoplankton growth.
Meanwhile there is no doubt that ISA-containing dust precipitation fertilizes the phytoplankton
which in turn affects the climate [256].
ISA triggers the phytoplankton reproduction and increases the formation of organic carbon
from the GHG $CO_2$ [42]. The vast majority of the oxygen thus formed and only slightly water
soluble (11 mg $O_2$ $l^{-1}$) escapes into the atmosphere. In contrast, the organic carbon formed
remains completely in the ocean, forming the basis of the marine food and debris chain.
From the primary produced phytoplankton carbon only a small fraction arrives at the ocean
bottom as organic debris and becomes part of the sediment. Cartapanis et al. [257] and
Jaccard et al. [258] found direct evidence that during the glacial maxima, the accumulation
rate of organic carbon was consistently higher (50 %) than during inter-glacials. This resulted
from the high dust concentrations during the glacial maxima, fertilizing the phytoplankton with
ISA.
The build-up of Ca-carbonate shell and frame substances by the calcification process at the
ocean surface extracts additional $CO_2$-C from the troposphere. The bulk of calcification can
be attributed to corals, foraminifera and coccolithofores; the latter are believed to contribute
up to half of current oceanic $CaCO_3$ production [259].
Both carbon fixation processes increase the removal of the GHG $CO_2$ and thus contribute to
cool the troposphere. The Fe-fertilizing process worked during the ice ages, as the
evaluations of Antarctic ice cores show: the minimum $CO_2$ concentrations and temperatures
in the troposphere are connected to the high dust phases [165].
It has been discussed that the alkalinity loss by phytoplankton calcification and $CaCO_3$ loss
with phytoplankton debris from the ocean surface is said to produce calcium and alkalinity
deficit at the ocean surface [260, 261], producing additional acidification at the ocean surface
by $CO_2$ generation:

$$Ca(HCO_3)_2 \rightarrow CaCO_3 + H_2O + CO_2 \quad \text{(Eq. 3)}$$

At least in part, this acidification is compensated by assimilative generation of organic carbon
by $CO_2$ consumption. Both organic debris and $CaCO_3$ become part of the ocean sediment.
But if the organic debris is re-oxidized during its journey downwards, some acidification could result. Acidification could result too if more $CO_2$ is absorbed by the ocean, then is assimilated
and changed to organic debris. Sedimentation of organic debris and $CaCO_3$, increase both,
according to the ISA-induced phytoplankton productivity.
The increasing amount of $CaCO_3$ sedimentation within iron fertilized ocean regions had been
discussed by Salter [262]. In a sufficient mixed ocean, alkalinity loss at the surface is more
than compensated by the different sources of alkali and earth alkali cations at the ocean
bottom and through continental weathering: in the first place these are the mechanisms of
alkalinity generated by the ocean water reactions within the ocean sediments and their bed-
rock, the oceanic crust. The latter mechanisms are described in more detail in chapters 4.1 –
4.3. The convection of the primary oxic ocean bottom water through the ocean crust
generates alkalinity by reduction of sulfate, nitrate and hydrogen carbonate, by dissolution of
silicates by reduced humic acids and further by serpentinization of basalt and peridotite
silicates [263, 264]. The alkalinity extracted from the oceanic crust keeps mainly positioned in
the dark water layers of the ocean basins if the decreased THC is not able to elevate the
alkaline extract into the phytoplankton layer in sufficient quantities.
The THC activation by the ISA method is described in the chapters 4.1 – 4.3.
Sudden ISA-induced phytoplankton growth generates increased calcite-shell production. This
lowers the Ca-concentration at the ocean surface. Even if the vertical cycling is not fast
enough to compensate the Ca-loss at once, or after a small time lag, this does no harm to
the phytoplankton growth, because Ca is not essential to it. Just the opposite is true:
phytoplankton uses the calcification as a detoxification measure to get rid of calcium ions
from their bodies [265]. As a consequence of this effect only the relation between Ca
carbonate sequestration and organic carbon sequestration will decrease during the time lag.
By additional direct alkalinity production of the phytoplankton itself, at least parts of the
acidity production by the lime shell production may be compensated: ISA-controlled
phytoplankton growth induces an increased synthesis of organic sulfur and of chlorine
compounds [266], emitted as dimethylsulfide (DMS) and methyl chloride (MC) [267].
Synthesis of organic sulfur and halogen organics as precursors of the volatile DMS and MC
emission is realized by the phytoplankton, by reduction of sulfate to organic sulfides, and
oxidation of chloride to carbon chlorine compounds. This precursor synthesis excretes
equivalent $Na^+$ and/or $Ca^{2+}$ alkalinity, as $Na_2SO_4$ reduction/formation to DMS generates Na
alkalinity; NaCl oxidation/formation to MC also generates Na alkalinity: cations formerly
bound to $SO_4^{2-}$ or $Cl^-$ loose their anions, producing alkalinity. According to [268, 269] the
sulfur content of phytoplankton exclusively, exceeds the $Ca^{2+}$, $Mg^{2+}$, and $K^+$ alkaline load of
phytoplankton lost with the phytoplankton debris. Only half of the organic carbon assimilated
by phytoplankton derives from dissolved $CO_2$. The other half derives from the ocean water
$NaHCO_3$ anion content [270]. The chemical reduction (reduction of $HCO_3^-$ to organic C + $O_2$

by assimilation of $HCO_3^-$ anions) produces alkalinity as further compensation of the alkalinity loss by calcification. $NaHCO_3$ reduction/formation to organic carbon generates Na alkalinity.

The cation previously bound to $HCO_3^-$ loses its anion and produces alkalinity.

These considerations demonstrate that any of the proposed enhanced weathering measures to prevent ocean acidification by increasing the alkalinity [271] might not be necessary if the

ISA method is in action and keeps the vertical ocean mixing sufficiently active.

During the down-dripping of the very fine-shaped phytoplankton debris, bacterial oxidation, fish and further food chain links minimize the organic debris up to an order of magnitude

[272]. Even the dissolution of the small carbonate debris reduces the carbonate fraction until arriving at the sediment surface. In order to maximize the effect of the ISA method, within the main ISA precipitation regions, the oxidation and dissolution of the organic and carbonate phytoplankton debris during its dripping down through the ocean water column can be reduced. To reach this goal, we suggest farming fixed filter feeders such as mussels and oysters within the ISA precipitation region.

Mussels and oysters produce faeces and so called "Pseudo-faeces" in the shape of rather solid pellets. Compared to the time of sedimentation of the unconditioned phytoplankton debris, this expands the sedimentation time difference between excreted filter feeder faeces and the phytoplankton faeces pellets sedimentation on the ocean floor by an order of magnitude. Bivalve farming would significantly reduce the oxidative and solution loss of phytoplankton debris attack. Mussel and oyster farming are well-known practices which have been employed for long time as a measure to produce protein rich food. They have been proposed as an element of climate engineering [273, 274].

To further optimize the $CO_2$-C conversion to sediment-bound C the biomass of oysters and mussels including their shells and fixing systems might be periodically dumped into the sediment.

Additional floating supports such as coral habitats, sponges, sea lilies and sea anemones between the mussel supports might complete and again optimize the ISA precipitation areas.

The oceanic water deserts can be changed into productive ecosystems and protein sources for an increasing population by these measures, among others, for an optimized $CO_2$ fixation induced by ISA.

A further proposal in order to maximize the $CO_2$ fixation induced by ISA is our suggestion to integrate a solution to the plastic waste problem on the ocean surfaces into the ISA method.

About 5 to 13 million metric tons of solid plastic waste per year are entering the oceans [275].

Over the last years the plastic waste drifting on the ocean has developed into a huge problem for the oceanic ecosystems [276]. Plastic keeps sunlight away from phytoplankton, hampering it from effective growth. The plastic waste drifts with the ocean currents. It then collects within accumulation zones predicted by a global surface circulation model [277].

Most plastic-covered ocean surfaces are concentrated in central-oceanic regions with low iron content with predestination for applying the ISA method. Due to the trash, there would be a reduction in the ISA efficiency so we propose the integration of the plastic depletion problem into our ISA method: on both the side of and the outside of a container ship vessel, a specific technology can be installed: plastic trash collection, plastic trash sorting, plastic trash extrusion, plastic trash burning, ISA production and emission. The aforementioned processes are well known and need no description here. Trash or waste burning has the advantages of delivering an effective hot carrier gas with high buoyancy for uplift of ISA and for delivering HCl as co-catalyst of ISA. With the plastic extruder, most carrier parts of floating supports on the reef coral, sponge, and mussel habitats could be produced.

Beside the larger plastic fragments, the floating plastic fine debris with particle diameters in the µm range is a further problem [278]. Instead of doing the micro-trash separation by technical means, the mussel and oyster farming may clean away this ocean surface environmental problem. The floating micro-trash particles are collected by the bivalves and excreted as pseudo-faeces pellets and at last become part of the sediment layer at the ocean bottom.

Within the iron cycle, the photolytic driven oxidant production with iron participation may not be reduced to $°Cl$ and $°OH$ in the troposphere and $O_2$ by assimilation: When iron is cycled through the mantle at temperatures above 2500 K, Fe(III) is reduced to Fe(II) by release of $O_2$ [279]. This phenomenon may be driven by the blackbody radiation containing a great fraction of photons with wave length shorter than 2 µm at and above this temperature level.

**3.2. ISA activates the $O_2$ input to the deep ocean**

Ocean ecosystems are based on certain balances between oxidizing and reducing agents. As a result of the ISA-triggered additional input of organic carbon in the ISA emission region (i.e. the ISA precipitation region), as described in chapter 3.1, oxygen consumption by increasing organic debris precipitation could increase. The recent $O_2$ decline in some oceanic regions may result, at least in part, from the deposition of soluble iron deriving from flue gas pollution. Equally discussed in chapter 3.1 is the decrease of the oxidation efficiency within the water column by measures to increase the sinking velocity of the organic C containing debris. The increase of the sinking velocity of the organic C containing debris, is an effect that might completely compensate the oxygen loss by oxidation of the ISA-induced debris mass increase.

Recently, and without ISA influence, oxygen deficiency seems to develop in many parts of the ocean as described in the introduction. Oxygen deficiency is usually due to insufficient vertical water exchange owing to increased vertical density gradient rather than the result of increased phytoplankton production.

Oxygen deficiency (hypoxia) is found frequently between the oxic surface layer (the
oxygenated one) and the oxic deep water layer [4, 280]. Due to the climate warming, the
localities with a lack of oxygen seem to intensify and expand already today [5].

The deepest water layer of most ocean basins results from the Antarctic wintertime ocean
surface ice generation by fractionating sea water into salt-poor sea ice and salt-rich dense
brine. This results in the production of cold, high density oxic brines which sink to the bottom
of the south ocean. The cold high density oxic brines spread as a thin oxic bottom layer up to
the ocean basins north of the equator. The most recent severe climate warming, which
induced disturbance of the THC, is likely to have been activated by the increasing inflow of
the fresh melt water from Greenland into the North Atlantic. This inflow disturbs the down flow
of the Gulf Stream water [281]. According to the increased melt of the glaciers of the
Antarctic, the salt content of the ocean surface around Antarctica decreased. This effect
increased the ocean surface covered by sea ice [282]. This freezing of the salt-poor melt
water layer decreases the production of dense brines. This again decreases the down flow of
brine, reducing again the vertical components of the ocean currents.

Through the ISA induced cooling, the oxygen and $CO_2$ flux into the deep ocean basins will be
restored due to the input of the cold dense oxygen and $CO_2$ enriched polar surface water:
Reduced melt water production of the Greenlandic and Antarctic ice shields by falling surface
layer temperatures will restore and intensify the thermohaline circulation within the northern
polar regions, by increasing the amount of Gulf Stream dumped, and by producing the circum
Antarctic sea ice cover without melt water dilution, which induces the production of cold high
density brines sinking to the ocean basin bottoms [283, 284]. Figure 5 illustrates the ocean
basins vertical mixing circles.

[Figure]

AABW   Antarctic bottom water
NADW   North Atlantic deep water
       Preferred ISA precipitation region

1059 **Figure 5**. The motor of the Antarctic bottom water (AABW) current is the sea ice production
1060 of the Southern Ocean area bordering Antarctica. The North Atlantic Deep Water (NADW)
1061 current is driven by decreasing Gulf Stream temperature on its way north. Climate warming
1062 especially the faster temperature rise at higher latitudes shifts the region of the Gulf Stream
1063 down flow as NADW further to the north, as a result of the lowering $\Delta t$ between equatorial
1064 and polar surface water. This shift sets additional Greenlandic coast regions in contact with
1065 warm Gulf Stream water and the rising air temperatures, as further component of poor
1066 increasing amounts of fresh melt water on the ocean surface. The rising melt water volume
1067 and the further north flowing Gulf Stream, increase the contact region between Gulf Stream
1068 water with fresh melt water. This produces increasing amounts of original Gulf Stream water
1069 but too low in density to sink and to become part of NADW.
1070 Temperature rise at higher latitudes reduce the salt content of ocean surface water around
1071 Greenland and Antarctica, inducing reduced NADW and AABW volumes. According to the
1072 reduced down flow current volumes, the amounts of $CO_2$ and $O_2$ to the deep ocean basin are
1073 reduced as well as the vertical fertilizer transport from the ocean basin bottom, to the
1074 phytoplankton at the surface.

**3.3. Phytoplankton fertilizer extraction from ocean sediments and underlying crust**

The oceanic crust is composed of peridotites, basalts and serpentine rock and has a layer of sediment on top. Sediments and bed rock contain reductive and alkaline components extractable by sea water. The cause of the ocean water flow through the sediment layer and base rock is the temperature difference driven convection. Sediment compaction by gravity, subduction-induced compaction and subduction-induced hydroxyl mineral dehydration may be further reasons for water movement through the sediment layer at the ocean bottom.

Olivine is one of the main mineral components of oceanic crust rock layers below the sediment layer. Hauck [285] simulated the effects of the annual dissolution of 3 Gt olivine as a geoengineering climate cooling measure in the open ocean, with uniform distribution of bicarbonate, silicic acid and iron produced by the olivine dissolution. An additional aim of this work was the development of a neutralization measure against the increasing acidification of sea water. All the components of olivine: $SiO_2$, Fe(II) and Mg are phytoplankton fertilizers. They calculated that the iron-induced $CO_2$ removal saturates at on average ~1.1 PgC $yr^{-1}$ for an iron input rate of 2.3 Tg Fe $yr^{-1}$ (1% of the iron contained in 3 Pg olivine), while $CO_2$ sequestered by alkalinization is estimated to ~1.1 PgC $yr^{-1}$ and the effect of silicic acid represents a $CO_2$ removal of ~0.18 PgC $yr^{-1}$. This data represent the enormous potential of the ocean crust rock as source of phytoplankton fertilizer.

The flow of sea water through anoxic sediments and bed rock results in the reduction of its $SO_4^{2-}$ content, as well as extraction of the soluble fraction from the sediment such as Mn(II), Fe(II), $NH_4^+$ and $PO_4^{3-}$. The chemical and physical extraction processes are enhanced by the action of microbial attack at the border lines between oxic sea water and anoxic sediment parts within this huge aqueous system.

At suboxic conditions soluble Fe(II) and Mn(II) have optimum solubility or may be fixed as solid $Fe(II)_3(PO_4)_2$, $FeCO_3$, $MnCO_3$, $FeS_2$, $S^0$ and further Fe-S compounds [286-290].

Silicon is mobilized too, from the dissolution of silicates and $SiO_2$ at methanogenic conditions by complexation with reduced humic acid (HA) [286, 291]. In the reduced conditions, HA is characterized by catechol and further polyphenolic functions, which allows HA to complex with silicon [292-294] and with further metal cations.

Silicate dissolution mobilized $Ca^{2+}$, $Mg^{2+}$, $Ba^{2+}$, $Fe^{2+}$, $Na^+$, $K^+$. $Fe^{2+}$, $Mn^{2+}$ and $PO_4^{3-}$ precipitate more or less as sulfides, carbonates, within the sediment ($Fe(II)S_2$, $CaCO_3$, $MgCa(CO_3)_2$, $Fe(II)CO_3$, $Mn(II)CO_3$, $Fe(II)_3(PO_4)_2$), and within its suboxic surface ($BaSO_4$) or at its oxic surface ($SiO_2$, $Fe(III)OOH$, $Mn(IV)O_2$, clay minerals). The authigenic formed ferromanganese nodules [295] are formed by in situ microbial precipitation from sediment pore water, squeezed out to the seafloor on the sediment layer [296, 297]. Main components of the nodules are the phytoplankton fertilizer components: $SiO_2$, Fe- and Mn-oxides [297].

Having left the borderline between anoxic and suboxic near-surface sediment the HA catechols are changed by reversible oxidation into quinone or quinhydrone configurations by decay of the Si catechol complex. Like most of the chemical reactions within the sediment compartment, oxidation of the HA-Si complex is directed by microorganisms. The microorganisms involved use HA as external red-ox ferment [298-305]. After arrival of the pore water originating from the anoxic deeper sediment, or bed rock at the suboxic surface-near sediment layers, the oxidized HA releases $Si(OH)_4$ and, $NO_3^-$ produced by microbial $NH_4^+$ nitrification [306, 307]. Depending on the $Si(OH)_4$ concentration produced, this can trigger the precipitation of layered silicates such as smectites, glauconite, and celadonite as well as silica [308-313]. Similar to HA, the clay mineral formation within the sediment, and the usage of the red-ox potential of these authigenic minerals, are, at least in part, the result of microbial action [314, 315].

According to its chelating properties, HA generate soluble to neutral Fe complexes of high stability even at oxic and weak alkaline ocean water conditions. As iron and HA have identical sources, especially chemoclines, even faeces HA can act as shuttles between Fe sources and phytoplankton [91]. But within oxic ocean milieu they become depleted, at last like every organic C substance, by oxidation.

The deep ocean currents take up the pore water percolates out of the sediment, and considerable amounts of the dissolved, colloidal or suspended sediment originating elements, are THC-conveyed to the surface [316] and activate there the phytoplankton production again. This as well, triggers the $CO_2$-conversion to organic C resulting in cooling the troposphere according to chapter 3.1. Repeatedly it also cools the troposphere by increasing the DMS formation according to chapters 2.5 and 3.1.

**4. The main cooling effects induced by the iron cycle on the ocean crust**

**4.1. Carbon storage as authigenic carbonate in the ocean crust**

The mechanism described in this chapter has the highest influence on the climate, due to its carbon storage capacity which is greater than that of their sediment layer. The convective water flow through the huge alkaline ocean crust volume is estimated to about 20 - 540 x $10^3$ km³ yr$^{-1}$ [29]. The oceanic crust comprises the largest aquifer system of the Earth, with an estimated rock volume of 2,3 x $10^9$ km³, and a fluid volume of 2 % of the total ocean or ~$10^7$ km³ [20]. The system of the mid-ocean rifts (MOR) and subduction zones and the sector between these volcanic active regions are part of the Earth Mantle convection cycle, and part of said interconnected aquifer system. The bottom water of the ocean basins are in close contact to this conveyor belt-like moving rock layer of the oceanic crust. New oceanic crust is produced at the MOR: during its cooling it is pulled apart from the MOR by the moving underlying mantel and, at last the moving mantle draws the crust down into the deeper
mantle below the subduction zones. The oceanic crust has a sediment layer on top of its
assemblage of multi-fractured crystalline and volcanic rocks. Both sediment and igneous bed
rock interior are in an anoxic reduced and alkaline state; temperature on top of the sediment
surface at the ocean bottom is round about 0 °C but temperature increases up to >1000 °C
within the igneous bedrock basement. As there is no effective sealing between cold bottom
water and high temperature zone, the water content of sediments and fractured basement
flows through the crust in multiple thermal convection cycles positioned between cold surface
and hot deep.

Alkalinity and alkalinity-inducing compounds of the ocean crust rock layers extract $CO_2$ and
$HCO_3^-$ from sea water by carbonate precipitation in the fissures during sea water percolation
through the multi-fractured rock [317]. A carbon uptake of 22 to 29 Mt C $yr^{-1}$ is estimated
during the hydrothermal alteration of the oceanic crust [318]. This is more than the carbon
uptake by the overlying sediment layer of the oceanic crust which is estimated to 13 to
23 Mt C $yr^{-1}$ [318]. The oceanic crust is composed of peridotites, basalts and serpentine rock
with a sediment layer on top. Said rock layers contain reductive and alkaline components.
Sea water circling through these rock layers loses its contents of oxygen, sulfate, nitrate and
even parts of hydrogen carbonate by reduction and precipitation, and becomes enriched with
methane and further reductants [319-326].

Figures 6A and 6B illustrate respectively the differences between a poorly and a sufficiently
mixed ocean.

[Figure]

[Figure]

**Figures 6A and 6B.** present the essential differences between an unstratified well-mixed ocean basin under a cold and dusty atmosphere during the cold main glacial, with low atmospheric GHGs concentration (6A) and a stratified ocean basin with a melt water layer on top of a saline ocean water layer during a warm interglacial, with a hot and dust-free greenhouse atmosphere (6B).

**Figure 6A:** According to the unstratified well mixed water column in Basin 6A $CO_2$ and $O_2$ absorbed at the water surface are distributed within all parts of the basin. High production rates of organic carbon produced by phytoplankton in the top layer are oxidized during their way down on the sediment layer, with only minor generation of organic sediment. Carbonate carbon produced by the phytoplankton becomes dissolved to great parts within the deeper basin parts generating $HCO_3^-$. $CO_2$ and $HCO_3^-$. By cycling of the basin bottom water through the alkaline bottom sediment and ocean crust aquifer, $CO_2$ and $HCO_3^-$ become precipitated and buried as carbonate C. The recycled bottom water becomes enriched by Fe fixed to organic chelators and is transported back to the surface. Due to the unrestricted down-flow and transfer of the $CO_2$ from the former surface water into sediments and into underlying base rock as carbonate carbon, the buried carbonate C exceeds the buried organic C amount.

**Figure 6B:** An interglacial episode with high GHGs levels accompanied by elevated surface temperatures generates increased melt water and surface water runoff. Because the saline poor water layer spreads on the saline ocean water and induces at least a regional stratification of the ocean basins water column: this stops the production of brine-induced surface water down-flow, as melt water freezing generates neither brine nor any vertical surface water movement. This stops any down transport of absorbed $CO_2$ and $O_2$ too and generates anoxic conditions within the underlying saline layer. The anoxic saline layer becomes anoxic and alkaline by sulfate and nitrate reduction. Any phytoplankton-induced organic and carbonate litter trickles down through the anoxic and alkaline layer: Ca- and $MgCO_3$ without dilution in the alkaline water and organic C without oxidation in the anoxic milieu. At the chemocline between light acidic $CO_2$ saturated water and the alkaline saline layer may precipitate Ca- and $MgCO_3$ in small amounts and mix with the down-falling phytoplankton-originating litter.

Due to the opposing chemical milieu differences between the oxic ocean water inflow and anoxic reduced and alkaline sediment and basement, the ocean water convection cycles through the ocean crust act as continuous chemical reaction systems and forms habitats of intensive acting microbial action [327]. The most intensive chemical reaction intensity is found at MOR, subduction zones and at volcanic sea mounts, between MOR and subduction within the abyssal plain convection cycling occurs [20]. Because the hydrogen carbonate load of the ocean water inflow comes to precipitation as carbonates of Ca, Mg, Fe, and Mn within the alkaline rock interior and by chemical reduction of sulfate, nitrate and hydrogen carbonate, the ocean basements act as huge $CO_2$-Carbon storages. No doubt: the ocean crust carbonate depot is the most effective carbon storage, more effective than any other organic carbon storages.

Within the huge ocean crust contact volume, sea water changes the alkaline pyroxenes and basalts into serpentine, diabase and carbonates; by producing heat, hydrogen, rock volume expansion and by permanent production of numerous fissures. The ocean water sulfates react with the silicate components to magnetite, pyrite and barite. The sea waters hydrogen carbonate load precipitates within the rock fissures as magnesite, calcite, siderite and dolomite. By heat transfer from hot rock and chemical reaction, heat circling through the primary and new generated multiple fissures in the former mantle rock, the sea water inflow heats up, producing convective flow. At fissures where the alkalized flow of convection water containing hot $CH_4$ and $H_2$ comes out with pH 9 to 11 and, contacts the fresh sea water, carbonate precipitates and builds up skyscraper high carbonate chimneys [328].

The convective seawater flowing only through the MOR system is estimated to about 20 to 540 x $10^3$ $km^3$ yr-1 [29]. This volume is more than the global river flow of about 50 $km^3$ yr-1 [329].

The weathering reaction conditions and the sea water alkalization during the intense sea water contact with the alkaline MOR rocks are much more aggressive, so respectively more effective, comparatively to reaction conditions and alkalization, during the precipitation water contact, during weathering reactions of continental rocks. This is confirmed by the alkaline pH of up to 11 of the "White Smoker" MOR outflow in spite of its haline salt buffered seawater origin [328]. Even the most alkali run-off from limestone karst spring fresh-waters or within karst cave fresh-waters does not exceed pH levels of 8.5 [330-332]. According to the enormous carbonate absorption capacity of the oceanic crust, it has been proposed to use it as a storage of $CO_2$ [333]. As the igneous crust rock aquifer generates $H_2$ during its contact with ocean water parts of the carbonate precipitation, carbonate is reduced in part to organic and / or graphitic C, depending on the reaction temperatures by botic or abiotic reduction [334-338].

There is no doubt that the efficiency of the pH dependent $CO_2$ absorption and carbonic acid neutralizing at the ocean surfaces and the hydrogen carbonate precipitation to carbonate processes at and within the oceanic crust, are dependent on the activity of the THC within the ocean basins. During cold climate epochs, with unstratified water column and undisturbed THC, the $CO_2$ conversion to ocean crust carbonate is activated, as well as the $CO_2$ conversion to the organic fraction of ocean sediments is activated. Just the opposite has been found to be true for the burial of organic C in ocean basin bottom sediments: according to Lopes et al. [96] the overwhelming organic debris fraction produced during main glacial episodes from the phytoplankton habitat at the surface, is oxidized and re-mineralized in the well-mixed ocean basin Lopes et al. [96]. As the $CO_2$ level in the atmosphere is at the lowest levels during the main glacials, the remaining C-sinks of the oceans seem to be of much bigger efficiency than the iron-induced production of organic C by assimilation: The most prominent C sink is the authigenic carbonate C burial in the alkaline ocean crust. There seems to be no doubt that the vertical well-mixed ocean during the main glacials works as an efficient pump, to transport dissolved $CO_2$ and $O_2$ to the ocean basin bottoms: There, $O_2$ act as mineralizer of organic C and $CO_2$-C is buried as authigenic carbonate C in the oceanic crust.

Table 2 gives an overview about some trends in C burial depending on the climate condition change between main glacial and interglacial.

**Table 2:** Interglacial climate episodes where hot, nearly dust-free, and had elevated levels of
GHGs. The interglacials coincided with stratified water columns. The stratified ocean has a
much reduced activity due to the reduced $CO_2$ transport to the bottom of the ocean basin. As
the $O_2$ transport is reduced, and the lower part of the basin is anoxic, the oxidative
mineralization of the organic litter fall from the phytoplankton activity at the surface is
reduced and generates sediments rich in organic substances. As sulfate, nitrate and in part
$CO_2$ within the anoxic water column are reduced to sulfide, ammonium and $CH_4$, the pH
increases to alkaline. This can induce carbonate precipitation near the chemocline. During
the glacial maxima with cold temperatures, dustiness and low greenhouse gas levels the
ocean basins had well and vertical mixed water columns with highest carbonate C burial and
lowest organic C burial.

| Effect on | | Sediment + crust below well and vertical mixed water column | Sediment + crust below stratified and anoxic water column |
|---|---|---|---|
| Mass ratio of buried sediment & crust carbon | sediment C / oceanic crust C | <<1 | <1 to 1 or >1 |
| Mass ratio of buried sediment & crust carbon | organic C / carbonate C | <<1 | up to 1 or >1 |
| Authigenic carbonate produced within the water column | | No | Yes |
| Tropospheric parameters | Dust | High | Low |
| | $CO_2$ | Low | High |
| | $CH_4$ | Low | High |
| | Temperature | Cold | Warm |

Lopes et al. [96] found just the opposite, in ocean sediment layers produced during the warm interstadial, in comparison to the cold main glacial: high burial rate of organic C in the ocean bottom sediment. But in spite of the high organic C burial rate, the interstadial $CO_2$ levels where kept higher than those of the main glacial. Even to this point the Lopes et al. [96]

results fits well to our $CO_2$ sink model. During the glacials climate warming events, enormous melt water volumes were generated and induced stratification effects in ocean basins by placing a melt water blanket on the saline ocean water surface [14]. The transport of $CO_2$

and $O_2$ into the basin bottoms became interrupted. The drizzle of phytoplankton litter kept un- oxidized, and as further consequence the amount of Carbonate C burial within the ocean crust ceased.

The continuous availability of chemical activity, as chemical reaction vessel and as an
alkalinity reservoir of the oceanic crust, is maintained by the continuous generation of new
crustal rock material of 21 km³ yr$^{-1}$ [20]. This huge rock volume production capacity has
enough alkalinity and fertilizer reserves to maintain the absorption, neutralization and
precipitation of a multiple of the recent incoming $CO_2$ and $HCO_3^-$.

THC is the main transport medium of carbon from the atmosphere into the deep on Earth.
This makes THC the most prominent climate stabilization element.

The realization of the significance of THC as stabilization element of our recent climate
model induces questions about the stability of the THC. As stated in chapter 1, the main
factors for destabilizing the THC seems to be stratification of the water column by the
desalting of surface ocean layers by freshwater dilution from increasing ice melting [6]. The
low density melt water generates a layer onto the ocean water, producing a stratified water
column. The stratification hampers or prevents the transport of $CO_2$ and $O_2$-containing
surface water into the deep ocean basin parts. The most severe consequence of such
stratification, to oceanic ecosystems, is the development of anoxic milieu within the stratified
ocean basins.

Typical marks of episodes with stratified water columns in ocean basins are the black shales
and black limestones as sapropel remnants. Repeated development of stratified ocean
basins during the Phanerozoic epoch occurred as a consequence of elevated $CO_2$ levels in
the atmosphere. This caused high sea surface temperatures [13], and as a global
consequence: global increase of evaporation, precipitation and as well production of brines
of higher concentrations.

Hansen [6] pointed out too, that the increasing melt water run-off from polar and subpolar ice
layers can induce the cover of denser ocean water by a melt water layer. But the generation
of increasing precipitation and surface water run-off accompanied by increasing brine
production during hot $CO_2$-high climate episodes has just the same consequences in the past
geological epochs as we learn from Meyers [13].

Just that we now have to fear this combination, of both the $CO_2$-dependent temperature rise-
generated precipitation increase, plus the melt water increase from glacier melt. Mankind has
to find now the appropriate tool to win or to fail this challenge.

A melt increase might drive the destabilization of THC. And at first the top layers of the ocean
basins will suffer from acidification and the deep layers will become alkaline and anoxic.

By starting the ISA process, the induced climate cooling will decrease the Greenland glacier
melt. The minimized freshwater inflow to the North Atlantic Ocean reduces the dilution of the
salty Gulf Stream and increases the down flow quantity of oxic and $CO_2$ containing salty
surface water. In parallel, the surface increase of sea-ice produced on the South Ocean
surrounding the Antarctic continent is followed by increased down-flow of oxic and $CO_2$

containing cold brine onto the bottoms of the oceanic basins. Both effects do increase the

THC activation: the flow of alkaline, phytoplankton fertilizer enriched, and oxygen depleted deep-ocean water to the surface. This activates $CO_2$ absorption from the atmosphere by phytoplankton growth and by $CO_2$ absorption

One of the proposed alternative climate engineering measures aims to absorb atmospheric

$CO_2$ by reducing the surface ocean acidity and by producing phytoplankton fertilizers. To transfer $1.1 \times 10^9$ t $yr^{-1}$ $CO_2$ carbon into the ocean a crushing of $3 \times 10^9$ t $yr^{-1}$ of the ocean crust and mantel rock mineral olivine to a particle diameter of 1 μm and suspend it at the ocean surface would be necessary [285, 339, 340]. These numbers seem to be two orders of magnitude too high. Keleman & Manning calculate a carbon mass subduction of about $50 \times$

$10^6$ t C $yr^{-1}$ (C in oceanic crust, bedrock and sediment layer) [318]. Independently of which of both calculations has a mistake – technical activities to do the Hauck et al. proposal are far from any economic reality.

The proposed reaction of $CO_2$ with olivine is done with much better effectiveness by nature, without any costs, within the ocean crust in sufficient quantity. To minimize $CO_2$ emission it has been proposed to minimize power stations flue gas $CO_2$ by absorption by lime suspension [341]. This measure seems to be unnecessary when the ISA method comes into practice.

The fertilizing elements the phytoplankton needs, such as Si, P, and Fe, are all present in the ocean crust [342] and a property of the ocean crust water extract. Intensification of the THC

would also increase the fertilizer concentration at the ocean surface in the phytoplankton layer. As demonstrated, the undisturbed THC is essential to keep the climate stabilized [32].

The ocean crust from the warm Mesocoic epoch which had no frozen polar regions contained about five times more authigenic carbonate than ocean crust younger than 60

million years [32]. Coogan interpreted this as possible consequences of higher bottom water temperature and/or different seawater composition. Insua et al. [343] found evidence, that the salinity of the ocean bottom water during the Last Glacial Maximum had been up to 4 %

greater than today. It seems evident that the cause of the latter had been the higher volume of brine produced during sea-ice freezing. This fact demonstrates that disturbed or weakened

THCs might be the cause of reduced carbonate C uptake of the ocean crust. The quantity of carbonate precipitation depends on the $CO_2$ and/or $HCO_3^-$ input with seawater. As a consequence, the quantity of the ocean crust $CO_3$ uptake varies according to the activities of the THCs or stratified ocean basins: strong THCs increase the crust carbon content; weak

THCs decrease it.

Independently of the cause of stratification events: by brine generation, by freezing or by evaporation, the ocean basins possess a removal mechanism which extracts salt from the brine and change the brine to sea water of normal salt concentration. This mechanism has kept the salt concentration of sea water rather constant during the past geological epochs. This effect to achieve a constant salinity level, depletes any brine-induced stratification and restores well-mixed ocean basins again.

According to Hovland et al. [344-346] this desalination takes place by continuous salt removal from the brine or seawater within the hot ocean crust. This desalination works independently of the salt concentration of brine or seawater. The salt removal process acts within the ocean crust aquifer at near critical to super-critical seawater temperature and pressure conditions. During subduction of the salty crust rock chloride and carbonate change their cations with silicate and are dissolved as HCl and $CO_2$. Accompanied by $H_2O$, these gases are recycled to the atmosphere, mainly by subduction volcanism, but at a much smaller amount by MOR and similar alkaline volcanism.

During the time lag between the onsets of the ISA method cooling and the appearance of the alkalinity and fertilizer increase at the ocean surface, the cooling effect of ISA remains reduced. But after this time lag, the ISA method increases to optimal efficiency. Even from an economic viewpoint it seems better to compensate this by increasing the ISA emission at the beginning during the time lag, than doing the proposed suspending of olivine dust at the ocean's surface. Even lime shell wearing phytoplankton is able to accept small pH changes of $CO_2$ induced dependent acidification, because it uses the build-up of calcium carbonate shells as a detoxification measure to get rid of calcium ions from their bodies [265]. As a consequence of this effect, only the relation between Ca carbonate sequestration and organic carbon sequestration may decrease during the time lag.

Summing up: through the huge aquifers of the alkaline and reducing ocean crust, any transport of former surface water enriched by $CO_2$ or $HCO_3^-$ induces carbonate C burial within the aquifer interior. This is the situation within well-mixed Ocean basins without stratification. Any stratification decreases carbonate burial or even stops it. Stratification changes the red-ox milieu below the stratification-induced chemocline. The MOR and sediment-induced exhalation of Fe and further metals by the black smokers into the sulfidic stratified ocean basin are prevented from contact with the planktonic surface water habitat. But surface water runoff, as well as melt water inflow and iceberg melt during warm glacial climate intervals may compensate the lack of Fe from the MOR and bottom sediment sources, as well as from the decreasing dust fall during the warm climate intervals [6, 7].

**4.2. Carbon storage as organic and inorganic marine debris and as authigenic carbonate in the ocean sediment**

The uptake of authigenic hydrogen carbonate from the ocean and precipitating it in the sediment, seems to play as well a major role in the carbon circle [347]. According to Kelemen [318] the carbon uptake by the sediment layer of the oceanic crust can be estimated to 13 to

Mt C yr$^{-1}$. The carbon inventory consists of life and dead organic carbon, carbonate carbon and authigenic carbonate produced by excess alkalinity deriving mainly from sulfate reduction and silicate solution by reduced humic acids. According to Sun & Turchyn the formation of calcium carbonate and its burial in marine sediments accounts for about 80 % of the total carbon removed from the Earth surface [348]. Meanwhile it seems possible to distinguish between marine formed sediment carbonate and authigenic carbonate [349].

As evidenced in chapter 4.1, stratified ocean basins can differ widely in quantity and quality of the buried C according to the prevailing climate conditions and their direct and indirect influences on ocean basin conditions. Table 2 lists some of the most prominent results.

The cooling of the Troposphere by ISA action stops melt water inflow, destructs the stratification and starts the vertical mixture. During the former stratification event, alkalized deep water layer had enormous $CO_2$ absorption capacity. The alkalized anoxic sediment behaves in a similar manner. This makes a much increased $CO_2$ absorption activity at the beginning of the movement.

Accordingly, excess alkalinity is produced by dissolution of silicates such as illite, kaolinite and feldspars, volcanic ash, pyroxene or other silicate components of ocean sediments and even opal by Si complexation with reduced HA at methanogenic conditions [286, 289, 350, 351]. Compensation by hydrogen carbonate induces authigenic precipitation of microbial dolomite [352], Ca or Fe carbonate [286, 291, 348, 350, 353, 354] and further minerals [355]. As mentioned in chapter 4.1, the biological processes of chemical sediment reduction induced by the ISA fertilization, changes $NO_3^-$, $SO_4^{2-}$, Fe(III), Mn(III/IV) and $HCO_3^-$ to their deoxygenated and reduced species, inclusive $CH_4$ and $NH_4^+$ generation, produces a pH increase and additional alkalinity. Further pH drop is induced by $H_2$ evolution from $FeS_2$ generation from FeS and $H_2S$ [356, 357] accompanied by $CO_2$ reduction to $CH_4$ [358] as well as $N_2$ reduction to $NH_3$ [359]. The alkalinity excess converts dissolved $HCO_3^-$ into solid lime and dolomite [360-363]. The solid carbonates and $CH_4$ hydrate stabilize the sediment. Outside the polar permafrost region, methane hydrates are stable below 300 m below sea level and at ocean temperatures of nearly 0 °C [364]. The carbonate precipitation sequesters additional parts of $CO_2$, prevents the ocean water from acidifying and at last improves the $CO_2$ absorption by ocean water from the atmosphere. This again cools the troposphere.

The enhanced dissolution of silicates from the ISA induced by methanogenic sedimentation additionally compensates the enhanced alkalinity loss at the ocean surface, attributed to the calcification due to foraminifera and coccolithofores phytoplankton growth by ISA fertilization.

Summing up: within a well-mixed and unstratified ocean basin the surface layer absorb $CO_2$ and $O_2$ and become well mixed into the unstratified ocean basin by the thermo-haline basin convection. Consequences of the good mixture are nearly quantitative oxidation of the food chain debris to $CO_2$ produced by phytoplankton. Most C is buried as carbonate in the ocean crust and its overlying sediment. The ratio of organic C burial to carbonate C burial is much smaller than 1. Results of Lopes et al. [96] from Northeast Pacific sediments demonstrate that, although estimated highest primary productivity during the Last Glacial Maximum, organic C burial was lowest. This coincides with our proposed optimum mixed $O_2$-rich milieu throughout the whole water column.

During situations with stratified water columns in the ocean basins or parts of them the THC convection is disturbed or does not exist at all. Surface water layer enriched with $CO_2$ and $O_2$ absorbed from the atmosphere cannot penetrate through the stratified water column, into the bottom of the basin. This induces sulfate reducing conditions below the surface layer. Only small parts of surface layer $CO_2$ are changed into carbonate C at the chemocline, with the alkaline sulfidic and anoxic parts below the chemocline. Below the chemocline, the water column is anoxic, the organic debris sediment with minor oxidation. Probably the ratio of organic C burial to carbonate C burial increases to a manifold during stratified conditions. Concerning to the huge fraction of organic C buried during the warm glacial intervals, according to the results of Lopes et al, [96] from Northeast Pacific, sediments demonstrate stratification events within their research area.

Stratification events may develop by warming the upper water layer, as well as by evaporation and precipitation [6-8].

**4.3. Minimizing CH$_4$ emissions from sediments and igneous bedrock**

The reaction product of oceanic crust minerals containing Fe(II) such as Olivine and Pyrrhotite with sea-water is hydrogen [365-367]. The hydrogen production rate at least along the MOR alone is estimated to ~$10^{12}$ mol $H_2$ yr$^{-1}$ [368]. Hydrogen is fermented by microbes with hydrogen carbonate into methane. The latter is known as constituent of the springs emitted by the ocean crust rocks (Früh-Green 2004).

Such and further $CH_4$ emissions, such as anoxic sediments outside the $CH_4$ hydrate stable pressure and temperature region, induce de-oxygenation within the overlying water layer by $CH_4$ emission [17, 369]. $CH_4$ emissions are induced for instance by hydrothermal springs [370], sediment movement [371, 372], seawater warming induced by climate change [373, 374], changing ocean circulation [375], ocean sediment subduction [376, 377]. At lower vertical sediment to ocean surface distances, the $CH_4$ emissions reach the troposphere. As the Arctic Ocean suffers at most from the climate change induced warming, the $CH_4$ release within this region rises extraordinary [16]. The most elevated Global surface-near oceanic $CH_4$ concentrations are located within the Arctic Ocean and the arctic troposphere [378]. This might be one of the reasons for the higher temperature rise of the Arctic region than the average surface Earth warming.

Within the sediment and within the suboxic ocean water column, $CH_4$ is oxidized by sulfate.
Iron is an accelerator of this microbial fermentation reaction [379]. The ocean water column
and the underlying sediment having had contact with ISA-originating iron are elevated in their
iron content. This has different cooling effects to the troposphere: at first the elevated iron
content in the uppermost suboxic sediment reduces the $CH_4$ content emitted by the sediment
by anaerobic oxidation of methane by sulfate-reducing bacteria.

Below regions with ISA precipitation, not only the sediment, but even the whole water column
of the ocean basin is enriched on iron. Any $CH_4$ molecule, independently of existent in the
sediment, or just above in the water phase, or excreted into the water column as bubbles, is
oxidized before it arrives at the water column top. By help of Fe containing enzymes the
methane oxidation by sulfate is possible. This prevents the water layers above the sulfate
oxidation zone from oxygen loss. Sulfate oxidizers of $CH_4$ are archaea and bacteria [380]. As
these microbes use Fe-containing enzymes to do their anaerobic methane oxidation
processes, they act better in iron-rich than in iron-poor environments [381, 382]. The iron
containing debris fall of ISA-fed dead phytoplankton and phytoplankton dependent food chain
links, feeds the methane depleting sulfate reducer community within or near the sediment
surface.

Next, the iron content reduces the $CH_4$ bubble-development within the sediment layer,
preventing catastrophic $CH_4$ eruptions by sediment destabilization, $CH_4$ bursts and sediment
avalanches.

Third: elevated iron content prevents the ocean water column from $CH_4$-induced oxygen
deficiency by the formation of ammonium. This oxygen deficiency prevention protects from
generation of the extreme stable and very effective GHG $N_2O$ [383].

The oxygen-dependent life will become problematic, due to its decreasing oxygen content
within a decreased vertical mixed ocean basin induced by climate warming. An additional
input of $CH_4$ would increase the oxygen deficit death zones. Any $CH_4$ injection into regional
oxygen deficit zones, will immediately increase their volume. Climate models predict declines
in oceanic dissolved oxygen with global warming. The climate warming dependent decline of
the oxygen content in many ocean regions has meanwhile become manifest [384]. Braking
or reversal of this trend by reducing the oxygen depleting $CH_4$ emissions at least should help
to prevent regions within the ocean basins from methane-induced oxygen deficit.

The glacial age proved that in spite of the multiplicity of the cooling processes induced, they
caused little disturbance to the ecosystems. This predestines ISA as a steering tool to
prevent climate fluctuations such as the recent climate warming mankind is suffering from.
The present study aims to describe in chapter 5 the technical means to realize this climate
engineering project by the ISA method.

This result is contradictious to the calculations of Duprat et al. [385]. They found within the iron containing melt water trail of the giant Antarctica icebergs increased phytoplankton concentration. Duprat et al. assume that the iceberg induced carbon export increase by a factor of 5 to 10 within its influence locality and they expect an increase in carbon export by the expected increase of the iceberg production that has been predicted (for instance Joughin et al. [386] ). We interpret the ongoing increase of icebergs and ice melt as a further severe warning sign that the ongoing destabilization might end soon in an insufficient mixed ocean.

The only artificially realizable restoration tool to change an insufficiently or poorly mixed ocean into a well-mixed ocean is definitely by climate cooling. The ISA method appears to be the climate cooling method by means of choice, because it accelerates the conversion of atmospheric carbons into solid and even liquid carbons with the means of nature. Comparing to the artificial aerosol systems based on $TiO_2$ or $H_2SO_4$ [387], the sea-salt aerosol has advantages, such as better controllability and economy.

**5. Iron effects onshore**

**5.1. Importance of iron on terrestrial landscapes**

As seen in previous sections, atmospheric deposition of iron together with other macronutrients and micronutrients set important controls on marine ecology and biogeochemistry: for terrestrial ecology and biogeochemistry the importance of iron is similar. Iron is one of 17 essential elements for plant growth and reproduction [388]. Iron is an essential micronutrient (or trace element) only required by plants in small amounts, for bio-functions such as production of chlorophyll and photosynthesis [389]. Iron is involved in many other important physiological processes such as nitrogen fixation and nitrate reduction and is required for certain enzyme functions [390].

Iron is the 4th most abundant element of the earth's crust (4.2%) and thus iron is seldom deficient, as despite its high abundance in soil, iron solubility is extremely low and its availability depends of the whole soil system and chemistry. Chlorosis (yellowing) is associated with iron deficiency in plants over land [59, 61], but the chemistry of iron in soils and its availability to plants [60] is out of the scope of this review, thus only a brief overview is given. However, while small amounts are necessary for growth, iron can become toxic to plants. Iron toxicity is associated with large concentrations of $Fe^{2+}$ in the soil solution [391] and leads to oxidative stress. As a consequence, iron-uptake systems are carefully regulated to ensure that iron homeostasis is maintained. Iron availability represents a significant constraint to plant growth and plants have developed distinct strategies to ensure Fe solubilization and uptake [392]. In forests, microorganisms such as fungi and bacteria, play a role in nutrient cycling [393]. A particularly efficient iron acquisition system involves the solubilization of iron by siderophores [394], which are biogenic chelators with high affinity and specificity for iron complexation.

Iron deficiency induced chlorosis represents the main nutritional disorder in fruit tree orchards and in crops grown on calcareous and/or alkaline soils [395] in many areas of the world. Iron deficiency is a worldwide problem has calcareous soils cover over 30% of the earth's land surface [396] specially in arid and semi-arid regions and has a large economical impact, because crop quality and yield can be severely compromised [397, 398], thus several methods of correction have been developed. Iron canopy fertilization (foliar fertilization) can be a cheaper, more environmentally-friendly alternative to soil treatments with synthetic Fe(III) chelates for the control of Fe chlorosis in fruit trees [399]. But iron chelates are expensive and have to be applied annually. Several sprays aiming to activate the Fe pools in a chlorotic leaf by foliar iron fertilization have been tested and were generally as effective as simple spay fertilization with iron sulphate (Abadía et al., 2000) and both are effective in re-greening treated leaf areas, both in peach trees and sugar beet plants [397]. Iron-deficiency chlorosis in soybean was solved by foliar sprays which significantly increased the yield of three cultivars tested and the yield responses obtained, were about 300 kg ha$^{-1}$ [400].

Although foliar Fe fertilization seems to be potentially effective, the scientific background for this practice is still scarce and we did not found evidence that soluble iron contained in atmospheric dust aerosols has already been proved to be able to play this role.

The fertilizing role of African dust in the Amazon rainforest is well known [401] but attributed to the P input. On a basis of the 7-year average of trans-Atlantic dust transportation, Yu [402] calculated that 182 Tg yr$^{-1}$ dust leaves the coast of North Africa (15°W), of which 43 Tg yr$^{-1}$ reaches America (75°W). The dust reaching the Caribbean and the Amazon come mainly of the northwestern Africa (Algeria, Mali, and Mauritania) [403].

An average of dust deposition into the Amazon Basin over 7 years is estimated to be 29 kg ha$^{-1}$ yr$^{-1}$ [401], providing about to 23 g ha$^{-1}$ yr$^{-1}$ of phosphorus to fertilize the Amazon rainforest, together with Mg and Fe. Although not directly related to ISA, this dust deposition allows biomass fertilization and thus $CO_2$ removal from the atmosphere.

The wide spread tropical soils, mostly laterites, are deficient in phosphate and nitrogen but not in autochthon iron. The only exception to this is for all the epiphyte plants and the plants growing on the soil-free localities without any autochthon iron. These plants might gain profit from the ISA method. Such plant communities are localized for instance on top of the famous Tepuis (table mountains north of the Amazon basin near the borderlines of Brazil, Venezuela and Guyana) and on the tree branches in the rain forests without roots into the ground. From Köhler et al. [404] the epiphytes flora on the tree branches of the rain forests may contain up to 16 t ha$^{-1}$ (Costa Rica) up to 44 t ha$^{-1}$ (Colombia) of epiphyte plant + humus dry weight on the tree branches.

The epiphytes, but much more the Tepui plants, would gain profit from ISA and even from undissolved iron oxides, because plant roots and fungal hyphae secrete iron-solubilizing organic acids and complexants. Microbial ferments have time enough to turn all kind of undissolvable Fe into dissolvable Fe.

Is there a climate relevance to rain forest fertilizing by dust? Rizzolo et al. [405] states that the iron limited Amazon rainforest profits from the seasonal deposition of iron by Saharan dust. Especially the deposition of iron plus further nutrients on the Amazon biota is likely to increase both epiphytic growth and fungal and bacterial decomposition within the canopy [405]. The increase in iron bioavailability is also known to increase nutrient cycling within the forest.

Large fractions of the organic biomass produced by help of iron and further eolic nutrients leave the Amazon region, are transported into the South Atlantic basin and at last become part of the shelf and basin sediments. This are aquatic life plants such as Water hyazinth and Water fern, plant litter such as driftwood, leaves, and particular, colloidal, and dissolved humic and fulvic acids. According to Ertel et al. [406] the flux of dissolved organic carbon fraction at Óbidos, situated about 800 km above the Amazon mouth, is $2 \times 10^{13}$ gC yr$^{-1}$.

Some rain forests such as the Amazonian, benefit from sporadic dust plume fertilization of Saharan origin. Others may profit from an artificial ISA precipitation resulting in a significant additional epiphyte plant growth.

**5.2. Importance of iron for human food and health**

All organisms on Earth ride upon a "*ferrous wheel*" made of different forms of iron that are essential for life [97]. Iron is an important micronutrient used by most organisms, including higher animals and human beings and is required for important cellular processes such as respiration, oxygen transport in the blood. Its bioavailability is of concern for all the Earth's living organisms, especially in aquatic ecosystems, including clear water and oceanic ones.

In humans, iron deficiency and anemia remain the most common nutritional disorders in the world today [407].

The World Health Organization [408] states that the lack of sufficient micro nutrients such as Fe and Zn, represents a major threat to the health and development of the world population. WHO [408] estimates that over 30 % of the world's population are anemic and even more in developing countries (every second pregnant woman and about 40% of preschool children). Iron deficiency affects more people than any other condition, and iron deficiency exacts its heaviest overall toll in terms of ill-health, premature death and lost earnings. Iron deficiency and anemia reduce the work capacity of individuals and of entire populations, causes maternal hemorrhage, impaired physical and cognitive development, reduced school performance and lowered productivity, bringing serious economic consequences and obstacles to national development.

Iron deficiency in humans has been associated with heart failure [409, 410]; gastric ulceration and anemia induced by Helicobacter pylori [411]; negative impacts on skeletal integrity [412], cognitive disorders [413]. Iron deficiency in infancy leads to long-term deficits in executive function and recognition memory [414]. In experiments with animals, even if the iron and the hemoglobin levels return to normal after treatment from an early induced iron deficiency, there are long-lasting cognitive, physiological and hematological effects [415]. Thus several strategies and technologies have been elaborated to manage iron deficiency in humans [416] such as food fortification (adding iron to food) [417] and biofortification (the process of enriching the nutrient content of crops, vegetables or fruit as they grow). WHO, FAO and UNICEF edit guidelines or recommendations on food fortification with micronutrients [418], for instance adding ferrous sulphate, ferrous fumarate, or iron complexes to wheat and maize flour (from 15 to 60 ppm depending on the regional average consumption ranges and on other iron food vehicles). Biofortification can be achieved by utilizing crop and soil management practices to increase micronutrient concentrations in the edible crop parts [419] and can provide a sustainable solution to malnutrition worldwide, as other methods, such as diversifying people's diets or providing dietary supplements, have proved impractical, especially in developing countries). Together with dietary modification and iron dietary supplementation, iron fortification (suitable food vehicle containing higher levels of bioavailable iron) are the main recommendations of WHO to increase iron intake, improve nutritional status and stop iron deficiency anemia. Increasing available iron levels in major staple food crops is an important strategy to reduce iron deficiency in people. WHO anticipates that benefits are substantial as timely treatment can restore personal health and raise national productivity levels by as much as 20%.

The biofortification of bioavailable iron in staple plants provides a sustainable and economical tool to use, in order to rescue iron deficiency in target populations globally [420]. In contrast with fruit trees, where foliar iron fertilization is generally used in chlorotic leaves, canopy, Fe-fertilization is increasingly being used in cereal crops to increase the Fe concentration in grains, in what is called biofortification. In these crops, which are generally treated with foliar iron sprays when there is no leaf chlorosis, applied iron has been shown to re-translocate efficiently to other plant organs, both in wheat [421] and rice [422]. Zuo and Zhang [419] have developed strategies to increase iron uptake by roots and transfer it to edible plant portions allowing absorption by humans from plant food sources.

**5.3. Active inhibition of methane emissions from wetlands, lakes, and sediments**

Lipson et al. [423] found that in Arctic peat ecosystem, Fe and humic reduction competes with methanogenesis as e- acceptors and inhibit some $CH_4$ production and that on the basis of conservative measurements of net Fe reduction rates, this process is comparable in magnitude to methanogenesis.

In wet sedge tundra landscapes Miller et al. [424] conducted experiments that showed an inverse relationship between dissolved iron and $CH_4$ concentrations and found that net $CH_4$ fluxes were significantly suppressed following the experimental addition of iron and humic acids. Iron and humic acid amendments significantly suppressed *in-situ* net methane flux.

Lipson et al. [425] conducted experiments on 2 different ecosystems: one with permafrost and naturally high levels of soil Fe and one with no permafrost and naturally low levels of soil Fe. The addition of Fe(III) and humic acids (electron acceptors) significantly reduced net $CH_4$ flux for at least several weeks post-treatment, without significantly altering $CO_2$ fluxes. There was no significant difference between the reduction of $CH_4$ flux caused by Fe(III) and the one caused by humic acids. The future release of GHGs from high latitude wetland ecosystems can significantly be altered by this natural and widespread phenomenon. These results also show that the suppression of $CH_4$ flux in this type of ecosystem can be induced by artificial addition of Fe(III), humic acids or other electron acceptors.

Zhang et al. [426, 427] found methanogenesis and sulfate reduction inhibition after ferric salt dosing to anaerobic sewer biofilms. Similar methanogenesis inhibition and even increases of rice productivity by ferric salt addition have been described by others [428-431].

Amos et al. [432] found support for the hypothesis that Fe(III) mediates $CH_4$ oxidation in crude contaminated aquifer.

Although some iron oxides such as magnetite and hematite have different properties and may facilitate methanogenesis by some types of micro-organisms [433] it is worth being noted that the iron solubility and bioavailability properties of the ISA are similar to the ferrihydrite which inhibits methanogenesis in the same experiments [433] and in general Fe(III)-reduction by methanogens contribute to Fe(III) inhibition of methanogenesis [434].

Experiments conducted in tropical humid tropical forest soils, which are also an important source of atmospheric $CH_4$ and where Fe(III)-reducing bacteria coexist with methanogens, show that upon addition of acetate, production increase of $CH_4$ is much greater (67 times) than that of $Fe^{2+}$ (2 times), indicating that the two process were acetate limited and suggesting that Fe(III)-reducing bacteria were suppressing methanogenesis when acetate availability is limited [435]. For Roden and Wetzel [436] a significant suppression of $CH_4$ production in freshwater wetlands could be mediated by Fe(III) oxide reduction within globally extensive iron-rich tropical and subtropical soil regimes.

All these results support the hypothesis, that additional to the many photolysis dominated $CH_4$-depletion actions by ISA in the troposphere, even after ISA precipitation on wetlands, marshes, lakes, rice paddies and shelf sediments it will inhibit the emission of $CH_4$. The degree to which Fe(III) reduction suppresses $CH_4$ emissions under different soil conditions should be considered by regional and global models of GHGs dynamics.

No published studies were found about the biogeochemical cycle of iron to the continents and land in specialized journals such as "Global Biogeochemical Cycles », nor in the chapter about the biogeochemical cycles of the latest IPCC report and, the recent Iron Model Intercomparison Project (FeMIP) seems concentrated in oceans interactions [55, 437].

It is now well known that in large areas of the open ocean iron is a key limiting nutrient and that in alkaline terrestrial landscapes iron deficiency induces plant chlorosis. The authors' hope is that bringing together under this review seemingly disparate lines of research from diverse disciplines, it will result a more global understanding of the global biogeochemical iron cycle, especially over terrestrial landscapes, peat-bogs, and other wetlands.

**6. Estimations of the ISA demand by the ISA method**

**6.1. ISA can induce a significant $CH_4$ depletion**

Wittmer [124-127] reported that the ISA method is very efficient for °Cl generation. Hence, ISA allows depletion of GHG methane by separation prior cooling effect. Therefore, ISA appears to be a very promising cooling method with technical and economic stakes. But the answer depends strongly on the volume of ISA to be produced and emitted. Indeed, ISA plume should be released high enough in the troposphere to get sufficient distribution and residence time in combination with °Cl generation quantity.

Based on results of Fe photolysis induced °Cl production, Wittmeret al. [124] estimated the feasibility of $CH_4$ depletion by NaCl-diluted ISA. Wittmer found a °Cl emission of $1.9 \times 10^5$ °Cl/cm$^3$ at a Cl$^-$/Fe(III) molar ratio of 101 within the pH range of 2.1-2.3. The same °Cl generation was found at the suboptimal pH of 3.3 – 3.5 and at a Cl$^-$/Fe(III) molar ratio of 51. This Cl generation is four times higher than the reference which corresponds to a significant $CH_4$ lifetime reduction in the troposphere [124]. A pH range of around 2 corresponds to the natural aerosol pH within the oceanic boundary layer. The optimum efficiency of °Cl production by photolysis of ISA corresponds to pH 2, whatever the source of Cl$^-$, NaCl or gaseous HCl and whatever if ISA is an iron(III) oxide or an iron(III) chloride aerosol [124].

According to Lim et al. [438] and to Meyer-Oeste [439] the optimum °Cl production by sunlight photolysis of $FeCl_3$ solutions or ISA, is generated in the acidic pH range. The efficient °Cl generation is necessary for an efficient $CH_4$ depletion by ISA. Except if made by condensation and hydrolysis of $FeCl_3$ vapor or by nebulization of pure $FeCl_3$ solution, or produced by combustion to pyrogenic FeOOH and reaction and hydrolysis with HCl and $H_2O$

to FeCl$_3$ solution: FeCl$_3$ has an acidic pH from the beginning because it hydrolyses according
to equation 4.

$$FeCl_3 + 2H_2O \rightarrow FeCl_2OH + H_3O^+ + Cl^- \qquad (Eq. 4)$$

**6.2. ISA demand calculation**

Current CH$_4$ depletion by $^\circ$Cl is estimated from 3.3% [440] to 4.3% [119]. According to the
results of Wittmer [124] at a Cl$^-$/Fe(III) molar ratio of 101, this amount would rise fourfold:
from 13 to 17%.

1.   Wittmer et al. used their results obtained at a Cl$^-$/Fe(III) ratio of 51 at the pH of
3.3-3.5: 1.9 x 10$^5$ $^\circ$Cl/cm$^3$. We consider that this pH is suboptimal. Instead it should be
used the results obtained at a Cl$^-$/Fe(III) ratio of 101 at the pH of 2.1-2.3: 1.9 x 10$^5$
$^\circ$Cl/cm$^3$.

Moreover, Wittmer et al. made two limitative estimations:

2.   They only focused on the Cl delivery in the condensed state by coagulation as
Cl$^-$ transfer option between ISA particles and the Cl source sea-salt aerosol ignoring
other Cl sources, Cl aggregate states, and Cl transfer mechanisms.

According to this model, the ISA particles should continuously lose in the daylight their Cl$^-$
load by $^\circ$Cl emission and as a consequence they could gain back Cl only by coagulation with
sea-salt aerosol particles. As further consequences of this model the Cl$^-$/Fe(III) ratio of ISA
particles would decrease, their diameter increase and their residence time in the troposphere
would decrease.
But according to Graedel and Keene [118] and Keene et al. [441] the next prominent source
of inorganic Cl in the troposphere beside sea-salt aerosol is vaporous HCl. This is the main
source where the ISA particles can refill the chloride lost by photolysis. The main Cl uptake
mechanism from this Cl source is the sorption from the gaseous phase.
Main HCl sources are the sea-salt reaction with acids, CH$_4$ and further hydrocarbon reactions
with $^\circ$Cl [441], flue gases of coal, biomass and garbage combustion [442], as shown in the
"global reactive chlorine emissions inventory" [441], HCl from chlorocarbons being a
significant part [443] in particular from CH$_3$Cl which is the largest, natural contributor to
organic chlorine in the atmosphere [444].

3.   They estimate that the global production rate of 1785 Tg yr$^{-1}$ of sea-salt
aerosol Cl$^-$ has to be doped with iron at a Cl$^-$/Fe(III) molar ratio of 51 meanwhile
we consider it has to be estimated at a molar ratio of 101 (according to 1.).

The calculations made with these limitative assumptions resulted in an iron demand of
56 Tg yr$^{-1}$ Fe(III) to obtain the desired CH$_4$ depletion effect [124].
Whereas, with the limitative assumption that there is no further Cl$^-$ source than sea-salt, the
calculations with a Cl$^-$/Fe(III) ratio of 101 results in a Fe(III) demand of only 18 Tg yr$^{-1}$.

ISA can be produced from pyrogenic iron oxides according to method I (see chapter 7).
Pyrogenic oxides have particle sizes lower than 0.1µm. Diameters of the NaCl-diluted ISA
particles of the Wittmer tests [124] are round about 0.5µm. This confirms the test results of
Wittmer et al. as calculation basis without any cut.

But Wittmer et al. made two other limitative assumptions:

4.    ISA has the same particle size and corresponding surface range as sea-salt;

5.    ISA has the same residence time as sea-salt aerosol in the troposphere."

According to their coarse aerosol particle range, the residence time of sea-salt particles in
the troposphere is inferior to 1 day [445] while the artificial ISA particles with diameters lower
than 0.5 µm have residence times in the troposphere of at least 10 days up to several weeks
[446, 447].

Known salt aerosol generation methods by vapor condensation or nebulization [448, 449]
allow not only the flame descending ISA type 1 [141], but also the condensation and
nebulization descending ISA variants 2 and 3 (see chapter 7) to be produced with aerosol
particle diameters between 0.1 and 0.01 µm. Diameters of salt aerosol particles according to
these physical aerosol generation methods are up to, or more, than one order of magnitude
smaller than of those used in the experiments by Wittmer et al. [124].

Analogue to CCN behavior in cloud processing [113] most of the small-sized ISA particles
are protected by their small sizes from coagulation or coalescence with sea-salt aerosol
particles. This effect prevents ISA from leaving the optimum active atomic chlorine emission
conditions: low pH and low particle diameter range.

The residence time difference of more than one order of magnitude in comparison to sea-salt
aerosol further reduces the Fe demand for ISA production from 18 Tg yr$^{-1}$ to less than
1.8 Tg yr$^{-1}$.

6.    The properties of the ISA particles produced by the most preferred ISA
  method variant are explained in chapter 4. Their difference to the NaCl-diluted ISA
  tested by Wittmer [124] are: ISA particles are made of $FeCl_3$ x $nH_2O$ undiluted by
  NaCl, or FeOOH coated by $FeCl_3$ x $nH_2O$ undiluted by NaCl [439, 450]. The $Cl^-$/Fe(III)
  molar ratios of $FeCl_3$ x $nH_2O$ are at 3 or even lower. The $Cl^-$/Fe(III) molar ratio of
  typical ISA particles is at least 30 times smaller than the molar $Cl^-$/Fe(III) ratio of 101
  of the tested ISA by Wittmer [124]. This reduces the Fe demand for ISA production
  again at least by 1 order of magnitude from <1.8 Tg yr$^{-1}$ to about <0.2 Tg yr$^{-1}$.

Wittmer et al. [124] considered only sea-salt aerosol particles as transport vehicles for ISA
and as only possible contact medium to gain chloride ions as °Cl source. It is well known that
coal combustion is a major source of active chlorine [441-443], as well as iron [78, 79, 83,
451], thus both iron and chlorine are jointly issued by other mechanisms and sources.

As stated in our chapter 6.2 below point 5, sea salt aerosol has residence times in the troposphere lower than one day according to its coarse particle diameters without any possible bridging of intercontinental distances.

In reality the chloride transfer between sea-salt aerosol particles and ISA particles may take place without any touch or coagulation, because the troposphere is an acidic environment. Troposphere is a source of organic and inorganic acids which are in permanent contact with the sea-salt aerosol. The acid ingredients in contact with sea spray produce HCl. Further ISA is produced by combustion and is elevated by flue gas plumes: acid precursors such as $SO_2$ or $NO_x$ are in higher concentrations within the flue gas plume comparing to the tropospheric environment. The acids generated by flue gas plume produce additional HCl by reaction with the sea-salt aerosol [167]. As a result, ISA and ISA precursors may absorb any chloride requirement via HCl vapor from the sea-spray source by itself [127].

Additionally to the °Cl emission increase with increasing iron concentration in the tested aerosols, the results of Wittmer verify an increase in °Cl emission with decreasing pH [124]. According to Wittmer and Meyer-Oeste [439, 450], oxidic ISA aerosol particles may be generated free from any pH-buffering alkaline components. This hampers their pH decrease by air-borne HCl to the optimum pH around pH 2. Sea-salt buffering of the absorbed HCl [452] by the alkali and earthen alkali content of sea-salt aerosol can occur only by coagulation, most probable in a minor ISA particle fraction but not in the bulk. From the beginning of its action in the troposphere, ISA keeps in the optimum °Cl emission mode: low pH, and high iron concentration levels.

Preferred ISA is produced by the ISA method variant 1 or variant 3 as described in chapter 7. Hence, ISA are composed of particles made by flame pyrolysis or iron salt vapor condensation. The mentioned ISA particles have diameters of $^1/_{10}$ of the particle diameters of the Wittmer tests. These ISA particles have optimum chlorine activation efficiency:

- In an appropriate chloride dotation or chloride delivering environment;
- At a pH <2;
- If they are emitted above the tropospheric boundary layer.

Then the Fe demand may fall up even shorter than the calculated 0.2 Tg Fe yr$^{-1}$ due to their far extended surface area and far extended residence time in the atmosphere.

It has to be noted that this ISA demand calculation result refers only to the ISA cooling property according to $CH_4$ depletion; further cooling properties according to cloud albedo, depletion of $CO_2$, black and brown aerosol, ozone decrease and further causes are still kept unconsidered.

Further oxidation activity on GHGs and aerosols are induced by the °OH generation activity of ISA: volcanic eruption plumes contain high concentrations of °Cl plus °OH [152] and are characterized by decreased $CH_4$ concentrations [153]. Co-absorption of $H_2O$ and HCl is the main reason of the generation of volcanic ash particle coats containing soluble Fe salts
originating from insoluble Fe oxides and Fe silicates [453, 454]. Gaseous HCl from the
eruption plume entails Fe chlorides covering the surfaces of volcanic ash particles [455].
Therefore, it is reasonable that photolysis of those chlorides is the origin of both: °Cl and °OH
generation in volcanic plumes.

Hydroxide radical °OH can change from the liquid aerosol phase into gaseous phase [169].
But by far, not as easy as °Cl can. Indeed, the Henry's law solubility constant of °OH is about
one order of magnitude higher than that of °Cl and is in the same range than that of $NH_3$
[166]. But when their hygroscopic water layer shrinks in dry air or by freezing, ISA particles
might act as °OH emitters. These additional °OH emissions might further increase the $CH_4$
oxidation potential of volcanic ash or artificial ISA and thus reduce even more the Fe demand
for ISA, though this has not been tested yet, it cannot be ruled out.

In order to take care not to overstep the cooling effect too far, a reasonable goal might be to
start the ISA method with a global ISA emission of 0.1 Tg Fe $yr^{-1}$. This quantity corresponds
to the magnitude of the actual Fe input from the atmosphere into the oceans under the form
of soluble salt, which is estimated to be from 0.1 up to 0.26 Tg $yr^{-1}$ [74, 80, 456]. Doubling or
even tripling of this input quantity by the ISA method is of easy technical and economic
feasibility as will been seen in chapter 7.

**7. The ISA method: how to increase artificial iron emissions**

Preceding calculation evidenced that the ISA method has the potential to cut back the rise of
$CH_4$ and $CO_2$ and, vice versa, the small decline of atmospheric oxygen content [457, 458]
because it acts by a bundle of chemical and physical means. The ISA method might retard,
stop or even help to restore these GHGs contents to pre-industrial levels. By the ISA method,
doubling or tripling of the ISA level in the troposphere seems to be possible by feasible
technical and economical means.

Since 2004 proposals have been published [141, 439, 450, 459, 460] to modify combustion
processes and flue gas emissions in order to use them as ISA plume emission sources in the
troposphere, by traffic and power generating combustions and their warm uplifting flue gases.
Predestined for the ISA method are any hot flue gas plumes emitted by ship and air traffic,
fossil and sunshine power.

At least three variants of ISA production are proposed:

• Variant 1: Emission of flame pyrolytic FeOOH aerosol with particle diameters smaller
than 100 nm [461, 462] as ISA precursor by co-combustion of organic iron or carbonyl
iron additives with liquid or gaseous fuels, or heating oils combusted in ship or and jet
engines, or by oil or gas combustors. Co-combustion of iron compounds is a possible measure in coal power stations and mixing the ISA precursor containing  combustion flue gas to the coal combustion flue gas after the dry flue gas cleaning stage. Useful side effects of iron additives are fuel efficiency optimization and soot emission minimizing [223, 224, 463, 464]. The emitted FeOOH aerosol plumes convert immediately into the ISA plume after leaving the emission sources, due to the high reactivity of flame pyrolytic Fe oxides. The period to cover the flame pyrolytic FeOOH particle surface by HCl absorption from the gaseous phase with Fe(III) chlorides is several times shorter comparing to the generation of iron chlorides from natural iron oxide minerals in loess dust particles [452, 465].

• Variant 2: Injection of vaporous ISA precursor iron compounds such as $FeCl_3$ into a carrier gas. By contacting the carrier gas and/or the atmosphere the vaporous iron compounds condenses and/or converts by physical and/or chemical means directly into ISA. Contrary to all other ISA precursors, the sunlit $FeCl_3$ vapor is photo-reduced by concomitant generation of °Cl [466]. Thus methane depleting °Cl emission can start even before this ISA precursor has changed into hydrated $FeCl_3$.

• Variant 3: Injection of ultrasonic nebulized aqueous $FeCl_3$ solution as ISA precursor into a carrier gas. By water evaporation from the aerosol droplets ISA is generated.

The preferred heights of ISA plume generation in the troposphere are 1000 m above ground or higher altitudes in order to pass the boundary layer. There, the ISA plumes have optimum conditions to spread over sufficient life-times. The necessary buoyancy to lift up the ISA plumes can be regulated by controlling their carrier gas temperatures. Uplift towers [467], vortex generators [468] or tethered balloons [469, 470] are preferential means to direct ISA by carrier gas uplift to said heights.

The primary ochre colored FeOOH aerosol particles emitted by ISA method I have diameters of <0,05 µm. According to previous studies iron oxides are strong absorbers at visible wavelengths and might play a critical role in climate perturbation caused by dust aerosols [108, 109]. But this effect is not applicable to the ISA methods FeOOH aerosol because it is emitted by parallel generated flue gas plumes containing $SO_2$ and $NO_x$ as sulfuric and nitric acid generators. Due to their small diameter dependent high surface area the aerosol particles immediately react with HCl. HCl is generated by the reaction between sea-salt aerosol and flue gas borne acids. Primary reaction product is the orange colored $FeCl_3$ aerosol: ISA. But the day time sun radiation bleaches ISA by $FeCl_2$ and °Cl generation; the night time re-oxidation of ISA plus HCl absorption regenerates $FeCl_3$ again. $FeCl_2$ is colorless at low humidity; pale green at high humidity.

Provision of the phytoplankton to optimize its growth with further nutrients such as Mn, Zn, Co, Cu, Mo, B, Si and P by the ISA method is possible by at least the variants 1-3 of the ISA method by co-combustion, co-condensation or co-nebulizing.

Global fixing regulations of GHGs emission certificate prices, values, and ISA emission certificate credit values would be simple but effective measures for the quickest world-wide implementation of the ISA flue gas conditioning method.

Anderson [471] reminded that of the 400 IPCC scenarios that keep warming below the Paris agreement target, "*344 involve the deployment of negative emissions technologies*", which he qualifies of "*speculative*" or requiring geoengineering.

A large part of the research devoted to climate engineering methods concerns SRM (sunlight reduction methods), such as mimicking the effects of large volcanic emissions by adding sulfates aerosols into the stratosphere as suggested for instance by Crutzen [242].

Numerous other types of particles have been suggested for these aerosols for instance titania by Jones [472]. But SRM only buys time and has numerous drawbacks.

On the one hand, SRM did not address the main cause of global warming (GHG emissions), nor prevents ocean acidification. On the other hand, several CDR technologies do, but their costs are much larger than SRM and the scale requested poses many technological challenges, for instance "*scaling up carbon dioxide capture and storage from megatons to*

*gigatons*" [473].

Very few CDR methods without emission of disadvantageous pollution are known. One of those is the Terra Preta method: it is characterized by the mixing of grinded bio-char into agricultural soils. The climate relevancies of this method are sustained fixation of former $CO_2$

carbon, minimizing fertilizer consumption and $N_2O$ emission reduction from the fertilized

Terra Preta soils. Char has similar properties within the soil environment than humic substances, but in the environment, char is resistant against oxidation.

Comparing the Terra Preta method to other CDR methods such as fertilizing the ocean by micro nutrients, results in lower specific material expenses by CDR methods per unit of $CO_2$

removed from the atmosphere [474]. The ISA method we propose is a member of this CDR

group, thus this result is also valid. in addition the further climate effects of the ISA method (such as depletion of $CH_4$, tropospheric ozone, and soot, plus cloud whitening) reduce the specific material expense level. Furthermore, the ISA method mimics a natural phenomenon (mineral iron-dust transport and deposition) and only proposes to improve the efficiency of an already existing anthropogenic pollution. Myriokefalitakis et al. [475] estimates that "*The*

*present level of atmospheric deposition of dissolved Fe over the global ocean is calculated to*

*be about 3 times higher than for 1850 emissions, and about a 30% decrease is projected for*

*2100 emissions. These changes are expected to impact most on the high-nutrient–low-*

*chlorophyll oceanic regions.*" Their model "*results show a 5-fold decrease in Fe emissions*

*from anthropogenic combustion sources in the year 2100 against in the present day, and*

*about 45% reduction in mineral-Fe dissolution compared to the present day*". Meanwhile the model used by [54] predicts by 2090 an iron supply increase to HNLC surface waters especially in the eastern equatorial Pacific attributed by the authors to changes in the meridional overturning and gyre-scale circulations that might intensify the advective supply of iron to surface waters. Furthermore, several authors [77, 87, 476-478] point out that both glacial and deep-water Fe sources may increase with continued climate warming due to Fe input from other sources, such as shelf sediments, melt water, icebergs, rivers, surface water runoff and dust input.

Recently Boyd and Bressac [67] suggested starting rapidly tests to determine efficiency and side effects of CDR ocean iron fertilizing methods, and analyzed possible geopolitical conflicts together with some other geoengineering methods [479].

Several experts, for instance Hansen et al. [6], expressed recently the urgent warning that mankind has only short time left to address and control climate warming. As a consequence mankind ought to find out as soon as possible climate controlling matter which might generate the most effective and reversible climate cooling effects within the shortest period. Lifetime of ISA emissions in the troposphere are much shorter than that of sulfates in the stratosphere. Of course, such tools and agents have to be rapidly evaluated against side-effects to ecosystems, human health, and last but not least their economic burdens.

**8. Interaction of the ISA method with further measures to protect the environment**

According to Wittmer & Zetzsch [127] elevated HCl content in the atmosphere triggers the methane depleting coating of oxidic ISA precursors by photolytic active Fe(III) chlorides. Any measure triggering the reduction of the HCl content of the atmosphere would impair the effectiveness of the ISA method based on this kind of method.

In this sense all kind of measures to reduce the sulfur and $NO_x$ content of the flue gas content of gaseous, liquid or gaseous fuels belongs would decrease the effectiveness of oxidic ISA precursors, as the S and $NO_x$ oxidation products sulfuric acid aerosol and gaseous nitric acid are the main producers of HCl by changing sea salt aerosol into sulfate and nitrate aerosol. Even the measures of reducing the energy production from fuel burning by changing to wind and photovoltaic energy would reduce this HCl source.

Sea salt aerosols produce HCl after contact with organic aerosol and organic volatile matter as the latter generates acid oxidation products from the latter such as oxalic acid [150, 480, 481]. A large fraction of organic aerosols and secondary organic aerosols originate from anthropogenic sources such as combustions. The change to wind and photovoltaic energy would reduce this HCl source.

The proposed CE measure of producing sulfuric acid aerosol within the stratosphere by inducing an albedo increase would increase the HCl content, during contact of the precipitating acid aerosol with tropospheric sea salt aerosol. Even the proposed CE measure of increasing the sea salt aerosol content of the troposphere by artificial sea salt aerosol as
cloud whitening measure could be used as ISA method trigger if flue gas is used to elevate
the sea salt aerosol.

**9. Discussion**

In order to fight global warming, this review proposes to enhance the natural actions of Cl
atoms in the troposphere, together with the synergistic action of iron in the atmosphere,
ocean, oceanic sediment and land compartments, as a climate engineering method. The
main results expected are a diminution of long lived well mixed atmospheric methane and
carbon dioxide, but the diminution of local short lived tropospheric ozone is also possible, as
well as effects on the Earth albedo, restoration of the oxygen flux into the deep ocean basins,
organic carbon storage, etc.

The most important actor in the process of $CO_2$ C transfer from atmosphere into the Earth
interior is the carbonate C precipitation in the crust rocks and sediments below the ocean.
The ocean crust acts like a conveyor belt between crust evolution at MOR and its subduction
zones into the mantle. Transported medium are carbonate C, small amounts of organic C,
ocean salt, ocean water and sediments. This process is part of the homeostasis of the
planet. Disturbances of this system part are induced by stratification processes within the
ocean basins caused by density differences between different layers of the water column.
Most stratification events are induced by climate warmings. Any of these homeostasis
disturbances are removed by the system within geological time scales. Signs of such
disturbances are more or less prominent events of extinction and of elevated organic C
content in the ocean sediments. Because the recent climate warming will induce a new
ocean stratification event, mankind ought to stop it. Like several interglacial stratification
events in the glacial periods, the actual stratification is also induced by increasing melt water
discharge. The past interruptions of the interglacial climate warmings teach us, that the
interruption events were accompanied as a rule by dust events. As demonstrated, the
climate cooling effects of these dust events are induced by the chemical and physical actions
of ISA.

In high-nutrient, low-chlorophyll oceanic areas, where the contribution of atmospheric
deposition of iron to the surface ocean could account for about 50% of C fixation, as well as
in oceanic nitrogen-limited areas, where atmospheric iron relieves the iron limitation of
diazotrophic organisms (thus contributing to the rate of N fixation), atmospheric deposition of
iron has the potential to augment atmospherically supported rates of C fixation [482] and thus
"cool the Earth" by removing $CO_2$ from the atmosphere.

Maybe the iron atmospheric deposition over terrestrial landscapes and wetlands has similar effects? Are there possible benefits of atmospheric deposition of soluble iron over the continents, where iron deficiency in plants occurs over 30% of them which are high pH calcareous soils that make soil Fe unavailable for plants [395]? Iron deficiency induced chlorosis in plants can be solved by addition of soluble iron complexes to the soil, or by foliar application of sprays containing mineral iron (for instance $FeSO_4$) [396] or iron chelates (Fe-EDTA among others) [399]. Iron, sulfate and several organic iron complexes such as iron-oxalate are known constituents of atmospheric dust [74], but unfortunately no published work was found about possible effects on plant chlorosis by foliar deposition of soluble iron from atmospheric dust.

We did not find studies about the impacts of atmospheric iron nutrient deposition on terrestrial ecosystems productivity. More research is needed to continue to enhance our understanding of the possible benefits of the iron cycling in freshwater and terrestrial landscape environments, as well as in atmospheric and sediment environments, in particular on its numerous potential capacities to fight global warming. The cooling effects of ISA and iron reviewed in this article already provide insight into the progress made on understanding the iron cycles from a range of perspectives.

There is abundant literature on the many geoengineering methods that have been proposed to *"cool the Earth"* [483, 484]. In particular, the injection of sulfate aerosols into the stratosphere is the most studied method, as it mimics the episodic action of natural volcanoes [163, 387]. Injected particles into the stratosphere reduce the radiative balance of Earth by scattering solar radiation back to space, so several types of particles are envisioned with a wide range of side-effects [472].

The literature also describes many options to deliver sulfates, their precursors (or other particles) to the stratosphere [469]. For instance, airplane delivery of the sulfate aerosols by the kerosene combustion process requires military jets due to commercial aircrafts limited altitude of 10 km (30,000 feet), and not the 20 km requested [469].

In the case of ISA, the altitude needed to *"cool the Earth"* is much lower: it is in the troposphere and the total quantities to deliver are 1 order of magnitude smaller. So air travel is a possible means for ISA delivery. But the global jet fuel consumption is only about 240,000 t yr$^{-1}$. Even by assuming the very high emission rate of 1 kg ISA precursor iron per ton of jet fuel, only 240 t yr$^{-1}$ might be emitted. This seems far away from the order of magnitude of the target ISA emissions.

From the many other possible delivery strategies envisioned for SRM by stratospheric aerosols, many are not suited for ISA, such as artillery, missiles and rockets [469]: it will be cheaper with less pollution to use the flue gas of a reduced number of thermal power plants. That might be efficient enough to deliver the artificial iron aerosol needed over the boundary layer, in order to the aerosols to stay several days or weeks in the troposphere and become
widely distributed [485].

According to Luo [79], deposition of soluble iron from combustion already contributes from 20
to 100% of the soluble iron deposition over many ocean regions.

As an example we calculated the possible production and emission of the ISA precursor
FeOOH aerosol using the flue gas of the German power station Niederaußem; with the input
of 25 million t yr$^{-1}$ of lignite (brown coal), this power station produces 3,600 MW.

According to ISA production variant 1 (chapter 6) the ISA precursor FeOOH aerosol may be
produced by burning of a ferrocene (Fe(C$_5$H$_5$)$_2$) oil solution containing 1% ferrocene in a
separate simple oil burner. The hot oil burner flue gas containing the ISA precursor FeOOH
aerosol is injected and mixed into the cleaned power station flue gas. The power station flue
gas emission rate is calculated to 9,000 m³ flue gas per ton of lignite. As the ISA precursor
containing flue gas will be elevated to heights of more than 1000 m above ground, dust
levels of the ISA precursor FeOOH aerosol of 20 mg m$^{-3}$ flue gas seem to be acceptable.
This allows a quantity of 180 g of FeOOH per ton of combusted lignite (9000 m³ t$^{-1}$ x 0,02
g m$^{-3}$). At a lignite quantity of 25 million t yr$^{-1}$, this corresponds to 4,500 t FeOOH yr$^{-1}$. FeOOH
has an iron content of 63%. This corresponds to a possible iron emission of 2,831 t yr$^{-1}$ and a
possible ferrocene consumption of 9,438 t yr$^{-1}$.

Corresponding to this calculation about 100 of such huge power stations should have the
ability to produce the sufficient ISA quantity of an equivalent of 200,000 to 300,000 t Fe yr$^{-1}$.
Further optimization of the cooling capacity of the produced ISA is possible by a co-emission
of HCl, for instance by co-burning of an organic HCl precursor.

This example illustrates that ISA emission at only 100 power stations, or any similar ISA
emission measures, is quite feasible compared to the alternative of CCS by CO$_2$ capture
from the flue gas of 40 Gt yr$^{-1}$, compression of the CO$_2$ until the liquid state, followed by
transportation and CO$_2$ storage by injection into underground rock aquifers or into old and
depleted fossil fuel reservoirs.

In order to increase the effectiveness of the buoyancy capacity of the power works the usual
wet cooling tower might be replaced by a dry cooling tower to mix the dry and warm air
emission from the cooling tower with the hot flue gas as additional buoyancy and due point
reduction mean. Further the flue gas buoyancy may increase by increasing the flue gas
temperature. This or other simple techniques to realize ISA plumes may be used within the
troposphere.

One alternative delivery method that seems promising and can easily be adapted to ISA
method, is the use of tethered balloons [486], and will cost much less as 1 or 2 km altitude
will be sufficient for ISA emissions, requiring much lower pressures in the pipes than for SO$_2$
delivery at 20 km for the geoengineering method. Technical and economic feasibility have already been studied for the SPICE project [470] which was planning to release sea water
spray at 1 km altitude.

Furthermore, as iron emissions only stay in the troposphere for weeks compared to SRM
sulfates in the stratosphere that stay 1 or 2 years. In case any unintentional side effect or
problem occurs, stopping the emissions is rapidly possible and the reversibility of its effects
are much shorter than for solar radiation management by sulfates aerosols.

Other geoengineering strategies to cool the Earth, such as carbon dioxide removal by iron
fertilization [64] have several pros and cons, such as localized release, less dispersion, in a
form that is not readily bio-available, resulting in restricted cooling effects and high expenses.
The idea of ocean fertilization by iron to enhance the $CO_2$ conversion by phytoplankton
assimilation came up within the last two decades. Proposed was the mixing of an iron salt
solution by ships into the ocean surface. This idea was debated controversial. Example of
this debate is the discussion between KS Johnson et al. and SW Chisholm et al. [68, 69].
Deeper insight into this debate is given by Boyd and Bressac [67].

The iron fertilization procedure tests done so far had been restricted to relatively small ocean
regions [51, 52, 487]. These tests produced iron concentrations orders of magnitude above
those produced by natural ISA processing which are in the single decadal order of milligrams
of additional dissolved iron input per square meter per year. In this sense the ISA method is
quite different from "iron fertilization". As known from satellite views, phytoplankton blooms
induced by natural dust emission events from the Sahara, Gobi and further dust sources,
there is no doubt about the fertilizing effect of iron. Meanwhile this kind of natural iron
fertilization enhancing the transfer of $CO_2$-Carbon into organic sediment carbon via the
oceanic food chain seems to be un-contradicted and accepted [6].

The ISA method allows the use of the same atom of iron several times by catalytic and
photocatalytic processes into the atmosphere, with different cooling effects (such as albedo
modification and enhancement of the methane destruction) and then reaches the oceans,
with further cooling effects such as the enhancement of $CO_2$ carbon fixation.

Harrison [488] estimates that a single ship based fertilization of the Southern Ocean will
result only in a net sequestration of 0.01 t Carbon $km^{-2}$ for 100 years at a cost of US$457 per
ton of $CO_2$, as the economic challenge of distributing low concentrations of iron over large
ocean surface areas, has been underestimated [489], as well as the numerous loss
processes (i.e.: soluble iron loss and organic carbon that do not sink till the bottom of the
ocean) resulting in reduced net storage of carbon per $km^2$ of ocean fertilized.

Figure 7 summarizes many of the cooling effects of the ISA method.

[Figure]

**Figure 7**. Summary of the principal cooling effects of the proposed iron salt aerosols method. The organic C / carbonate C burial ratio in sediments and bedrock increase after ISA method start, until a maximum. Then this ratio begins to decrease as soon as the vertical current components in the ocean basin begin to act. Then the ratio arrives to a very low permanent level, while the total of buried C arrives at a permanent maximum level when the maximum vertical mixing conditions have been obtained by the ISA method.

Why does ISA appear to be more effective than ocean iron fertilization? For ocean iron fertilization several tons of Fe(II) are dispersed in a short time (hours) over only some $km^2$ of ocean with several drawbacks and a massive algae bloom can change the local biotopes. Meanwhile ISA releases iron continuously, reaching the entire 510 million $km^2$ of Earth surface. The current iron inputs (in the form of soluble salts) into the oceans are estimated between 0.1 and 0.26 Tg $yr^{-1}$ [74, 80, 456]. As water covers nearly 72% of Earth surface (362 million $km^2$), if ISA delivers 1 Tg Fe $yr^{-1}$ evenly distributed (in addition to natural and anthropogenic current emissions), which is 4 times more than the expected needs (chapter 5.2), on average every $km^2$ of ocean receives 5.4 g Fe $km^{-2}$ $day^{-1}$ ($^1/_{510}$ t Fe $km^{-2}$ $yr^{-1}$).

**10 Conclusion**

At ideal circumstances the ocean acts as an optimum transport medium for $CO_2$ carbon from the atmosphere into the ocean crust. Such circumstances are present when the vertical cycling components between ocean surface and ocean bottom are undisturbed.

Any stratification event disturbs this cycling and interrupts the $CO_2$ transport. Climate warming can induce stratification events by producing huge amounts of melt water. Recent research found signs of at least regional development of a beginning stratification.

The numerous climate cooling effects of natural dust show in this review, according to its soluble iron content, demonstrate that dust is of a central significance as steering element of this carbon transport from the atmosphere into the ocean crust.

This review article demonstrates the enormous effects of atmospheric iron dusts and focuses first on the tropospheric aerosol particles composed partly of iron and chloride (iron salt aerosols ISA), showing their cooperation and interactions with several components of the atmosphere for instance with $CH_4$, as the chlorine atom is responsible for the removal of a significant part of this GHG (3 to 4 % of $CH_4$) in the troposphere [118, 119]. This article summarizes a dozen of other possible direct and indirect natural climate cooling mechanisms induced by the iron biogeochemistry in all the Earth compartments: atmosphere, oceans, land (surface, soil), sediment and crust.

These dozen possible climate cooling effects due to the multi-stage chemistry of iron within the atmosphere, hydrosphere, geosphere and lithosphere are described all together for the first time and are summarized in table 3, which shows the most probable climate cooling effects of ISA. They include the ocean fertilization effect which allows enhanced algal and phytoplankton growth, which removes mineral $CO_2$ from the atmosphere and transforms it in organic carbon, a part of which can sink to the bottom of the oceans and be stored for long periods of time by different mechanisms that are described.

**Table 3**: principal effects of the ISA method proposed - or its natural equivalent - and their probable effect on the different biosphere compartments.

| Compartment | Locality and/or action | Effect | Most probable cooling efficiency | Time delay between cooling on-set or off-set after ISA method start or stop |
|---|---|---|---|---|
| Troposphere | Boundary layer and lower | Cloud albedo increase | +++ | <1 yr |

| Location | Compartment/organism | Process | Strength | Timescale |
|---|---|---|---|---|
| troposphere | | Methane and VOC depletion | +++ | <1 yr |
| | | Black and brown carbon precipitation | ++ | <1 yr |
| | | Ozone depletion | ++ | <1 yr |
| **Continent** | Forests and further primary producer | Organic  C burial increase by assimilation increase | + | <5 yr |
| | Wetlands, marshes, peat bogs, lake sediments | Methane emission decrease by methanogenesis inhibition | +++ | <5 yr |
| | Desert surfaces | Methane and VOC depletion | + / - | <1 yr |
| **Ocean and ocean sediment aquifer at the ocean bottom** | Phytoplankton and the further food chain links | Organic and Carbonate C burial increase by assimilation increase | 1)  ++++ | <1 yr |
| | | | 2)  + | <1 yr |
| **Ocean crust aquifer** | Activation of the ocean basin vertical cycling | Carbonate C burial increase in the ocean crust rock | 3)  +++++ | >10 yr |
| | | | 4)  + / +++ | >10 yr |

1) The euxinic and alkaline bottom water of the stratified ocean have no oxidation and calcite solution capacity, thus produce a high burial rate of organic sediment C and carbonate C

2) The oxic, hydrogen carbonate and $CO_2$-containing bottom water of the well-mixed ocean have high oxidation capacity and high calcite dissolving capacity, thus produce a low burial rate of organic and inorganic Sediment C

3) The high inorganic C load of the oxic, hydrogen carbonate and $CO_2$-containing bottom water of the well-mixed ocean comes to total precipitation within the alkaline and reducing crust aquifer, thus produce a very high burial rate of inorganic C and small amounts of organic C precipitation

4) The euxinic and alkaline bottom water of the stratified ocean has low content of dissolved inorganic C and contains methane C up to saturation, thus produce low to medium C burial rate during cycling through the crust aquifer.

In order to explicitly handle the interaction of climate and biogeochemistry, the complex interactions between climate and the cycles of C, N, P, $H_2O$ and micronutrients call for models that integrate global biogeochemical cycles of terrestrial, oceanic and atmospheric components of the biosphere.

While the iron biogeochemical cycle between the atmosphere and the ocean is considered in numerous publications, the treatment of key processes and feedbacks within the terrestrial compartment has been rather limited, and further development is urgently needed.

Mineral dust aerosols containing iron and other important nutrients or micro-nutrients are well studied components of the iron biogeochemical cycle in the atmosphere and the oceans, but the absence of recent bibliography about the full iron biogeochemical cycle over terrestrial landscapes, soils, wetlands and all clear water compartments (glaciers, ice, snow, lakes, and groundwater) points out a lack of up-to-date overview. In our opinion, the atmospheric chemistry models need to incorporate all relevant interaction compartments of the Fe-cycle with sun radiation, chlorine, sulphur, nitrogen, oxygen, carbon and water in order to model the several planetary cooling effects of the iron cycle.

Acid rain sulphate ($SO_4^{2-}$) deposition on peatlands and wetlands from natural sources (volcanoes), or anthropogenic sources (fossil fuel combustion) is a known suppressant of $CH_4$ production [490, 491] and emissions [492-494] and may be an important process in terms of global climate. The importance of the Fe input associated with anthropogenic aerosol deposition in terrestrial biogeochemistry deserves further investigation as well as the possible impacts of a drastic diminution of anthropogenic iron and sulfates emissions from combustion processes expected by 2050 to satisfy the Paris climate agreement.

This review completes the previous global iron cycle visions [50, 52, 74, 97, 98, 495-497] and advocates a balanced approach to make profit of the iron cycle to fight global warming by enhancing natural processes.

Climate cooling by natural ISA involves the troposphere, dry solid surfaces, ocean waters, ocean sediment, ocean crust and land. Several GHG factors are controlled by ISA: $CO_2$, $CH_4$, tropospheric $O_3$, black carbon, dust, cloud albedo, and vertical ocean mixing.

Using mineral dust as a natural analogue tool, this article proposes to enhance the natural ISA in order to raise and heighten the cooling impacts of at least two of the dozen natural effects found: i.e. $CH_4$ removal by tropospheric °Cl and $CO_2$ removal by soluble-Fe ocean fertilization.

The ISA method proposed is feasible, probably with few to no-environmental side-effects, as it relates to chemical and/or physical combustion processes occurring currently. Actual iron production and coal combustion together with other combustions sources already release in the atmosphere a very significant part of the global bioavailable iron in the northern oceans: from 15% [80] to 80% [82, 83] depending on the iron solubility parameters taken into account.

The present level of atmospheric deposition of soluble Fe over the global ocean is evaluated to be about 3 times higher than for 1850 emissions [475], as increases in anthropogenic and biomass burning-emissions resulted in both enhanced Fe combustion emissions and a more acidic environment and thus more than double soluble Fe deposition (nearly 0.5 Tg-Fe yr$^{-1}$ nowadays versus nearly 0.2 Tg-Fe yr$^{-1}$ in 1850).

Inevitable reduction of aerosol emissions to improve air quality in the future might accelerate the decline of oceanic productivity per unit warming and accelerate decline in oceanic NPP [498]. Myriokefalitakis model projected results for 2100 indicate about a $\frac{1}{4}$ decrease in atmospheric deposition of soluble Fe, with a 5-fold decrease in Fe emissions from anthropogenic combustion sources (~0.070 Tg-Fe yr$^{-1}$ nowadays against ~0.013 Tg-Fe yr$^{-1}$ in 2100). These changes are expected to impact most on the high-nutrient–low-chlorophyll oceanic regions. According to Myriokefalitakis [475], in view of the importance of Fe as a micronutrient for marine ecosystems, the calculated projected changes in soluble iron emissions, requires the implementation of comprehensive mineral-Fe dissolution processes as well as Fe combustion emissions in coupled climate-biogeochemistry models to account for feedbacks between climate and biogeochemical cycles. This review shows that the effects on $CH_4$ of ISA and of anthropogenic Fe emissions in the troposphere also deserve to be taken into account.

According to Wang et al. [83], taking into consideration the relatively high solubility of anthropogenic iron, combustion sources contribution to soluble Fe supply for northern Pacific and northern Atlantic oceanic ecosystems could be amplified by 1–2 orders of magnitude. To stop global warming, we estimated the requirements in terms of ISA by extrapolation of experiments of iron catalyzed activation by artificial sea-salt aerosols [124, 127]. Our first estimations show that by doubling the current natural Fe emissions by ISA emissions into the troposphere, i.e. by about 0.3 Tg Fe $yr^{-1}$, artificial ISA would enable the prevention or even the reversal of GW.

The adjustable flue gas temperatures for different types of combustions are a means to lift the ISA plumes to optimal heights within the troposphere. Thus, we believe that the ISA method proposed integrates technical and economically feasible tools that can help to stop GW.

According to our remarks in chapter 2, the reactions of ISA in the troposphere are the most prominent results for a surface temperature decrease [439]. This stops further ice melting, which activates the different vertical ocean water movements. As a result, the dissolved $CO_2$ is then buried as carbonate C within the ocean bottom sediments and crust.

**Abbreviations:**

Carbon capture and storage: CCS; Cloud condensation nuclei: CCN; Global Warming: GW; Intergovernmental Panel on Climate Change: IPPC; Iron salt: IS; Iron salt aerosols: ISA; Humic-like substances: HULIS; Hydroxyl radical: °OH; Chlorine radical: °Cl; Bromine radical: °Br; Ligand: L; Methane: $CH_4$; Mid-ocean rift: MOR; Secondary organic aerosol: SOA; Thermohaline circulation: THC; Volatile organic compounds: VOC.

**Authors contribution**:

F.D. Oeste suggested the review idea and performed initial bibliographical search completed by R. de Richter. F.D. Oeste and R. de Richter prepared the manuscript and the figures with contributions from all co-authors. T. Ming and S. Caillol also contributed to structuring the manuscript, ideas, submitting bibliography and English corrections.

**Competing interests**

The authors declare that they have no conflict of interest.

**Acknowledgment:**

This research was supported by the Scientific Research Foundation of Wuhan University of Technology (No. 40120237) and the ESI Discipline Promotion Foundation of WUT (No.35400664).

The co-authors would like to thank both reviewers S.M. Elliott and Anonymous for their constructive and thoughtful reviews which greatly improved this manuscript, in particular chapters 5, 9 and 10. We also thank Rolf Sander and Cornelius Zetzsch for their constructive comments and Louise Phillips for grammatical corrections and re-reading.